# Recurrent humid phases in Arabia over the past 8 million years

Monika Markowska[1,2 ✉], Hubert B. Vonhof[1 ✉], Huw S. Groucutt[3,4 ✉], Paul S. Breeze[5], Nick Drake[5], Mathew Stewart[6], Richard Albert[7], Eric Andrieux[8], James Blinkhorn[9,10], Nicole Boivin[11,12,13], Alexander Budsky[14], Richard Clark-Wilson[15], Dominik Fleitmann[16], Axel Gerdes[7], Ashley N. Martin[2,17], Alfredo Martínez-García[1], Samuel L. Nicholson[1], Gilbert J. Price[18], Eleanor M. L. Scerri[3,4,28], Denis Scholz[19], Nils Vanwezer[11], Michael Weber[19], Abdullah M. Alsharekh[20], Abdul Aziz Al Omari[21], Yahya S. A. Al-Mufarreh[22], Faisal Al-Jibreen[21], Mesfer Alqahtani[21,23], Mahmoud Al-Shanti[22], Iyad Zalmout[24,25], Michael D. Petraglia[6,12,26 ✉] & Gerald H. Haug[1,27]

The Saharo-Arabian Desert is one of the largest biogeographical barriers on Earth, impeding dispersals between Africa and Eurasia, including movements of past hominins. Recent research suggests that this barrier has been in place since at least 11 million years ago[1]. In contrast, fossil evidence from the late Miocene epoch and the Pleistocene epoch suggests the episodic presence within the Saharo-Arabian Desert interior of water-dependent fauna (for example, crocodiles, equids, hippopotamids and proboscideans)[2–6], sustained by rivers and lakes[7,8] that are largely absent from today's arid landscape. Although numerous humid phases occurred in southern Arabia during the past 1.1 million years[9], little is known about Arabia's palaeoclimate before this time. Here, based on a climatic record from desert speleothems, we show recurrent humid intervals in the central Arabian interior over the past 8 million years. Precipitation during humid intervals decreased and became more variable over time, as the monsoon's influence weakened, coinciding with enhanced Northern Hemisphere polar ice cover during the Pleistocene. Wetter conditions likely facilitated mammalian dispersals between Africa and Eurasia, with Arabia acting as a key crossroads for continental-scale biogeographic exchanges.

Arabia is at the centre of the largest near-continuous chain of drylands on Earth; a harsh and often hyperarid belt stretching from the Sahara to the Thar Desert. The Saharo-Arabian Desert barrier limits animal dispersal, dividing Africa and Eurasia into the Afrotropical, Palaearctic and Indomalayan biogeographic realms, each characterized by distinct assemblages of plant and animal species and communities[10]. The permanence of this desert barrier serves as a major control on the delineation of these biogeographic realms, whereas climatic amelioration across the Saharo-Arabian region permits dispersal between them[11,12]. Consequently, this region is a 'transition zone', hosting complex faunal admixtures that exhibit African, Eurasian and South Asian features[10].

Fossil evidence from large mammals suggests reduced connectivity between Africa and Eurasia from the late Miocene epoch (11.7–5.3 million years ago (Ma)) onwards[11], limiting species admixture and exchange, and providing an explanation for the high degree of faunal endemism that developed in the Afrotropics[13]. Despite this, other fossil data, including the earliest evidence of hominin dispersals from Africa to Eurasia (*Homo* in Eurasia at around 2 Ma)[14] makes it clear that the Saharo-Arabian biogeographic barrier was temporarily permeable, although the palaeoclimatic context in which these early dispersals occurred remains unknown.

Recent research has suggested that drying across this desert barrier may have begun in the late Miocene with fully arid conditions in the

[1]Department of Climate Geochemistry, Max Planck Institute for Chemistry, Mainz, Germany. [2]Department of Geography and Environmental Sciences, Northumbria University, Newcastle upon Tyne, UK. [3]Department of Classics and Archaeology, University of Malta, Msida, Malta. [4]Institute of Prehistoric Archaeology, University of Cologne, Cologne, Germany. [5]Department of Geography, King's College London, London, UK. [6]Australian Research Centre for Human Evolution, Griffith University, Brisbane, Queensland, Australia. [7]Frankfurt Isotope and Element Research Center (FIERCE), Department of Geosciences, Goethe University Frankfurt, Frankfurt am Main, Germany. [8]Department of Archaeology, Durham University, Durham, UK. [9]Human Palaeosystems Group, Max Planck Institute of Geoanthropology, Jena, Germany. [10]Department of Archaeology, Classics, and Egyptology, University of Liverpool, Liverpool, UK. [11]Department of Archaeology, Max Planck Institute of Geoanthropology, Jena, Germany. [12]School of Social Science, University of Queensland, Brisbane, Queensland, Australia. [13]School of Environment and Science, Griffith University, Brisbane, Queensland, Australia. [14]Department of Geosciences, Landesmuseum für Kärnten, Klagenfurt am Wörthersee, Austria. [15]Department of Geography, Royal Holloway, University of London, Egham, UK. [16]Department of Environmental Sciences, University of Basel, Basel, Switzerland. [17]Institute of Earth System Sciences, Leibniz University Hannover, Hannover, Germany. [18]School of the Environment, The University of Queensland, Brisbane, Queensland, Australia. [19]Institute for Geosciences, Johannes Gutenberg University Mainz, Mainz, Germany. [20]Department of Archaeology, College of Tourism and Archaeology, King Saud University, Riyadh, Saudi Arabia. [21]Heritage Commission, Ministry of Culture, Riyadh, Saudi Arabia. [22]Geotourism Department, Saudi Geological Survey, Jeddah, Saudi Arabia. [23]Department of Anthropology, University of Pittsburgh, Pittsburgh, PA, USA. [24]Directorate of Geological Survey, Survey and Exploration Centre, Saudi Geological Survey, Jeddah, Saudi Arabia. [25]Museum of Palaeontology Research Museum Center (RMC), University of Michigan, Ann Arbor, MI, USA. [26]Human Origins Program, Smithsonian Institution, Washington DC, USA. [27]Department of Earth and Planetary Sciences, ETH Zürich, Zürich, Switzerland. [28]Present address: Human Palaeosystems Group, Max Planck Institute of Geoanthropology, Jena, Germany. ✉e-mail: monika.markowska@northumbria.ac.uk; hubert.vonhof@mpic.de; huw.groucutt@um.edu.mt; m.petraglia@griffith.edu.au

Sahara from at least 11 Ma[1] and hyperaridity in the northern Arabian margins starting at 9 Ma (ref. 13). This aridification appears in conjunction with an overall global expansion of tropical and subtropical deserts as a feature of long-term late-Cenozoic cooling. An acceleration in aridification is also evidenced at the Pliocene–Pleistocene transition and is associated with further cooling in the Northern Hemisphere, triggered by the intensification of glacial–interglacial cycles at about 2.6 Ma together with atmospheric circulation changes[15–17]. Global cooling also marked the start of a pronounced global shift in terrestrial environments[17], in which ecosystems transitioned from those dominated by plants following the $C_3$ photosynthetic pathway (for example, trees, herbs and shade-loving grasses) to those dominated by $C_4$ plants (for example, arid-adapted grasses)[18] (Fig. 2). Although the increased distribution of arid-adapted vegetation and, consequently, $C_4$ mammalian grazers, has often been attributed to higher aridity, another plausible explanation is the ability of $C_4$ plants to outcompete $C_3$ in many parts of the world under decreasing atmospheric carbon dioxide ($CO_2$) levels where $C_4$ photosynthesis is physiologically advantageous[19]. As there are limited terrestrial records of water balance in continental desert interiors, it accordingly remains challenging to assert whether changing precipitation regimes are responsible for the aforementioned shift in vegetation, and further, whether this aridification state was a permanent feature of the late-Cenozoic era. This leaves the driving mechanisms behind what appears to be a key late-Cenozoic aridity trend unresolved.

In Arabia, there is evidence for significant 'humid episodes' from middle-to-late-Pleistocene lake deposits in the Nefud Desert and the Empty Quarter (the largest area of continuous sand on Earth), which are frequently associated with vertebrate fossils and stone tool assemblages[3,8,12]. Wetter-than-present conditions are also evidenced in a late Miocene (7.7–7.0 Ma)[7] sequence of sandstones and mudstones in the Baynunah Formation, United Arab Emirates, which contain fossils of savannah-adapted mammals[4–6]. However, it is unclear whether this evidence for humidity was owing to increased precipitation in the Arabian interior itself or, for example, sourced distally from continental margins. Precise dating in southern Arabia (up to 1.1 Ma (ref. 9)) and in the Mediterranean Levant[20] of cave carbonates (speleothems), the formation of which is directly influenced by water balance, provides some of the first in situ evidence for more humid conditions on the continental margins of the Peninsula. The lack of direct, and precisely dated, hydroclimatic information from the continental interior before the middle Pleistocene epoch renders the climatic modulation of the Arabian Peninsula and the degree of permanence of its hyperarid interior largely unknown.

## Recurrent humid episodes

We recovered a set of speleothems from the Arabian interior that extends the region's terrestrial hydroclimate record to 8 Ma and reveals recurrent humid periods characterized by increased water availability and vegetation cover. This represents one of the longest palaeoclimatic records currently available for Arabia, as well as one of the longest speleothem palaeoclimate records globally. Our study area is in a hyperarid (approximately 104 mm yr$^{-1}$; Fig. 1) part of central Arabia, positioned under the descending arm of the Hadley Cell, north of Intertropical Convergence Zone (ITCZ) precipitation, and associated with semi-permanent anticyclonic high-pressure systems. The region sits between two major moisture-bearing systems: mid-latitude westerlies, which originate in the eastern Mediterranean and traverse the peninsula during boreal winter months (Fig. 1b); and monsoon-derived precipitation from the southwest, which is limited to the southerly parts of the peninsula during boreal summer (Fig. 1c).

Desert speleothems provide direct evidence of past humid climates, as their growth requires two general conditions: (1) sufficient regional precipitation, with previous estimates based on the distribution of

active and inactive speleothems in the Negev Desert being on the order of about 300 mm yr$^{-1}$ (ref. 20); and (2) the development of sufficient vegetation and soil–$CO_2$ concentrations to react with water to produce carbonic acid, dissolving the limestone bedrock and initiating speleothem formation[21]. A particular strength of speleothem archives is that they can be precisely and accurately dated using routine radiometric methods, enabling the production of chronometrically controlled, in situ records with geochemical proxies directly attributable to precipitation change. The preservation of speleothems in stable cave environments permits consecutive wet phases to be recorded over million-year timescales. So far, most speleothem studies utilize the uranium–thorium (U–Th) chronometer, which is limited to the past approximately 0.6 Myr. Although the uranium–lead (U–Pb) chronometer, suitable for dating beyond 0.6 Ma, is regularly used for precise age determinations in other archives, its routine application has only recently been exploited in speleothems[9,20,22,23]. Our approach combines the U–Th and U–Pb chronometers to determine the timing of humid phases in the Arabian interior from the late Miocene to the late Pleistocene. We present 74 radiogenic ages and show the frequency of ages from the two different chronometers through time (Fig. 2g). We utilized 22 representative individual speleothems of various morphologies, often with multiple growth phases separated (in some cases) by more than 1-Myr hiatuses (Extended Data Figs. 1–3), collected from 7 cave systems within a 10-km radius in central Arabia (Methods).

The oldest evidence for increased precipitation in our dataset occurred in the late Miocene from 7.44 Ma (Fig. 2) until 6.25 Ma (Fig. 3f). A period of further humid episodes occurred from 4.10 Ma in the early Pliocene until 3.16 Ma (Fig. 3f), just before the Pliocene/Pleistocene transition. Further speleothem humid episodes were identified in the early Pleistocene, between 2.29–2.01 Ma and 1.37–0.86 Ma (Fig. 3f). In the middle-to-late Pleistocene, speleothem growth is resolved at glacial–interglacial resolution, owing to the higher precision of U–Th dating (Methods). Growth intervals occurred during Marine Isotope Stages (MISs) 7, 11 and 15 coinciding with humid periods also identified in the Negev Desert[20] and in southern Arabia[9] (Extended Data Figs. 8 and 9). Speleothem growth in MIS 9 and MIS 13 was likely, as speleothem formation is observed in southern Arabia during these interglacial phases[9,24], but dating uncertainties prevent precisely constraining these growth phases. A period of gypsum and calcite precipitation occurred from 0.53 Ma to 0.06 Ma, culminating in only gypsum overgrowth after MIS 7, covering older calcite speleothem layers (Extended Data Fig. 5). Unlike calcite that forms via degassing and typically requires vegetation and a soil zone for the formation of carbonic acid which dissolves $CaCO_3$, gypsum speleothem formation is mostly controlled by evaporation and typically occurs in arid and semiarid regions. This suggests that despite increased recharge to groundwater reservoirs, only a slight increase in local precipitation occurred after MIS 7, which was insufficient to promote calcite speleothem formation[25] in central Arabia.

The 8-Ma speleothem record demonstrates recurrent wetter intervals, reflecting the alternating nature of wet–dry phases in central Arabia. Humid episodes in subtropical drylands such as southern Arabia are typically pulsed and short-lived (several to tens of thousands of years)[9,24] and in phase with Northern Hemisphere summer insolation maxima[9,26]. The amount of incoming solar radiation is predominantly influenced by 20-kyr orbital precession cyclicity, modulated by eccentricity[27], and to a lesser extent obliquity (40-kyr cycles), and is associated with meridional shifts in the African monsoon belt[26]. Output from global climate models project that increased summer monsoonal precipitation over the Arabian Peninsula occurs at peak Northern Hemisphere high-latitude insolation (for example, at 125 thousand years ago (ka)), as revealed by a series of time-slice experiments associated with different orbital configurations and global ice volumes[28]. Consequently, low-latitude insolation is an important pacer of humid episodes in the Arabian Peninsula, the frequency of which is also governed by glacial boundary conditions and the degree of warming in the North Atlantic region.

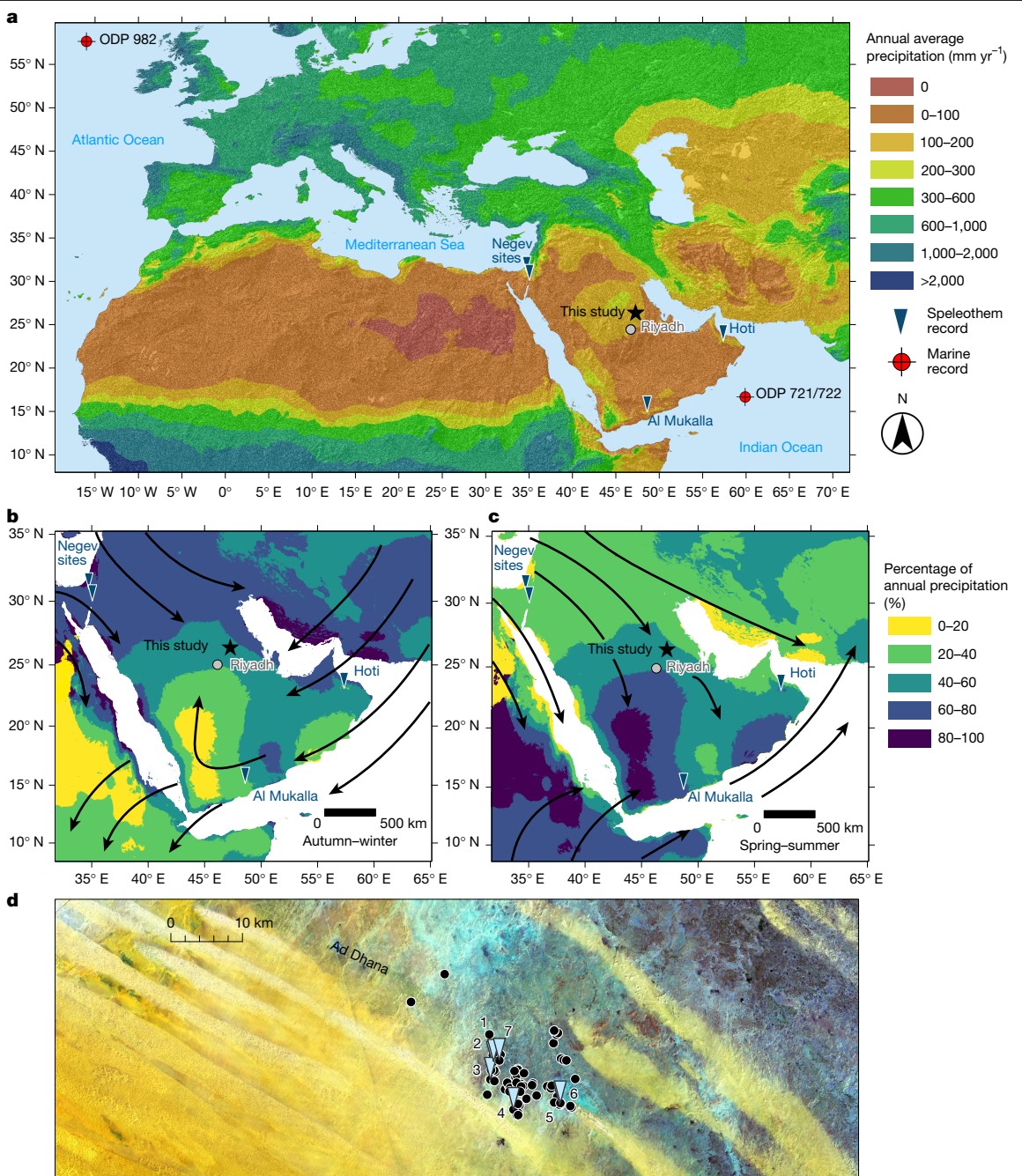

**Fig. 1 | Hyperarid Central Arabia is at the interface between two moisture-bearing systems. a**, Modern precipitation of the Arabian Peninsula, showing the location of our study site (black star) and other discussed speleothem sites, and Oceanic Drilling Program (ODP) sites 721/722 and 982. **b,c**, Percentage contribution of autumn and winter precipitation (September–February; **b**) and of spring and summer precipitation (March–August; **c**). The black arrows show general wind patterns. Data sourced from WorldClim dataset[43]. **d**, Detailed view (Landsat false-colour RGB, 7,4,1) of the region where surveys took place (surveyed caves represented as black points). Speleothems used in this study were retrieved from caves (*n* = 7) represented by blue triangles: 1, Kahf al alf Janah/جناح الألف كهف (1,000 Wings Cave); 2, Kahf at Tahaaleb/الطحالب كهف (Mossy Cave); 3, Kahf Assadaqah/الصداقة كهف (Friendship Cave); 4, Kahf Almorabbaa/المربع كهف (Murubbeh Cave); 5, EP19.8; 6, Kahf Alfakhamah/الفخامة كهف (Luxury Cave); 7, Kahf al Fondoq/الفندق كهف (Hotel Cave).

## Precipitation moisture sources

Over the past 8 Myr, there has been a shift in moisture-bearing systems delivering precipitation to central Arabia revealed by the isotopic ratio of oxygen-18 to oxygen-16 ($\delta^{18}O_{FI}$, ‰ Vienna Standard Mean Ocean Water (VSMOW)) and the isotopic ratio of hydrogen-2 to hydrogen-1 ($\delta^{2}H_{FI}$, ‰ VSMOW) of speleothem fluid-inclusion waters ('fossil dripwater'). This is important, as identifying the origin of moisture sources allows links to large-scale atmospheric circulation patterns, such as the Hadley

Circulation, revealing the key mechanisms driving hydroclimate. The modern local meteoric water line (Fig. 4a) reflects a mixture of precipitation derived from summer monsoon southwesterly and winter mid-latitude (Mediterranean) northwesterly sources (Fig. 1b,c). In contrast, the fossil $\delta^{18}O_{FI}$ and $\delta^{2}H_{FI}$ fluid-inclusion waters from speleothems generally plot on the global meteoric water line, synonymous with a southerly moisture source (southern local meteoric water line (S-LMWL); Fig. 4b) and distinct from the modern local meteoric water line (Fig. 4a). This strongly indicates a different precipitation pattern during these

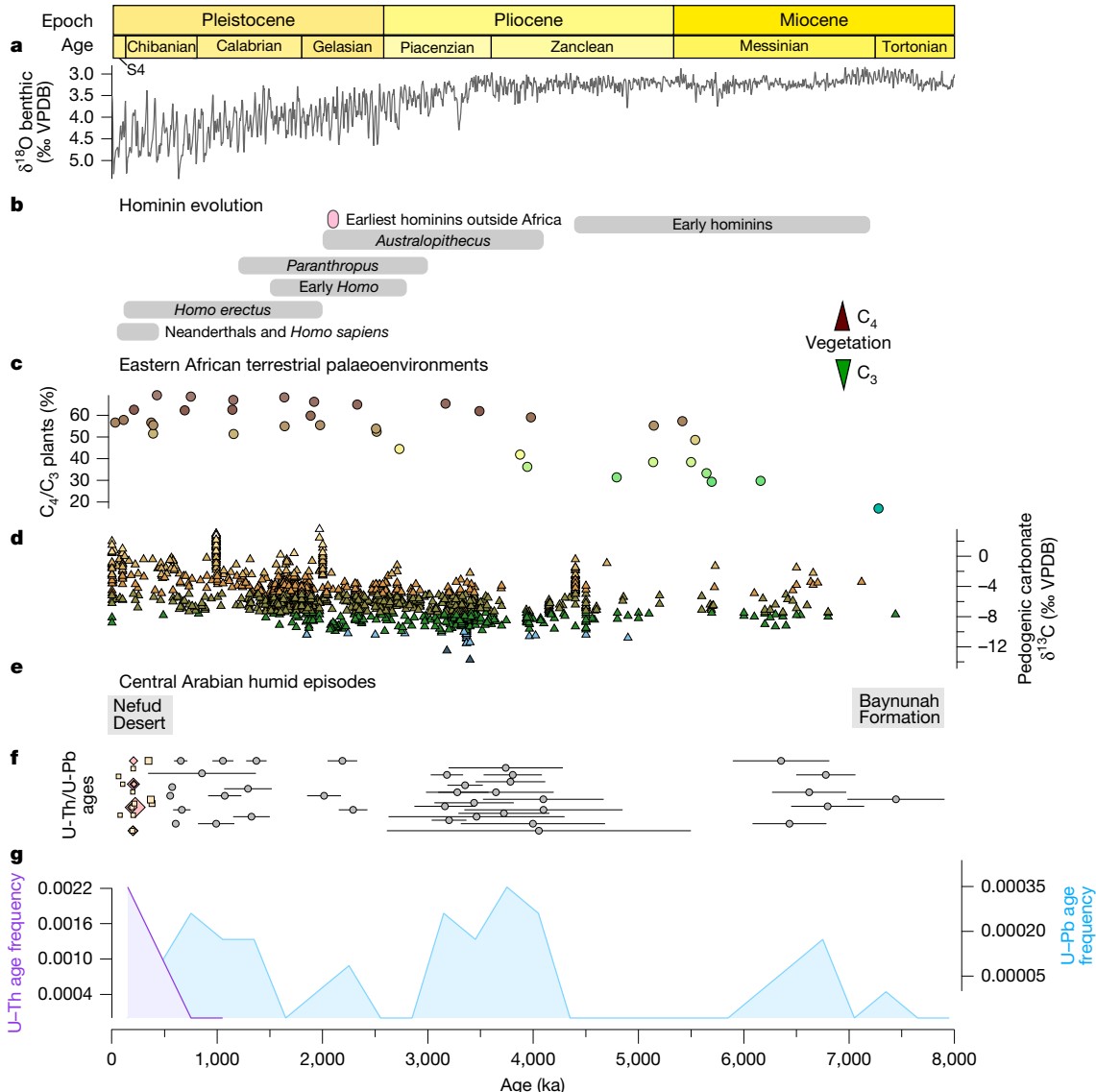

**Fig. 2 | Central Arabian speleothems reveal the episodic occurrence of humid intervals over the past 8 Myr. a**, The δ[18]O (‰ VPDB) benthic stack records covering 8,000 ka (ref. 15) to present relative to epoch and age (S4 refers to Stage Four). **b**, Temporal distribution of key events and hominin information based on appearance dates and their associated assemblage uncertainty (grey bars). **c,d**, Summary of terrestrial environmental change for northeastern Africa including the percentage of C$_4$ plants in eastern Africa from plant wax biomarkers[44] (**c**) and pedogenic carbonate δ[13]C (‰ VPDB) (**d**)[45]. **e**, The timing of the two major fossil deposits in Arabia: the Baynunah Formation[2] in the late Miocene and the Nefud Desert deposits during the middle-to-late Pleistocene associated with stone tools and fauna. **f**, U–Pb-derived ages (grey-filled circles) from this study, and U–Th-derived ages using solution multi-collector inductively-coupled-plasma mass spectrometry (yellow squares) and laser ablation multi-collector inductively-coupled-plasma mass spectrometry (red diamonds) with the 95% confidence interval uncertainty indicated by the horizontal error bars. One age with an uncertainty greater than >±1.5 Myr is not shown but can be found in Supplementary Data 1. **g**, Frequency distribution statistics of both U–Pb-derived and U–Th-derived ages from this study using 300-kyr bins (the average uncertainty of the U–Pb ages). As not all speleothem layers were dated, frequencies should not be used as a record of palaeoclimate.

older humid episodes, relative to the present (Fig. 4). These palaeo-δ[18]O$_{FI}$ and -δ[2]H$_{FI}$ values are instead consistent with reported speleothem fossil water data from southern Arabia dominated by a tropical moisture source during recent insolation-driven peak interglacials[9,24,29] (S-LMWL; Fig. 4b) and are similar to those frequently associated with low-intensity stratiform monsoonal precipitation[30]. Modelling studies have shown that in Arabia, monsoonal winds bring greater moisture transport from the nearby Gulf of Aden and Arabian Sea source regions[31] and are attributed to an increase in the zonal wind component over Africa and the Arabian Peninsula during wetter intervals. Furthermore, the resulting boreal summer precipitation[31] has a lower isotopic composition compared with pre-industrial values, that is in part owing to an amount effect. Fluid-inclusion isotope data indicate a general trend towards

higher δ[18]O$_{FI}$ and δ[2]H$_{FI}$ values through time, consistent with a gradual move towards the present state (Fig. 4a), where annual precipitation is delivered by a combination of both winter and summer moisture sources and associated with very low overall amounts of precipitation owing to its location at the boundary of the two atmospheric delivery systems (Fig. 1b,c). We ascribe this trend to the time-transgressive reduction in the contribution of southerly sourced precipitation to, and monsoonal influence on, the Arabian interior over time during humid episodes.

## Progressive aridification

This progressive reduction in monsoonal precipitation occurs over the past 8 Myr, in tandem with an overall aridity trend resulting from

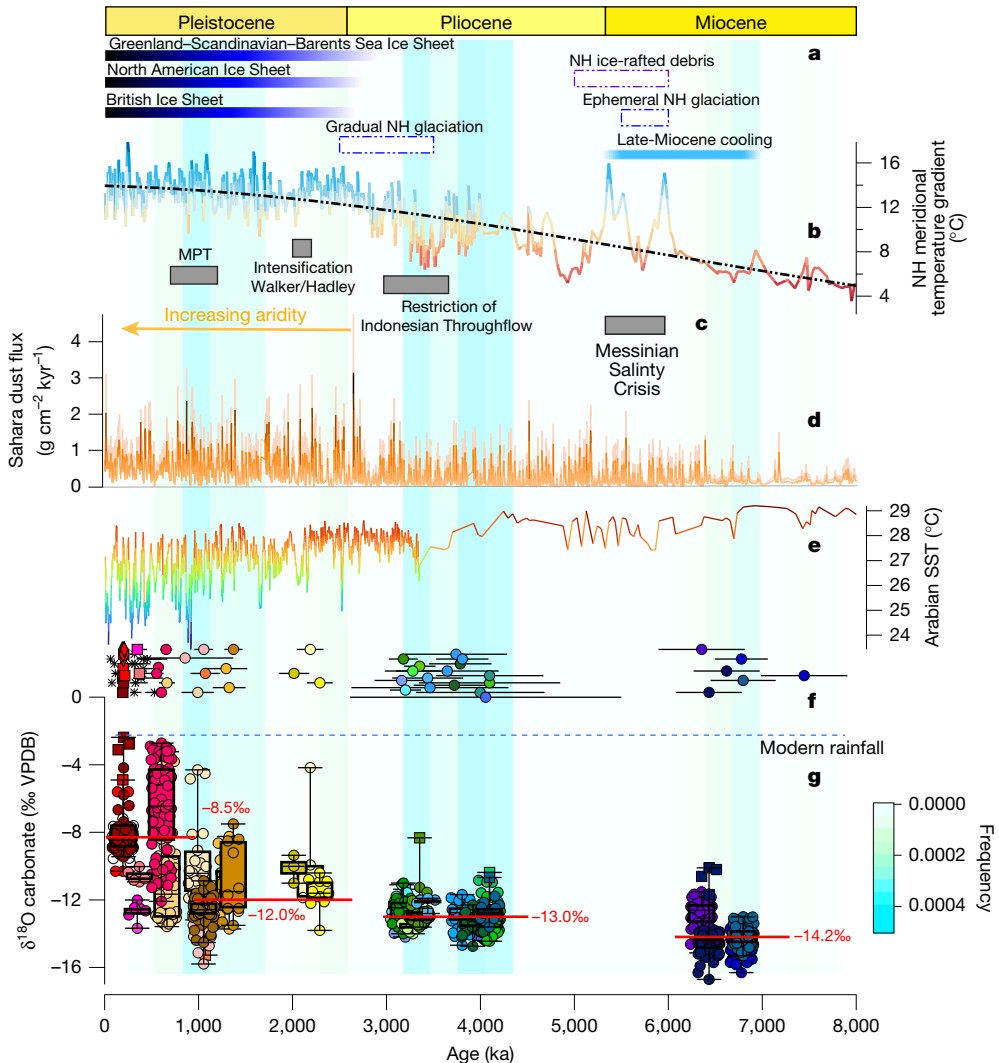

**Fig. 3 | Recurrent central Arabian humid episodes over the late Miocene to late Pleistocene are associated with increasing regional aridity and higher Northern Hemisphere meridional temperature gradients. a**, Key stages of Northern Hemisphere (NH) cooling, including late Miocene cooling[17] associated with ephemeral glaciation[35] and ice-rafted debris[36], as well as the Pliocene/ Pleistocene gradual increase in the Northern Hemisphere in glacial period ice volume[46] and the timing of Northern Hemisphere ice-sheet advancement[47]. **b**, Latitudinal temperature gradient (°C) calculated using the SST differences between Arabian Sea site ODP 722 and North Atlantic site ODP 982[17] (see Methods for calculations). The solid coloured line has shading that varies according to the calculated temperature gradient and the black dashed line is the linear trend line. **c**, Major global events are shown including the Messinian Salinity Crisis[37], restriction of the Central American Seaway and Indonesian Throughflow[48], intensification of Walker–Hadley circulation[16] and the MPT[49]. **d**, Sahara dust flux from North Atlantic marine core 659 indicating high and low inputs, which are indicated by orange to yellow shading, respectively, and the 99% percentile

shown in pink shading[1]. **e**, SSTs from the Arabian Sea based on an alkenone unsaturation method with warmer to cooler SSTs indicated by red to blue shading, respectively[17,33]. **f**, This study; U–Pb-derived ages (filled circles) and U–Th-derived ages using LA-MC-ICP-MS (filled diamonds) and solution MC-ICP-MS (filled squares), with the 95% confidence interval uncertainty indicated by the horizontal error bars. The ages of gypsum 'crust' samples are indicated by asterisk symbols and the colours of the filled symbols refer to the individual speleothem isotopic composition shown in the box plots in **g**. **g**, The average modern rainfall $\delta^{18}O$ (‰ VPDB) value[50] is indicated by the blue dashed line converted to VPDB scale (see Methods for details). Box plots of speleothem $\delta^{18}O$ (‰ VPDB) values ($n = 1,138$), with each colour denoting an individual speleothem growth period. Circles show all individual datapoints, squares indicate outliers and diamonds indicate extreme outliers. Mean values of −14.2 ($1\sigma = 0.9$), −13.0 ($1\sigma = 0.9$), −12.0 ($1\sigma = 1.7$), and −8.5 ($1\sigma = 2.8$) are shown near clustered groups. The background shading shows the U–Pb histogram frequency in Fig. 2g and is indicated by the colour bar.

decreasing water availability over the identified humid phases. Unlike $\delta^{18}O_{FI}$, the oxygen isotope ratios of speleothem calcite ($\delta^{18}O_{carb}$, ‰ Vienna Pee Dee Belemnite (VPDB)) can be influenced by processes not directly related to the isotopic composition of precipitation, such as temperature, non-equilibrium precipitation and so on, and in dryland speleothems lower $\delta^{18}O_{carb}$ typically reflects higher rainfall and ground-water recharge whereas higher $\delta^{18}O_{carb}$ values suggest drier conditions[32]. All the speleothem $\delta^{18}O_{carb}$ values are significantly lower than the theoretical carbonate mean value that would result from modern precipitation (Fig. 3g and Methods). This establishes that past humid episodes

in Arabia experienced higher and more effective precipitation than the present, but show a gradual trend towards higher values through time (Fig. 3g). Mean $\delta^{18}O_{carb}$ values in the Miocene (−14.2 ± 0.9‰) and Pliocene (−13.0 ± 0.9‰) are very low (Fig. 3) and narrowly distributed. An isotopic increase of about +1.3‰ in the Pliocene relative to the Miocene is followed by a +0.8‰ shift towards higher mean $\delta^{18}O_{carb}$ (−12.0 ± 1.7‰) in the early-to-middle Pleistocene. However, the most significant change followed the Mid-Pleistocene Transition (MPT) at 1.1–0.7 Ma (mean $\delta^{18}O_{carb} = −8.5 ± 2.8$‰) together with a significant increase in variability (Fig. 3). The gradual trend through time towards

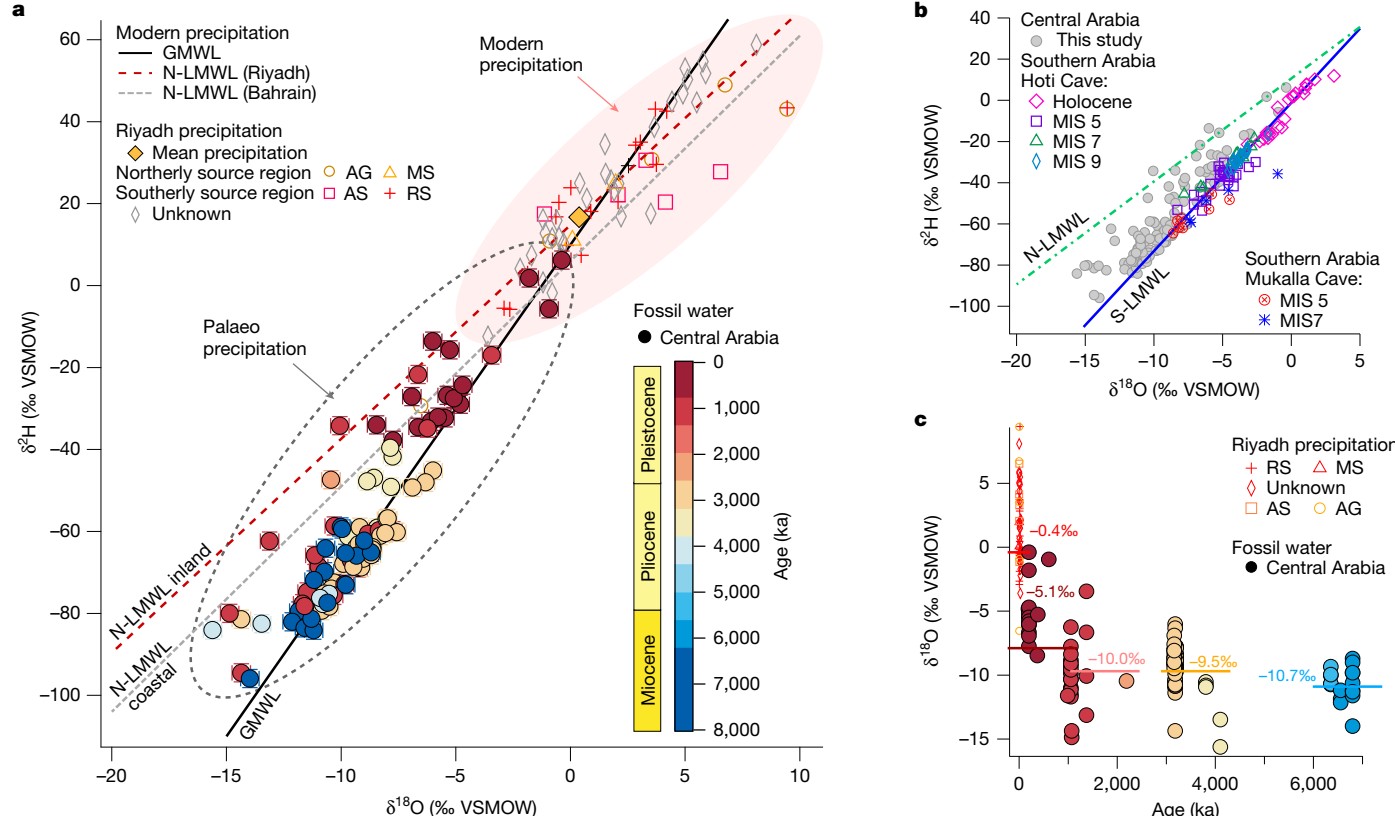

**Fig. 4 | Palaeoisotopic composition of central Arabian speleothem fluid-inclusion waters were associated with a monsoonal southerly source and show lower values relative to present-day rainfall. a**, Late Miocene to Pleistocene isotope composition of fluid-inclusion fossil water within the speleothem calcite lattice ($n = 104$) compared with Riyadh precipitation[50], back-trajectory analysis indicating rainfall sources which originate from the Arabian Gulf (AG), the Mediterranean Sea (MS), the Arabian Sea (AS), the Red Sea (RS), and unknown sources and the weighted mean rainfall for Riyadh (yellow diamond). Meteoric water lines are presented for the global meteoric water line (GMWL), the Riyadh local meteoric water line (LMWL) and the coastal Bahrain LMWL (ref. 50 and references therein). The uncertainty error bars on the fluid-inclusion values are 0.3 (‰ VSMOW) and 1.1 (‰ VSMOW) for $\delta^2H$ and $\delta^{18}O$, respectively, with further information provided in Methods. **b**, Modern LMWLs for northern (N) and southern (S) sources for Arabia[24] compared with fossil water isotope data from speleothems from Hoti Cave, Oman (open symbols)[24], from Mukalla Cave, Yemen (open symbols)[9] and this study (filled symbols). **c**, The central Arabia speleothem $\delta^{18}O_{FI}$ fossil water with respect to time. Mean values of $-10.7$ ($1\sigma = 1.3$), $-9.5$ ($1\sigma = 1.8$), $-10.0$ ($1\sigma = 2.6$), and $-5.1$ ($1\sigma = 2.3$) are shown near the clustered groups.

higher $\delta^{18}O_{carb}$ values (Fig. 3g) is indicative of a gradual decrease in effective moisture availability during humid episodes.

Physical evidence for a temporal trend towards progressively lower humidity over the past 8 Myr is reflected in a change in the speleothem fabrics through time in our record (Methods). Late Miocene speleothems mainly comprise clear, open columnar calcite, indicating adequate recharge from rainfall to maintain a relatively stable karst aquifer above the cave providing constant discharge and regular drip rates. In contrast, post-MPT speleothems mainly comprise dendritic, colourful/opaque fabric, often with more frequent hiatuses and micro-hiatuses, indicative of an intermittent water supply to the vadose karst reservoirs, leading to non-equilibrium isotopic fractionation processes[32] during more 'pulsed' humid phases. The synchronous shifts in $\delta^{18}O_{carb}$ and speleothem fabrics suggest reduced effective groundwater recharge towards the late Pleistocene, and we suggest that this is owing to a gradual reduction in monsoonal precipitation in the Arabian interior.

A steady contraction of the polewards (northwards) extent of the tropical rainbelt during humid episodes over the past 8 Myr is consistent with increases in the meridional sea surface temperature (SST) gradient between the Arabian Sea and the North Atlantic Ocean (Fig. 3b and see Methods for expanded text). Higher meridional SST gradients are associated with the Pleistocene and late Miocene global cooling intervals and are strongly coupled with increased Northern Hemisphere polar ice extent (Fig. 3) and Hadley Circulation contraction, bringing large-scale

subsidence and prolonged dry conditions over Arabia. Northern Hemisphere cooling is reflected in our record by the gradual shift to higher $\delta^{18}O$ values, both in speleothem calcite and fluid-inclusion water, which provides evidence for a gradual shift towards precipitation originating from mid-latitude westerlies, and reduced effective precipitation during interglacials, particularly after the MPT, owing to the reduced influence of the summer monsoon. This change in the position of the ITCZ and changes in atmospheric circulation patterns in the Saharo-Arabian Desert are also evidenced by increased dust transport from the Sahara Desert from 2.3 Ma (ref. 1; Fig. 3d) and greater terrigenous input into the Arabian Sea[33]. Enhanced dust transport has been attributed to the Pleistocene intensification of glacials, associated with increased aridity, less vegetation, lower soil moisture, and changes in atmospheric wind direction and intensity, coupled with lower precipitation[34]. The production of fine-grained sediments from source areas such as Lake Mega-Chad during interglacials is followed by desiccation, deflation and their transport in glacials, acting as dust-producing 'hotspots'[1]. Importantly, however, as our speleothems are selective records of interglacial humid intervals (opposed to glacial aridity), the drying trend observed in the speleothem record may occur in tandem with higher dust transport in glacial phases. Furthermore, during the globally warmer and higher-atmospheric-$CO_2$ world of the Pliocene and the late Miocene, the wetter humid episodes in central Arabia were probably associated with lower wind speeds, more vegetation and higher recharge.

Although most, if not all, gaps in our record are too small to exclude sampling bias as the cause for their appearance, the largest gap in our dataset, from about 6.3 Ma to about 4.1 Ma is contemporaneous with both regional and global evidence for increased Northern Hemisphere low-latitude aridity. Late Miocene cooling (6–5 Ma), when global temperatures and meridional temperature gradients temporarily shifted to approximately modern-day values[17] in conjunction with Northern Hemisphere ephemeral glaciation (6.0 to 5.5 Ma)[35] and evidence of ice-rafted debris in the North Atlantic[36] (Fig. 3a), probably led to increased aridity on the Arabian Peninsula. Regional evidence for terminal Miocene aridification is provided by palaeoclimate data from the Mediterranean and Red seas during the Messinian Salinity Crisis, which suggest significant desiccation (6.0 Ma to 5.3 Ma (refs. 37,38)) likely driving the regional aridity through reduced moisture availability from changes in regional land–sea configurations. Furthermore, modelling of the Mediterranean–Atlantic exchange has revealed a significant influence on the North Atlantic, where the Messinian Salinity Crisis may have induced mid–high-latitude cooling by a few degrees Celsius and a decline in the Atlantic Meridional Overturning Circulation[39]. Higher induced meridional SST gradients (Fig. 3) provide a mechanism for a temporary equatorial excursion of the ITCZ away from the Arabian Peninsula during the Messinian Salinity Crisis in the terminal Miocene.

## Biogeographical significance

Our speleothem-based hydroclimatic record demonstrates recurrent humid phases in central Arabia, likely facilitating dispersals and biogeographical exchange at this crossroads between Africa and Eurasia (see Methods for extended discussion). The importance of climatically induced dispersals through Arabia is underpinned by late Miocene palaeontological evidence, notably the extensive fossil fauna recovered from the Baynunah Formation (about 7.7–7.0 Ma)[40], which indicate that terrestrial Arabia was suitable for hosting a highly diverse array of large mammals. The Baynunah Formation contains abundant fossils of water-dependent fauna that are currently absent in Arabia, including crocodiles, hippopotamids, proboscideans, bovids, giraffids, equids and numerous carnivores of primarily African origin but also Indomalayan bovids (for example, *Pachyportax latidens*) and suids (for example, *Propotamochoerus hysudricus*)[41] that have never been recorded in Africa[4]. The rich diversity of bovid and giraffid fossils that exhibit mixtures of Afrotropical, Indomalayan (Siwalik) and minor Palaearctic affinities, and the broad absence of endemic species, substantiates Arabia as a dispersal corridor or mixing cul-de-sac[42] between Eurasia and Africa[2]. It also suggests a level of selective filtering between neighbouring biogeographic regions[41], driven by the frequency of humid episodes in the Saharo-Arabian Desert. Notably, there are currently no known vertebrate fossil deposits in Arabia dated between the late Miocene and the middle Pleistocene. Previous work has argued for a sustained arid barrier in northern Arabia (33° N) between 5.6 Ma and 3.3 Ma, which hindered animal dispersals and consequently promoted endemism of African mammals into their present-day clades[13]. Our data, which highlight in situ hydroclimatic sensitivity, reveal that this aridity did not extend to the entire Arabian Peninsula and that the Pliocene experienced episodes of increased precipitation sufficient to support a range of taxa not present in the region today.

Fossil evidence of interconnectivity of mammalian assemblages over the late-Neogene period shows that longitudinal dispersals across the 'Old World savannah palaeobiome' were favoured in the late Miocene, but became increasingly latitudinally fragmented from the Pliocene onwards, with an overall trend towards reduced faunal exchange[11]. This eventually led to the reduction of the Old World savannah palaeobiome and the development of Plio–Pleistocene African savannah fauna, including the presence of early *Homo* and *Australopithecus*[11]. Our data suggest that over this same period, Arabia became progressively more arid through time (Fig. 3g), triggered by a reduced contribution of monsoonal precipitation (Fig. 4). However, despite drier humid intervals in the Pleistocene relative to the Pliocene and the Miocene, the presence in Arabia of both African and Asian mammals, particularly those with large water requirements, during humid periods[3] in the middle-to-late Pleistocene indicates that dispersals were still possible. Our record constrains both the timing and frequency of such climatic amelioration phases, providing critical environmental context for past mammalian dispersals into and across the Arabian Peninsula.

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

## Methods

### Speleothem samples

The caves are in the As Sulb Plateau within the Paleogene Umm Er Radhuma Formation, which is a light-grey-to-white calcarenite limestone and dolostone. In 2019, speleothems were sampled from five cave systems previously identified by the Saudi Geological Survey: Kahf at Tahaaleb/كهف الطحالب (Mossy Cave), Kahf al Fondoq/كهف الفندق (Hotel Cave), Kahf al alf Janah/كهف الألف جناح (1000 Wings Cave), Kahf Almorabbaa/كهف المرب (Murubbeh Cave) and Kahf Assadaqah/كهف الصداقة, and two other caves that we believe are previously unreported: EP19.8 and Kahf Alfakhamah/كهف الفخامة (Luxury Cave), which are all within a 10-km radius at about 450 m above sea level (Fig. 1d). The surface above the cave systems mostly consists of exposed bedrock but in places is covered by dune sands with areas of evaporite deposits (Fig. 1d). Details of individual specimens are shown in Extended Data Figs. 1–3. The general cave locations have been provided (Fig. 1); further detail is not provided herein for conservation purposes. Precise locational information is available to professional organizations and scholars from the Saudi Heritage Commission and the Saudi Geological Survey.

Preservation bias due to 'natural attrition' processes may result in less speleothem preservation further back in time. This is owing to processes such as downwards karst erosion, cave collapse, in-cave erosional processes, in-cave sedimentation and speleothem precipitation covering older material. A previous study[51] that focused on a tropical site in Sulawesi, Indonesia, suggested that stalagmites follow an exponential relationship of decreasing numbers through time. Although this is likely very relevant for tropical settings, which are not periodically water limited, the opposite trend is observed in central Arabia, whereby more speleothem material by volume has been deposited in the late Miocene and the Pliocene than recently. This was also reflected in an earlier sampling campaign[52] where most of the samples found were beyond the U–Th dating technique (more than 500 ka). We suggest that dryland environments do not experience high rainfall erosivity like the tropics; thus, they may be more suitable for speleothem archive preservation in deep time. For example, the Nullarbor speleothem record in arid southern Australia[22] shows a similar trend to our data, as most speleothems in those cave localities formed in the Miocene and the Pliocene, with little formation in the Quaternary period. Furthermore, a similar story is emerging from speleothems collected from caves in the Negev Desert[53]. Therefore, the fact that a large number of speleothems are preserved in these deeper time intervals suggests that potential gaps in our data (that is, late Miocene) are unlikely to be a result of natural attrition.

### U–Th dating

A total of 68 U–Th measurements (46 via solution and 22 via laser ablation) were determined to produce 35 ages for 22 speleothems (Extended Data Figs. 1–3) and measured at the Institute for Geosciences, Johannes Gutenberg University, Mainz (Fig. 2, yellow squares). A total of 46 subsamples of 100–300 mg were cut using a micro-bandsaw from the central growth axis of the speleothems. The weighed samples were dissolved in 7 N HNO$_3$ and a mixed $^{229}$Th–$^{233}$U–$^{236}$U spike was added (see ref. 54 for details on spike calibration). Potential organic material was removed by adding a mixture of concentrated HNO$_3$, HCl and H$_2$O$_2$. The dried samples were dissolved in 6 N HCl, and U and Th were separated using ion-exchange columns[55]. Samples were analysed using multi-collector inductively-coupled-plasma mass spectrometry (MC-ICPMS) (Neptune Plus) and technical details about the MC-ICPMS procedures are described in ref. 56. All activity ratios were calculated using the decay constants from ref. 57 and corrected for detrital contamination assuming a $^{232}$Th/$^{238}$U weight ratio of 3.8 ± 1.9 (50%) for the detritus, which is propagated into the final age uncertainty, and that $^{230}$Th, $^{234}$U and $^{238}$U are in secular equilibrium. All the corrected ages agree within error with the uncorrected ages.

In addition to solution-derived ages, 22 $^{230}$Th/U ages were determined by in situ laser ablation multi-collector inductively-coupled-plasma mass spectrometry (LA-MC-ICPMS) at the Institute for Geosciences, Johannes Gutenberg University, Mainz (Fig. 2, red diamonds, Extended Data Figs. 1–3 and Supplementary Information). A Neptune Plus (Thermo Scientific) MC-ICP-MS was coupled to an ArF Excimer 193-nm laser ablation system (ESI NWR193) equipped with a TwoVol ablation cell. A CETAC Aridus 3 desolvating system was coupled to the sample line to introduce nitrogen and argon to increase the sensitivity. Each age was calculated using data from two separate 1,000-μm line scans with $^{230}$Th or $^{234}$U collected on the central secondary electron multiplier (see Supplementary Data 1 for cup configuration and instrumental settings) after a short pre-ablation to clean the surface. Samples were analysed using a standard-sample-bracketing approach, using a subsample from flowstone WM1 from Wilder Mann Cave, Austria, as reference material, which yielded a U–Pb isochron age of more than 2.0 Ma (ref. 58). Therefore, the WM1 sample is in secular equilibrium and ($^{234}$U/$^{238}$U) and ($^{230}$Th/$^{238}$U) are equal to one. A global correction factor including instrumental mass fractionation and ion counter yield was applied for each line to correct the activity ratios of the samples using the decay constants from ref. 57. Detrital contamination was corrected for as described above for the solution-based $^{230}$Th/U dating.

### U–Pb dating

A total of 39 radiometric ages were calculated from 22 speleothems wafers, which were mounted in 25-mm resin mounts and manually polished using silicon carbide abrasive papers and 18.2-MΩ-cm deionized water to limit metal contamination and produce a smooth surface for laser ablation analyses. Mounts were cleaned before analyses in an ultrasonic bath, submerged in ethanol for 10 min. In situ laser ablation inductively-coupled-plasma mass spectrometry at FIERCE, Goethe University Frankfurt was used to measure U, Th and Pb isotopes using previously described procedures[59–61]. Isotopic measurements were conducted using a sector-field inductively-coupled-plasma mass spectrometer (Thermo Scientific Element XR) coupled to a RESOlution 193-nm ArF excimer laser (CompexPro 102), equipped with a 2-volume ablation cell (Laurin Technic, S155). Samples were ablated in a helium atmosphere (300 ml min$^{-1}$) and mixed in the ablation funnel with argon (1,100 ml min$^{-1}$) and nitrogen (5 ml min$^{-1}$). The inductively-coupled-plasma mass spectrometer was tuned for a balance between maximizing signal sensitivity and minimizing (UO/U) oxide formation. An ablation square spot size of 213 μm was used. Samples were ablated using a fluence of about 2 J cm$^{-2}$ at 12 Hz.

Each analysis consisted of 21 s of background measurement, followed by 22 s of ablation. Before the measurement, each spot was pre-ablated by ten laser pulses, using the same parameters as the main ablation, to remove any surface contamination. During the analysis, isotopic data for 5 masses were acquired: 206, 207, 208, 232 and 238. Analyses were conducted overnight in a fully automated mode. The raw data were corrected offline[62,63]. NIST 614 was used as a primary reference material for inter- and intra-element mass biases, instrumental drift corrections and concentration determinations. Offsets between NIST glass and calcite were corrected with WC-1 reference material[64]. Validation reference materials were ASH–15–D[65,66], BraT (in-house reference material) and B6[67].

The initial $^{234}$U/$^{238}$U (($^{234}$U/$^{238}$U)$_{initial}$) assumption and related correction in the U–Pb age calculation can have a significant bearing on the resultant calculated ages. Measured activity ratios and calculated ($^{234}$U/$^{238}$U)$_{initial}$ for the data in this study can be found in Supplementary Information. The ($^{234}$U/$^{238}$U)$_{initial}$ was estimated from the average measured ($^{234}$U/$^{238}$U) on younger (less than 1.5 Ma) U–Th- and U–Pb-dated samples to calculate the ($^{234}$U/$^{238}$U)$_{initial}$, based on the isotope decay constants yielding an average value of 1.013 ± 0.178 (2$\sigma$). The U–Pb ages of samples younger than 1.5 Ma were calculated using speleothem-specific ($^{234}$U/$^{238}$U)$_{initial}$ values obtained from solution U–Th

measurements and samples older than 1.5 Ma were corrected using an average $(^{234}U/^{238}U)_{initial}$. Disequilibrium ages were calculated using DQPB software[68] and uncertainties are given in Supplementary Information.

To further assess how representative the $(^{234}U/^{238}U)_{initial}$ central Arabia speleothem values are compared with the general regional $(^{234}U/^{238}U)_{initial}$ data, we collated all the speleothem data for Arabia ($n = 1,300$), as well as modern groundwater activity ratios (Extended Data Fig. 10b,c). Central Arabian mean $(^{234}U/^{238}U)_{initial}$ values are representative of averages for the region, with the largest proportion of $(^{234}U/^{238}U)_{initial}$ values falling between 1.0 and 1.1. Furthermore, the modern activity ratios of the carbonate aquifers proximal to our cave site show similarly low $(^{234}U/^{238}U)$.

### Stable isotope analyses

Samples were analysed on a Thermo Delta V mass spectrometer equipped with a GASBENCH-II preparation device at the Max Planck Institute for Chemistry, Mainz. Approximately 20–50 µg of $CaCO_3$ sample was placed in a helium (He)-filled 12-ml exetainer vial and digested in water-free $H_3PO_4$ at a temperature of 70 °C. Subsequently, the $CO_2$–He gas mixture was transported to the GASBENCH in He carrier gas. In the GASBENCH, water vapour and various gaseous compounds are separated from the $CO_2$–He mixture before sending it to the mass spectrometer. Isotope values are reported as $\delta^{13}C$ and $\delta^{18}O$, relative to VPDB. A total of 20 replicates of 2 in-house $CaCO_3$ standards were analysed in each run of 55 samples. $CaCO_3$ standard weights were chosen so that they span the entire range of sample weights of the samples. After correction of isotope effects related to sample size, the reproducibility of these standards typically is better than 0.1‰ VPDB (1 s.d.) for $\delta^{18}O$ and for $\delta^{13}C$. $\delta^{18}O_{carb}$ is reported relative to the VPDB standard and is given in Supplementary Data 1. The calcite $\delta^{18}O$ equivalent of modern mean rainfall $\delta^{18}O$ in Fig. 3 was calculated following ref. 69 and using the modern mean annual temperature of 25 °C.

### Fluid-inclusion isotope analyses

Cave dripwater is preserved within numerous microscopic inclusions in the speleothem carbonate (0.1 wt% on average). The isotope analysis of fluid-inclusion water allows us to identify the $\delta^{18}O$ and $\delta^2H$ values of the fossil dripwater directly, with respect to global meteoric water lines, circumventing the complications of other secondary fractionation processes inherent with the $\delta^{18}O_{carb}$ signal. Speleothems were sampled for fluid-inclusion isotope analysis using a micro-bandsaw device, equipped with a 0.3-mm-wide diamond-covered circular blade. Depending on the water content of the calcite, small calcite blocks between 0.1 g and 1.5 g were cut. The calcite was crushed by a downwards rotary movement in a percussion device (Potsdam-type crusher) at a temperature of 100 °C. Subsequent isotope analysis of the water vapour released from crushing was performed on a Picarro L2140i cavity ring-down spectrometer following an established protocol[70] at the Max Planck Institute for Chemistry, Mainz. The reported uncertainties are based on the previously established reproducibility for this instrument[70] and confirmed by repeated analysis of standard water injection in the crusher system during series of crushes of speleothem samples. The typical $1\sigma$ reproducibility is 0.3‰ VSMOW for $\delta^{18}O$ values and 1.1‰ VSMOW for $\delta^2H$ values[71] for this instrument and is reported in Supplementary Information.

### Statistical tests

To determine whether the isotopic composition of the carbonate and fluid inclusions statistically differed in $\delta^{18}O$ mean and variance, one-way analysis of variance and post hoc Tukey tests were performed using the 'dplyr' package in the R software. Isotope values were split into four groups based on geological time epochs: late Miocene, Pliocene, early-to-middle Pleistocene and late Pleistocene. Early-to-middle Pleistocene (2.56–1.00 Ma) and late Pleistocene (1.00–0.00 Ma) were defined by the start of the MPT at about 1 Ma. Of the 4 epochs

investigated, all showed statistically significant differences in the $\delta^{18}O_{carb}$ values ($P < 0.05$; Extended Data Table 1). The fluid-inclusion isotope data showed that the late Miocene, Pliocene and early-to-middle Pleistocene were all statistically different to the late Pleistocene, which had significantly higher $\delta^{18}O$ and $\delta^2H$ values. In addition, the late Miocene was also significantly different to the early-to-middle Pleistocene.

### Raman spectroscopy

Speleothems were petrographically screened for evidence of diagenesis and changes in calcium-carbonate mineralogy by a combination of optical microscopy and Raman microspectroscopy. Raman analyses were conducted on thick and thin sections using a confocal Bruker Senterra micro-Raman spectrometer equipped with an Olympus BX 51 microscope with an Olympus LWD objective and an Andor DU420-OE charge-coupled-device camera. Measured spectra were collected at room temperature using a 532-nm-wavelength laser (green) with 20 mW of power, under ×50 magnification, an acquisition time of 1 s and 10 acquisition repetitions. Accuracy was determined using a silicon standard to be within ±2 cm$^{-1}$. The focused sampling spot was about 2 µm in diameter. The wavenumbers of Raman shifts for common carbonate minerals (for example, calcite, aragonite and dolomite) are unique and, thus, diagnostic of speleothem mineralogy[72,73]. This revealed that aliquot powders from all measured speleothems comprised only pure calcite (Extended Data Fig. 5).

Further mineralogical investigations were conducted to assess mineralogy of the 'crust' layers on speleothems SA07 and SA10. The presence of gypsum crusts was identified with the most intense peak found at 1,008 cm$^{-1}$, corresponding to the $v_1$ symmetric-stretch vibration mode of $SO_4$ (ref. 8; Extended Data Fig. 5). The minor gypsum peaks associated with the $SO_4$ tetrahedra were also found at 415 cm$^{-1}$ ($v_2$), 495 cm$^{-1}$ ($v_2$), 1,138 cm$^{-1}$ ($v_3$) and 672 cm$^{-1}$ ($v_4$)[70]. The characteristic bands for OH stretching vibration modes for water in gypsum (3,200 cm$^{-1}$ to 3,500 cm$^{-1}$) were not found as we measured only the spectral range from 50 cm$^{-1}$ to 1,550 cm$^{-1}$.

### Petrographic screening of speleothem material

Speleothem growth mechanisms and the resulting fabric and morphology are related to the hydrology and dripwater properties at the time of formation[74]. Open columnar fabric is characterized by calcite crystals marked by the presence of linear inclusion layers or pore spaces (for example, Extended Data Fig. 4a). In general, columnar-type fabrics form under constant drip regimes with low colloidal and particulate organic matter in dripwaters[74]. In our material, this fabric is typical of speleothems (Extended Data Fig. 4a,b) growing in the Miocene and the early Pliocene, and suggests a relatively constant drip flow, and by extension karst reservoir, where the fastest drip rates are typically associated with open columnar fabrics (for example, Extended Data Fig. 4a). Pliocene speleothem fabric changed from more open columnar at 3.806 Ma, 4.098 Ma and 4.054 Ma (stable karst reservoir) to more opaque/coloured speleothems at 3.353 Ma, 3.785 Ma and 3.438 Ma and then to dendritic fabrics at 3.200 Ma. Dendritic fabric forms in environments with variable drip rates and is often associated with bio-influenced carbonate precipitation (that is, from higher organic matter content in dripwaters) and places where speleothem surfaces dry out owing to inadequate dripwater supply[74,75], which contributes to the 'scaffold type' structure[74]. This range of fabrics may suggest some environmental instability in the Pliocene or the beginning of a shift towards more arid conditions.

The largest shift in fabric followed the MPT, after which colourful/opaque speleothem predominate, often with more frequent hiatuses and micro-hiatuses, indicative of the increased intermittent supply of karst water in reservoirs above the cave. SA40 (a flowstone) shows evidence of microsparite and mosaic calcite structures, suggesting the presence of organic compounds and microbes and or diagenetic alteration. Furthermore, speleothems such as SA31 and SA30 (Extended

Data Fig. 1) are darker (yellow/brown) in colour, suggesting potential for microbial communities to grow on speleothem surface layers, a clear indication of hydrologic instability[74], or the presence of higher amounts of dissolved or particulate organic matter in speleothem-forming groundwaters above the karst aquifer. Total organic matter concentrations in dark speleothems have been found to be twice that of lighter speleothems[76] and global groundwater data suggest that significantly higher dissolved or particulate organic matter is associated with aquifers with low annual recharge (less than 100 mm yr$^{-1}$)[77]. Furthermore, microsparite formation is specially favoured after precipitation resumes following a dry phase, again potentially indicating low flow and intermittent growth favouring dissolution and reprecipitation of calcite phases.

### Potential effect of sea-ice volume on precipitation δ[18]O

A correction for the effect of ice volume on the δ[18]O of precipitation was considered, to determine the potential extent that changes in ice volume were responsible for the shift in δ[18]O observed in our speleothem record over the past 8 Ma. We calculated the impact for our most extreme endmembers of sea-level change, the Miocene (low global ice volume) and the present day (high global ice volume). The late Miocene was characterized by a global sea level that was 22 m higher than at present[78]. Following an ice-volume correction of 0.008‰ m$^{-1}$ of sea-level change[79], the maximum isotopic difference evoked by sea-ice-volume changes to seawater δ[18]O composition is −0.176‰. This is insufficient to explain the changes in speleothem δ[18]O over the late Miocene to the late Pleistocene. The offset would be larger if speleothems grew over glacial phases (as more water is locked away as ice); however, the ages of speleothems from central and southern Arabia[9] suggest that most deposition mainly occurred through warmer interglacial periods. Nevertheless, if we calculate the theoretical δ[18]O precipitation offset for the sea level during the Last Glacial Maximum, which was approximately 125 m lower relative to present[80], this would have an impact on seawater δ[18]O of approximately +1‰. Again, the shift in δ[18]O in our speleothems far exceeds this and therefore changes in ice volume cannot account for the total observed shift in speleothem δ[18]O observed in our data.

### Meridional temperature gradient shifts over the past 8 Myr

Humidity in subtropical deserts is strongly linked to the position of the descending arm of the Hadley Cell, bringing semi-permanent high-pressure systems and maintaining dry conditions. The strength of the Hadley Circulation and the position of its descending subtropical branch is related to the meridional temperature gradients (MTGs) between the polar and the equatorial SSTs[81,82]. During glacial periods, when the Northern Hemisphere has more expansive polar ice coverage, the pole-to-equator MTG is typically greater[83]. The reverse is true when polar ice is reduced, both of which have implications for the positioning and strength of the Hadley Circulation.

To visualize the extent of meridional temperature variability and the potential shift of the Hadley Cell, we calculate the MTGs over the past 8 Myr from published SST records from the Arabian Sea compared with high-latitude sites (Extended Data Fig. 6). We consider only records based on the alkenone palaeothermometer (Uk′37) because they offer the best spatial and temporal coverage for the period of interest[17]. However, it should be acknowledged that there is attenuation of the Uk′37 response to temperature as the ratio approaches one, and the Uk′37 alkenone proxy becomes saturated when SSTs are above 28 °C, potentially leading to underestimates of the MTGs before the late Miocene cooling[17]. We first used a linear interpolation to convert the raw data to an evenly spaced time series at 5-kyr resolution. We then subtracted the polar from the tropical SSTs to produce an MTG between the polar and tropical oceans, where higher values suggest greater MTGs. Temperature gradients were calculated between the Arabian Sea site (Oceanic Drilling Program (ODP) site 722) and the North Pacific (ODP 883/884/887), Norwegian Sea (ODP 907) and the North Atlantic

(ODP 982). In addition, South Atlantic sites ODP 1088 and Deep Sea Drilling Project (DSDP) 594 were also compared (Extended Data Fig. 7). The overall trend towards higher MTGs from the late Miocene to the present reflects the greater differences in SSTs between the Northern Hemisphere polar regions and the tropical oceans. The δ[18]O composition of our speleothems reflects a similar trend towards higher values over the late Miocene to the Pleistocene, which suggests a sensitivity of Arabian hydroclimate to Northern Hemisphere ice cover, and cooler high-latitude SSTs.

MTG minima occur in both Northern and Southern hemisphere oceans at approximately 7.5 Ma. During the late Miocene cooling, MTGs diverge significantly, particularly in the North Atlantic (Extended Data Fig. 7a) between 6 Ma and 5.5 Ma. SSTs in polar regions increased in the early Pliocene relative to the late Miocene-cooling period, resulting in reduced MTGs. Towards the end of the Pliocene, MTGs increase once again, driven by a greater cooling of polar oceans (Extended Data Fig. 7). However, tropical areas, such as the Arabian Sea, show a smaller overall change in SST's compared with high-latitude areas (Extended Data Fig. 7); thus, the MTG is mainly controlled by changes in polar oceans.

During the Pleistocene, glacial–interglacial cycles intensified, allowing further exploration of the sensitivity of Arabian hydroclimate to Northern Hemisphere glacial boundary conditions. Although the dating resolution often does not permit individual cycles to be resolved, we can examine the overall strength of interglacial phases within a given period and the frequency of humid phases (Extended Data Fig. 8). To do this, we compare the MTGs over the Pleistocene where the data are available (at sites ODP 722 and ODP 982) as well as the glacial index, which is indicative of the Northern Hemisphere ice-volume extent[84]. Furthermore, the glacial index identifies periods based on the lack of Northern Hemisphere land ice outside of Greenland, to account for skipped terminations relating to obliquity cycles without an interglacial. This shows that terminations responsible for creating new interglacial phases in the Pleistocene are more irregularly spaced than the patterns seen in the LR04 benthic record[85], and not necessarily forced by obliquity changes. The Pleistocene moves from an interglacially dominated (2.6–2.1 Ma and 1.7–1.1 Ma) period to a glacially dominated period in the late Pleistocene (Extended Data Fig. 8). In general, our speleothem ages align with periods in the Pleistocene that are 'interglacially dominated' (Extended Data Fig. 8, indicated by red panel background) and appear associated with a lower Northern Hemisphere ice extent outside of Greenland. Although there are only two U–Pb ages published so far for southern Arabian speleothems, the data suggest that their growth coincided with significant interglacial phases; super interglacial 31 and interglacial 25[9]. After the MPT, where glacial–interglacial cycles moved to a 100-kyr frequency, there were warmer successive interglacials (that is, 11, 9, 7 and 5) even though generally the past 1 Myr is a period of greater Northern Hemisphere ice extent. Further, about 0.2 Ma marks the start of an uninterrupted phase of high global ice volume until present, coinciding with the absence of speleothem growth, and only gypsum deposition in central Arabia from this point forwards. Together, this suggests that the central Arabian hydroclimate is sensitive to longer-term Earth system feedbacks, and Northern Hemisphere glacial boundary conditions, as demonstrated in southern Arabia[86]. Furthermore, Pliocene modelling simulations of the Sahel and East Asia also suggest that the mid Pliocene wetter hydroclimate state is driven by a reduction in Northern Hemisphere high-latitude ice extent and continental greening[87].

### The vertebrate fossil record and palaeobiogeography of Arabia

The Neogene and Quaternary vertebrate fossil record of Arabia is scant, with fossils deriving from two principal areas: the Baynunah Formation in the United Arab Emirates and the Nefud Desert in northwestern Saudi Arabia. Combined magnetostratigraphy and biostratigraphy place the Baynunah Formation within the late Miocene, between about 7.7 Ma and 7.0 Ma (ref. 40). It represents the only late Miocene vertebrate

fossil deposit for the entire Arabian Peninsula. The faunal assemblage is diverse, comprising various species of fish, birds, amphibians, reptiles, proboscideans, carnivores, rodents, primates, bovids, equids and hippos[88]. The majority of fossils show strong biogeographical connections to Africa, although some Eurasian elements are also present[2,89]. Overall, the fossil fauna, stable isotope and sedimentological data indicate a less arid environment than the present-day region, which is characterised by highly seasonal conditions and composed of open $C_4$ grasslands, woodlands and rivers[2,7]. Research suggests that this savannah environment formed part of the widespread Old World savannah palaeobiome, which is thought to have extended across large parts of Africa and Eurasia during the global cooling and aridification of the middle and late Miocene[2,11]. This period saw significant faunal interchange between Africa and Eurasia up until around 6 Ma when the Eurasian Pikerminan and Baodean chronofaunas broke down[11,90]. When present, the Saharo-Arabian Desert acted as an arid barrier to dispersal, restricting the movement of fauna between Africa and Eurasia and promoting endemism[13,91,92]. Indeed, a previous study[11] suggested that the modern African savannah fauna evolved through increasing endemism in the Pliocene and the Pleistocene.

There is a substantial, approximately 6.5-Myr gap, between the Baynunah Formation fossils and those of the Nefud Desert, the latter dating to the middle-to-late Pleistocene, or roughly the past 500 kyr (refs. 3,12). The fossil assemblages of the Nefud Desert are primarily associated with interglacial palaeolake deposits of varying ages, although there is also evidence of some lake and river formation during glacial phases[12,93,94]. Fossil fauna, stable isotope and sedimentological data suggest that interglacial phases in northern Arabia were characterized by open $C_4$ grassland environments with large, permanent freshwater lakes[95–97]. Similar to the Baynunah Formation, the Nefud Desert fauna show strong affinities with Africa, although some Eurasian elements are certainly present[92]. Climate modelling[28] and speleothem isotopic data[9,29] strongly suggest that increased humidity and precipitation was the result of the polewards expansion of the ITCZ, which brought increased precipitation to the Sahara and Arabia, effectively transforming these deserts into open grasslands and permitting the spread of $C_4$ grassland-adapted taxa, including hominins, across them. In the intervening glacial periods, the hyperarid deserts returned and most fauna, barring those adapted to hyperarid conditions, would have retreated or become locally extinct[92]. Simply put, the Pleistocene of Arabia consisted of recurrent glacial–interglacial cycles that transformed the deserts into savannahs and grasslands, facilitating the repeated dispersals of fauna and hominins. A sparse late Pleistocene fossil record exists in the southern part of the peninsula, such as at Shi'bat Dihya in Yemen[93] and in the Empty Quarter in southern Arabia[98], and these sites paint a similar picture to those farther north.

Despite the large temporal gap between the Baynunah Formation and the Pleistocene fossil assemblages, fossils from the intervening period from Africa and Eurasia provide insights into intercontinental faunal movements. Taxa with African affinities that appear in Asia, and vice versa, are oftentimes considered to reflect dispersals between these two regions following periods of climatic and environmental change[13,99–104]. Vrba[99,100], for instance, interpreted the patchy Caprini fossil record in Africa to represent repeated episodes of faunal immigration from Eurasia during periods of global cooling and opening of land-bridge connections. Vrba's 'traffic light' hypothesis established four or more potential independent dispersal events[90,105], although conflicting studies cautioned that the various species of Caprini might represent in situ speciation and not necessarily repeated intercontinental movements[103]. Other bovids in Africa with Asian origins include: the reduncin *Kobus porrecticornis*, which has a first appearance date (FAD) in Africa in the late Miocene deposits at Mpesida, Kenya, dated to about 7.3–6.2 Ma (refs. 106–108); the antilopin *Prostrepsiceros vinayaki* with a FAD in the late Miocene Middle Awash deposits, Ethiopia, dated to about 5.77–5.54 Ma (refs. 90,109); and which is also found in the Baynunah

Formation[2], and potentially another antilopin that is recorded in the late Pliocene Shungura Member C deposits, Ethiopia, dated to about 3.0–2.6 Ma (refs. 110,111), and may be closely related to the South Asian *Antilope subtorta*. Three canid species dispersed to Africa from Asia during this period: *Canis* sp., which originated in North America, has its FAD in Africa at the late Pliocene site of South Turkwel, Kenya, dated to about 3.6–3.2 Ma (ref. 112); *Nyctereutes lockwoodi* is first recorded in Africa at the late Pliocene site of Dikika, Ethiopia, dated to about 3.42–3.24 Ma (ref. 113); and *Lycaon* sp. is first recorded at the early Pleistocene sites of Ain Hanech in Algeria and Olduvai in Tanzania, dated to about 1.8–1.7 Ma (refs. 114,115). In addition, leporids make their first appearance in Africa at the late Miocene sites in Ethiopia and Chad, dating to about 7.5-6.8 Ma (refs. 116,117); the suid *Metridochoerus* is first recorded at the late Pliocene site of Usno 12, Ethiopia, dated to about 3.3–3.0 Ma (ref. 118); and the hippopotamid *Hexaprotodon bruneti*, with strong affinities to the South Asia hexaprotodons, was discovered in the Middle Awash of Ethiopia, dating to about 2.5 Ma (ref. 119).

Movements in the opposite direction also occurred. The hippotragin *Hippotragus brevicornis* and the proboscidean *Elephas planifrons* both have FADs in Eurasia in the Tatrot Formation (about 3.5–3.3 Ma) of the Upper Siwaliks, a geological formation exposed along the foothills of the Himalaya[120–122]. The overlying Pinjor Formation, broadly dated to about 2.7–0.6 Ma, also includes the first Eurasian occurrences of the hippotragines *Hippotragus bohlini* and *Sivoryx sivalensis*, the alcelaphin *Damalops palaeindicus* and the proboscidean *Elephas hysudricus*[100,120,122,123]. The first occurrence of the equid *Eurygnathohippus* outside of Africa was recently reported from Potwar Plateau in Pakistan and the Siwalik Hills, dated to about 3.6–2.58 Ma (ref. 124). Earlier in the Siwaliks in northern Pakistan, remains of *Hexaprotodon sivalensis* have their FAD at about 6.2 Ma and perhaps as early as about 7.2 Ma (refs. 125,126). Further east in China, the hyaenid *Crocuta* has its FAD in the Longdan Basin at about 2.2 Ma (ref. 127) and the larger *Pachycrocuta brevirostris* has its FAD in the Nihowan Basin at about 3.0–2.5 Ma (ref. 128).

It is clear from the above that various taxa moved between Africa and Eurasia at various times during the Neogene and Quaternary and following the reduction of the Old World savannah palaeobiome[90–92]. Poor dating and a patchy fossil record makes constraining the timing of these movements difficult, with taxonomic uncertainties and ambiguities complicating matters even further. Nonetheless, considering our findings, it seems probable that movements between Africa and Eurasia would have taken place through the Arabian Peninsula, which, during humid phases, was likely characterized by well-watered grasslands and woodlands.

Indeed, most of the aforementioned herbivores were grazers, as shown by studies of stable isotopes, tooth morphology and wear, and reference to modern analogues[129–134]. Perhaps the one exception to this are members of the tribe Antilopini, although these animals tend to live in relatively arid environments (for example, gazelles), subsisting largely on dryland shrubs and bushes[129]. The aforementioned carnivores were likely either generalists or were well suited to grassland and arid environments. Today, hyaenas occur primarily in the arid and savannah zones of sub-Saharan Africa, and, in the case of the striped hyena (*Hyaena hyaena*), northern Africa and western Asia[135]. Of the African carnivores, the African wild dog (*Lycaon pictus*) is the most widespread and occupies the greatest range of habitats[135]. The living racoon dogs (*Nyctereutes* spp.) occupy rather forested environments; however, on the basis of tooth morphology, a study[113] suggested that *N. lockwoodi* may have had a similar niche to the side-striped jackal (*Canis adustus*), which today is found in the wooded areas of the African savannahs[135].

In summary, it appears that many of the abovementioned taxa would have been well suited for life in Arabia during the wet episodes identified in our speleothem record. The scant fossil record aside, we suggest that Arabia probably acted as a hitherto unrecognized but important crossroad for biogeographic exchange between Africa and Eurasia

over the past 8 Ma. The nature of these exchanges, and the exact role of Arabia in these, may only be elucidated with an improved fossil record, better dating, and phylogenetic studies using ancient DNA and proteomics.

## Data availability

The data supporting this paper are publicly available in various repositories. The software used for base maps is ESRI Arc GIS Pro, using a topographic hillshade underlay created from the CGIAR Version 4 SRTM dataset (Fig. 1), which is available from the CGIAR-CSI SRTM 90 m Database (http://srtm.csi.cgiar.org) as described in ref. 136. The dataset on global land precipitation from WorldClim is available at https://www.worldclim.org/data/v1.4/worldclim14.html. The dataset on the Sahara dust flux used in Fig. 3d can be found at https://doi.org/10.5281/zenodo.6594643, and the dataset for the benthic $\delta^{18}O$ record is available at https://doi.org/10.5281/zenodo.6311999. The East Africa Soil Carbonate Stable Isotope Data used in Fig. 2d is available at https://doi.org/10.1594/IEDA/100231. The dataset for the Glacial Index used in Extended Data Fig. 8 is available in the PANGAEA repository at https://doi.org/10.1594/PANGAEA.914483. The dataset for the SSTs in the meridional temperature gradient calculations (for example, Fig. 3b) can be found at https://doi.org/10.1594/PANGAEA.885390. In addition, we declare that the data presented in this study are available within the paper and its Supplementary Information files, which can also be accessed via the figshare repository at https://doi.org/10.6084/m9.figshare.28297397.

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

**Acknowledgements** We thank HH Prince Badr Bin Abdullah Bin Farhan Al-Saud, Saudi Minister of Culture, for permission to conduct research in Saudi Arabia. We acknowledge J. Al-Herbish, CEO of the Heritage Commission (HC), Saudi Ministry of Culture, for support and assistance with the fieldwork. A. Al Zahrani, Head of the Archaeology Sector at the HC, assisted the team in implementing all phases of the project. We thank the Saudi Geological Survey past presidents and current chairman and CEO Eng. A. M. Shamrani for logistical and technical support. We also thank geologists S. A. Alsoubhi, M. Haptari, N. Al Asmari and H. Hashim for their assistance in the field. M.M. acknowledges the support of the Royal Society University Research Fellowship grant entitled 'Unlocking the drivers of global desert expansion in warmer and colder worlds' (grant ref: URF\R1\231546). M.D.P. acknowledges support for the Green Arabia project from the Australian Research Centre for Human Evolution, Griffith University. A.M.A. acknowledges the support of the Nature and Science Researchers Supporting Project (NSRSP-2024-5), DSFP, King Saud University, Riyadh, Saudi Arabia. Participation in the fieldwork for E.A., R.C.-W. and J.B. was funded by the Leverhulme Trust 'Unravelling the pattern, impacts and drivers of early modern human dispersals from Africa' project (grant ref: RPG-2017-087). Funding for A.N.M. was provided by the German Research Foundation (DFG) priority programmes 'SPP-1833 Building a Habitable Earth' (WE 2850/17-1) and 'SPP-2238 'Dynamics of Ore Metals Enrichment' (MA 9571-3-1). D.S. thanks the DFG for funding through grant INST 247/889 FUGG. Substantial funding was provided by the Max Planck Society (through N.B. and G.H.H.) with additional support from the Dr Abdulrahman Al Ansari Award to the Green Arabia Project, the Saudi Heritage Commission, Saudi Geological Survey and the Leverhulme Trust. R.A. and A.G. acknowledge this work as Fierce contribution number 175.

**Author contributions** M.M. wrote the paper with substantial scientific contributions from M.D.P., H.B.V., H.S.G., M.S., N.B. and N.D. Radiometric dating was conducted by M.M., D.S., M.W., A.G. and R.A. Fieldwork campaigns were conducted by H.B.V., H.S.G., P.S.B., N.D., M.S., E.A., J.B., N.B., A.B., R.C.-W., G.J.P., E.M.L.S., N.V., A.M.A., A.A.A.O., Y.S.A.A.-M., M.A., M.A.-S., I.Z. and M.D.P. Fieldwork was planned by H.S.G., M.D.P., the Saudi Heritage Commission (Ministry of Culture) and the Saudi Geological Survey. Sample preparation and cutting was performed by M.M. and A.B. The maps in Fig. 1 were created by P.S.B. and M.M. Raman measurements and were conducted by A.N.M. and M.M. Calculation and interpretation of meridional sea surface temperature gradients was conducted by M.M. and A.M.-G. Permits and local support was facilitated by A.M.A., F.A.-J., A.A.A.O., Y.S.A.A.-M., M.A., M.A.-S. and I.Z. All authors contributed to reading and editing the text.

**Funding** Open access funding provided by Max Planck Society.

**Competing interests** The authors declare no competing interests.

**Additional information**
**Correspondence and requests for materials** should be addressed to Monika Markowska, Hubert B. Vonhof, Huw S. Groucutt or Michael D. Petraglia.

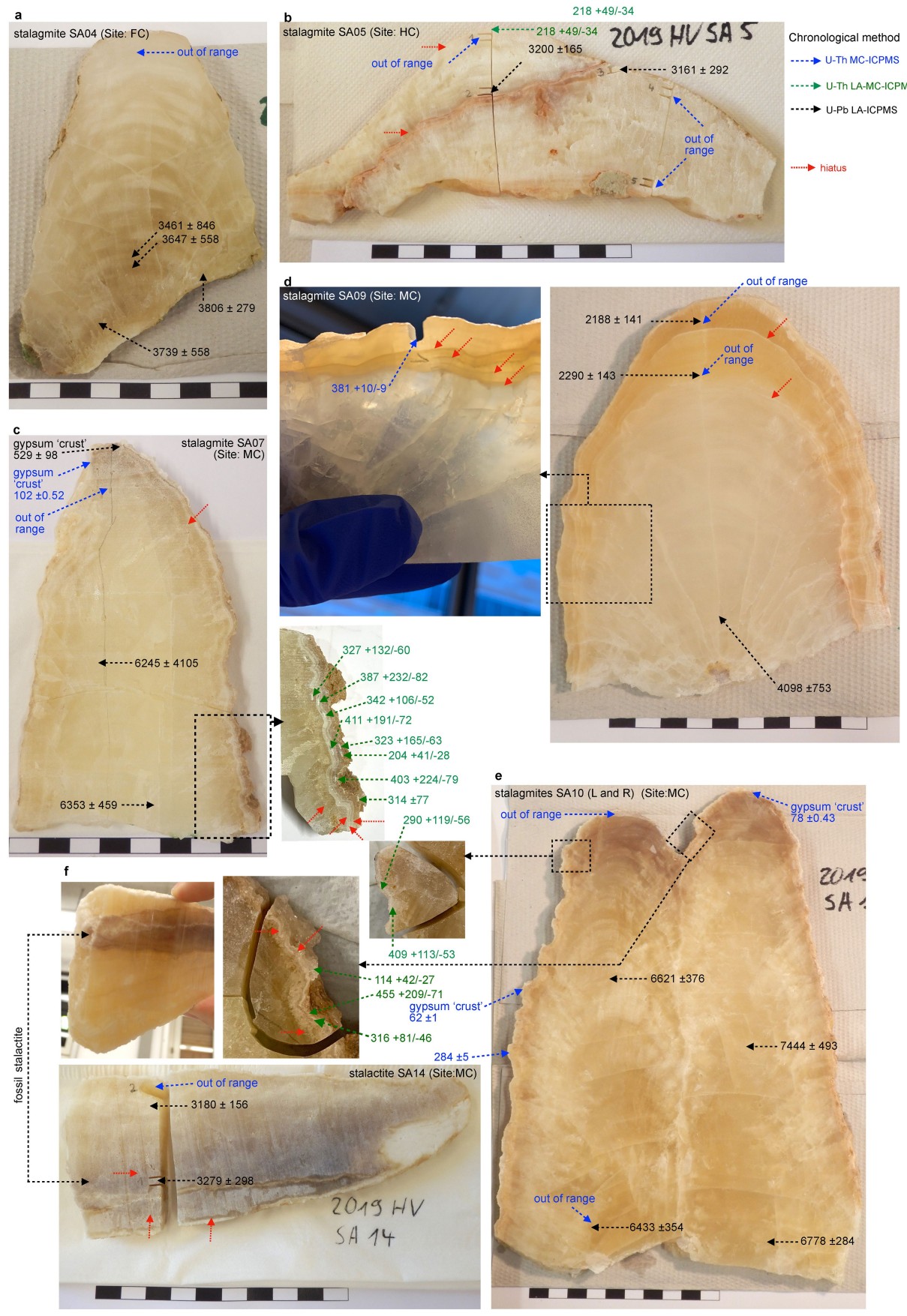

**Extended Data Fig. 1 | Central growth axis thick sections (a to f) with U-series ages in ka and sampling locations, with the 95% confidence interval uncertainty.** Cave abbreviations include FC: Friendly Cave, HC: Hotel Cave, MC: Mossy Cave, LC: Luxury Cave, 1000 WC: 1000 Wings Cave, EP19.8: EP19.8 Cave, MuC: Murubbeh Cave.

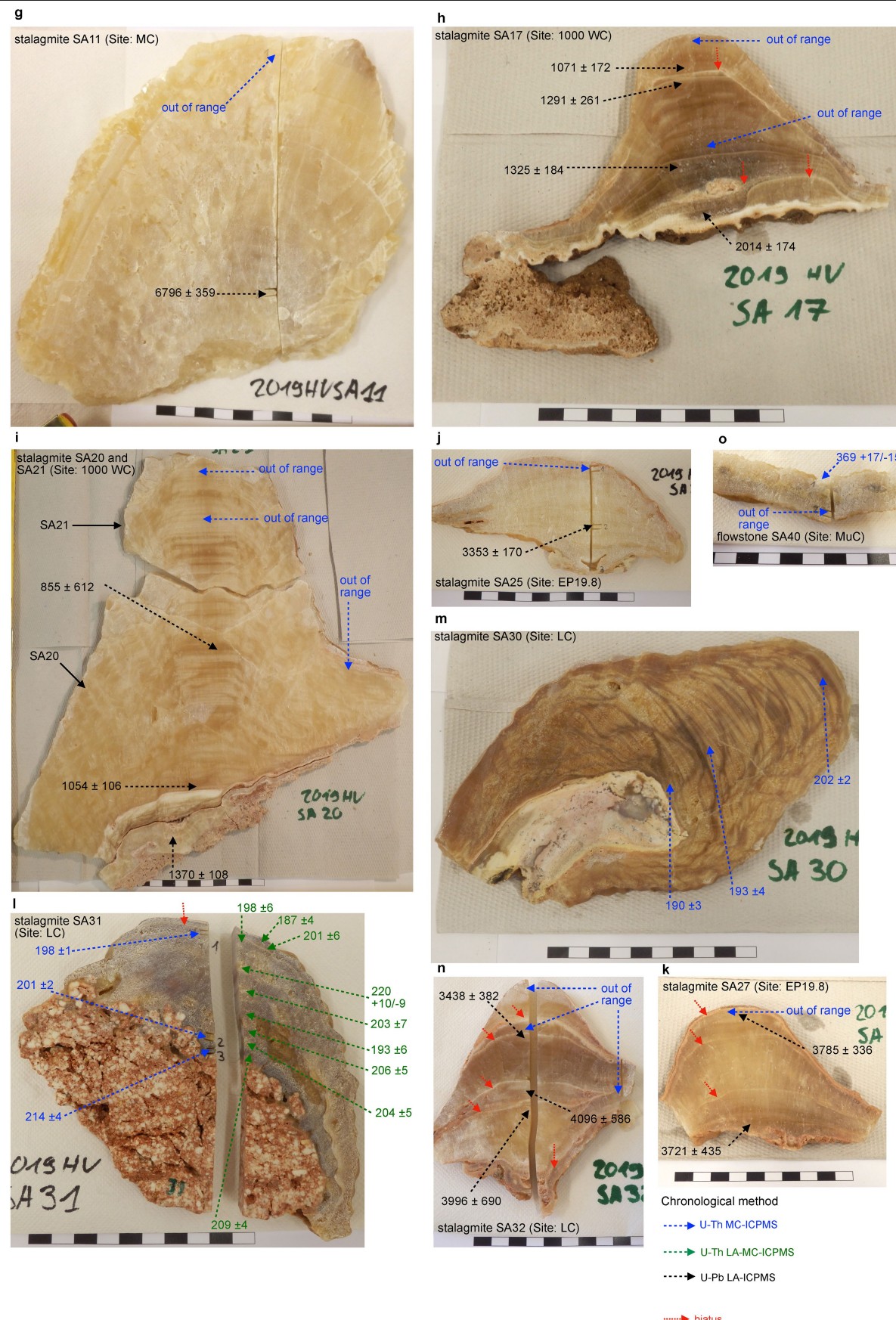

**Extended Data Fig. 2 | Central growth axis thick sections (g to o) with U-series ages in Ka and sampling locations, with the 95% confidence interval uncertainty.** Cave abbreviations include FC: Friendly Cave, HC: Hotel Cave, MC: Mossy Cave, LC: Luxury Cave, 1000 WC: 1000 Wings Cave, EP19.8: EP19.8 Cave, MuC: Murubbeh Cave.

Chronological method

- - - ▶ U-Th MC-ICPMS

- - - ▶ U-Th LA-MC-ICPMS

- - - ▶ U-Pb LA-ICPMS

······ ▶ hiatus

**p**

stalagmite SA35 (Site: LC)

out of range - - - ▶

571±23

out of range

604 ±19

653 ±66

553 ±20

19 HV SA 35

**q**

stalagmite SA01 (Site: MC)

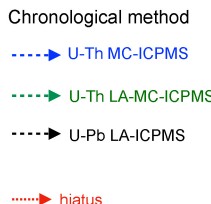

out of range

2019 H SA

4054 ±1471

out of range

**s**

stalagmite SA26
(Site: EP19.8)

out of range

663 ±84

2019 HV SA 26

**r**

stalagmite SA19 (Site: 1000 WC)

out of range

990 ±176

**t**

347.0 ±16

stalagmite SA18 (Site: 1000 WC)

**Extended Data Fig. 3 | Central growth axis thick sections (p to r) with U-series ages in Ka and sampling locations, with the 95% confidence interval uncertainty.** Cave abbreviations include FC: Friendly Cave, HC: Hotel Cave, MC: Mossy Cave, LC: Luxury Cave, 1000 WC: 1000 Wings Cave, EP19.8: EP19.8 Cave, MuC: Murubbeh Cave.

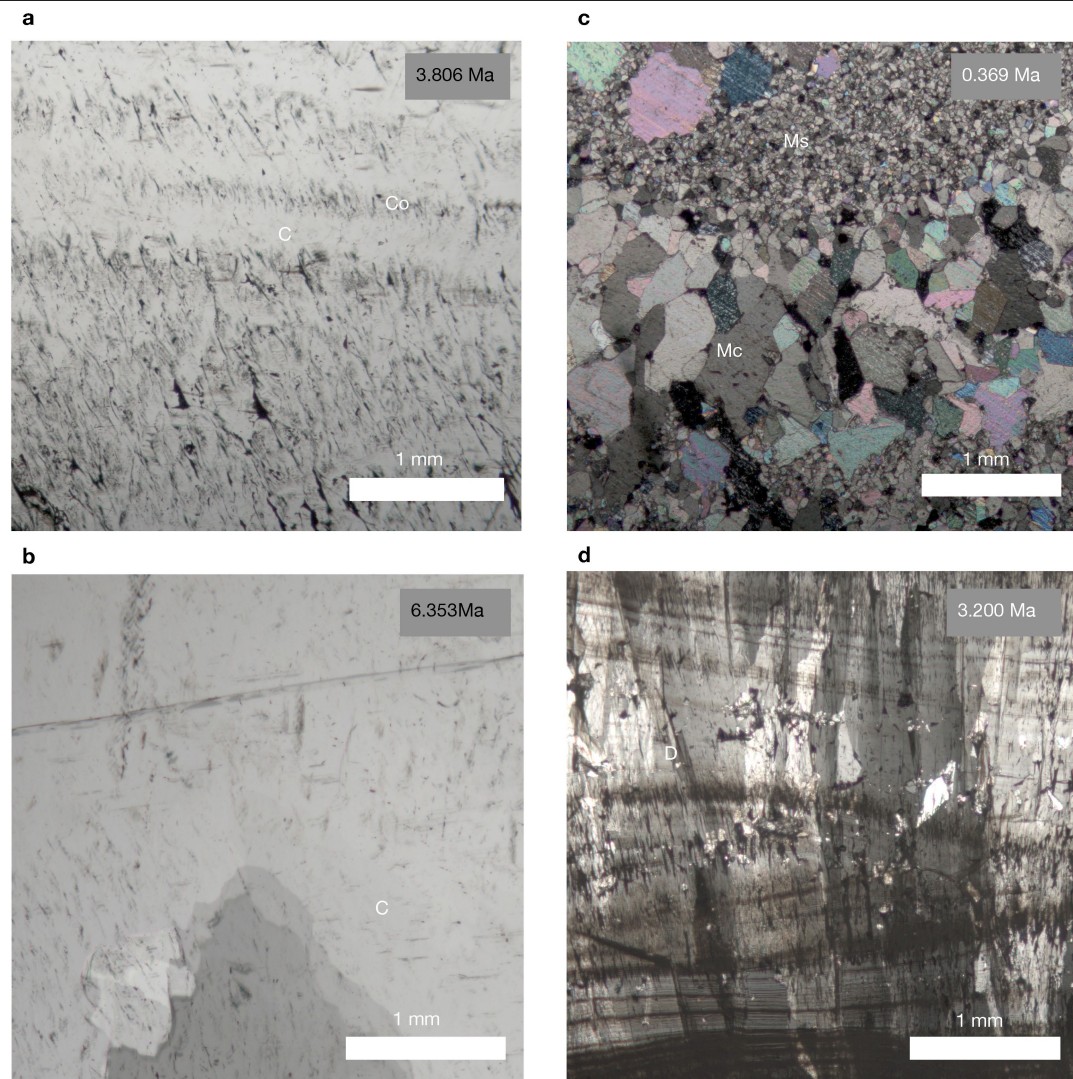

**Extended Data Fig. 4 | Thin section microscope images with a cross-polarising lens.** a) SA04, b) SA07, c) SA40 and d) SA05 (top section). Fabric codes are Columnar (C), Columnar Open (Co), Dendritic (D), Microsparite (Ms) and Mosaic Calcite (Ms).

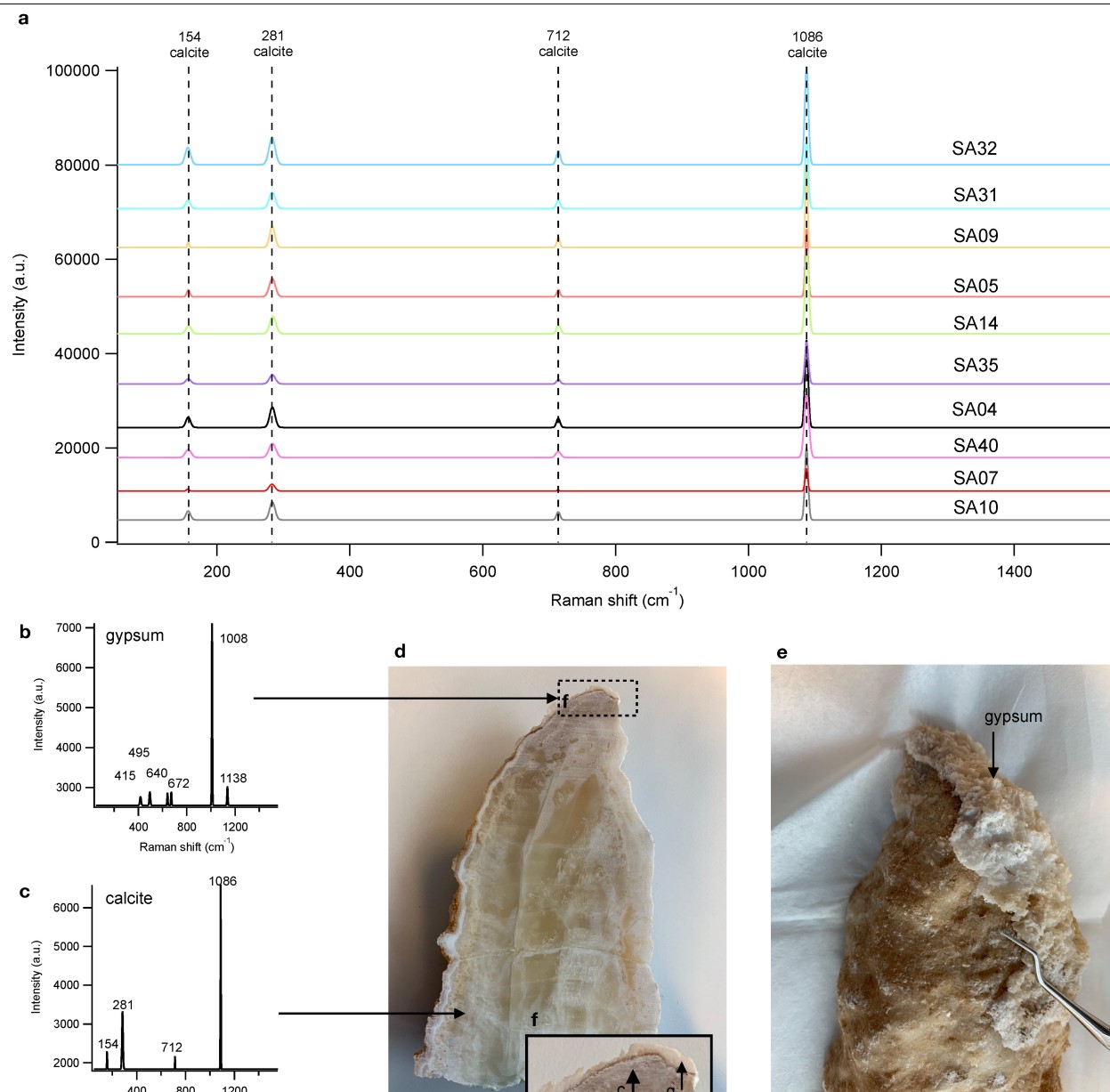

**Extended Data Fig. 5 | Carbonate mineralogy results from raman spectroscopy.** Panel a: General minerology identification for speleothems SA32, SA31, SA09, SA05, SA14, SA35, SA04, SA40, SA07 and SA10. Panel b: SA07 example of calcite speleothem covered by a gypsum crust. Panel c shows the Raman spectra for gypsum measured and Panel d shows the same for calcite. Panel d is a thick section of stalagmite SA07 with the calcite and gypsum layers indicated and Panel e shows the outside of the stalagmite SA07 with the exposed gypsum crust. Panel f shows a zoomed in picture of the thick section in Panel d with calcite (c) and gypsum (g) layers indicated.

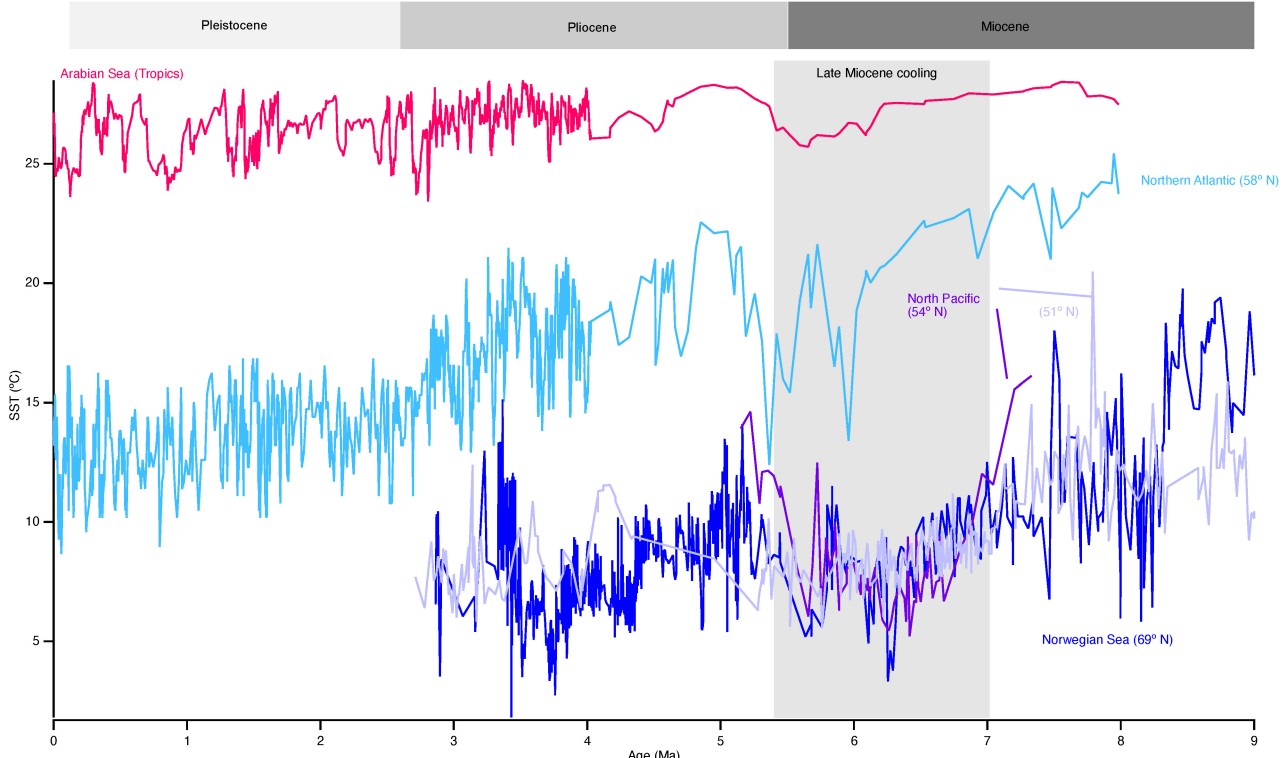

**Extended Data Fig. 6 | North Hemisphere Sea Surface temperature (SSTs).** The Arabian sea SSTs (site ODP 722; 16°37.31′N, 59°47.76′E), North Atlantic SSTs (site ODP 982; 57°30.05′N,15°52.03′W), North Pacific (sites ODP 887; 54°21.921′N, 148°26.765′W, and 883; 51°12′ N, 167°46′ E), and Norwegian Sea (site ODP 907; 69°14.989′N, 12°41.894′E) from the compilation in Herbert et al.[17] and references therein.

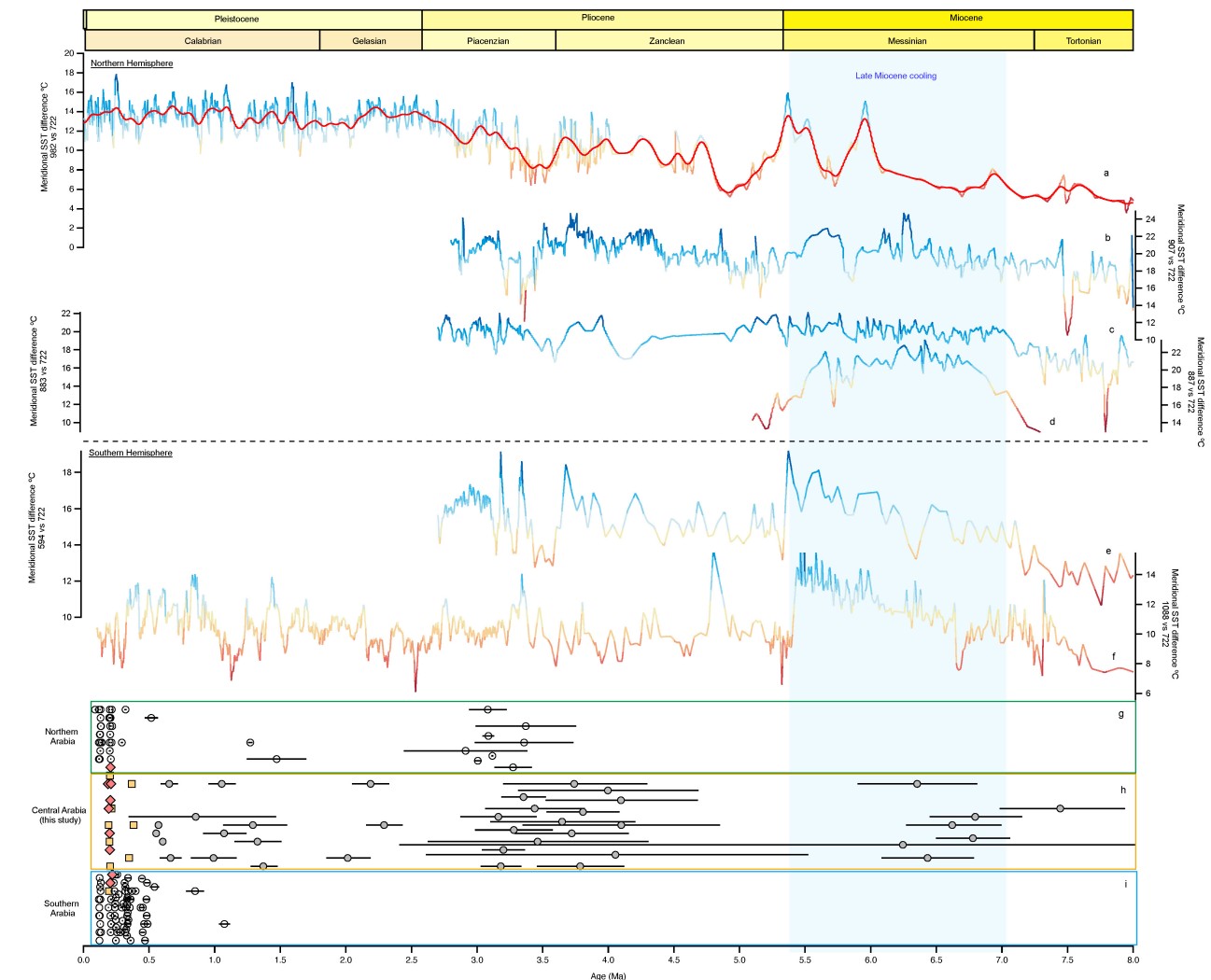

**Extended Data Fig. 7 | Global Meridional Temperature Gradients (MTGs) and Arabian speleothem growth periods over the last 8 Ma.** MTGs in the northern (a-d) and southern hemispheres (e-f)[137] are compared to speleothem growth periods in this study (h) as well as in northern Arabia (Negev Desert[20] (g)) and southern Arabia (Yemen and Oman[9]) with the 95% confidence interval uncertainty indicated by the horizontal error bars. (i). Northern hemisphere MTGs for sites ODP 982 (57°30.05′N, 15°52.03′W) (a), ODP 907 (69°14.989′N, 12°41.894′E) (b), ODP 883 (51°12′N, 167°46′E) (c), ODP 887 (54°21.921′N, 148°26.765′W) (d), and southern hemisphere sites DSDP 594 (45° 31′24.6′S, 174° 56′ 52.8′E) (e) and ODP 1088 (41° 8′ 9.96′S, 13° 33′ 46.08′E) (f) from the data compiled in Herbert et al.[17].

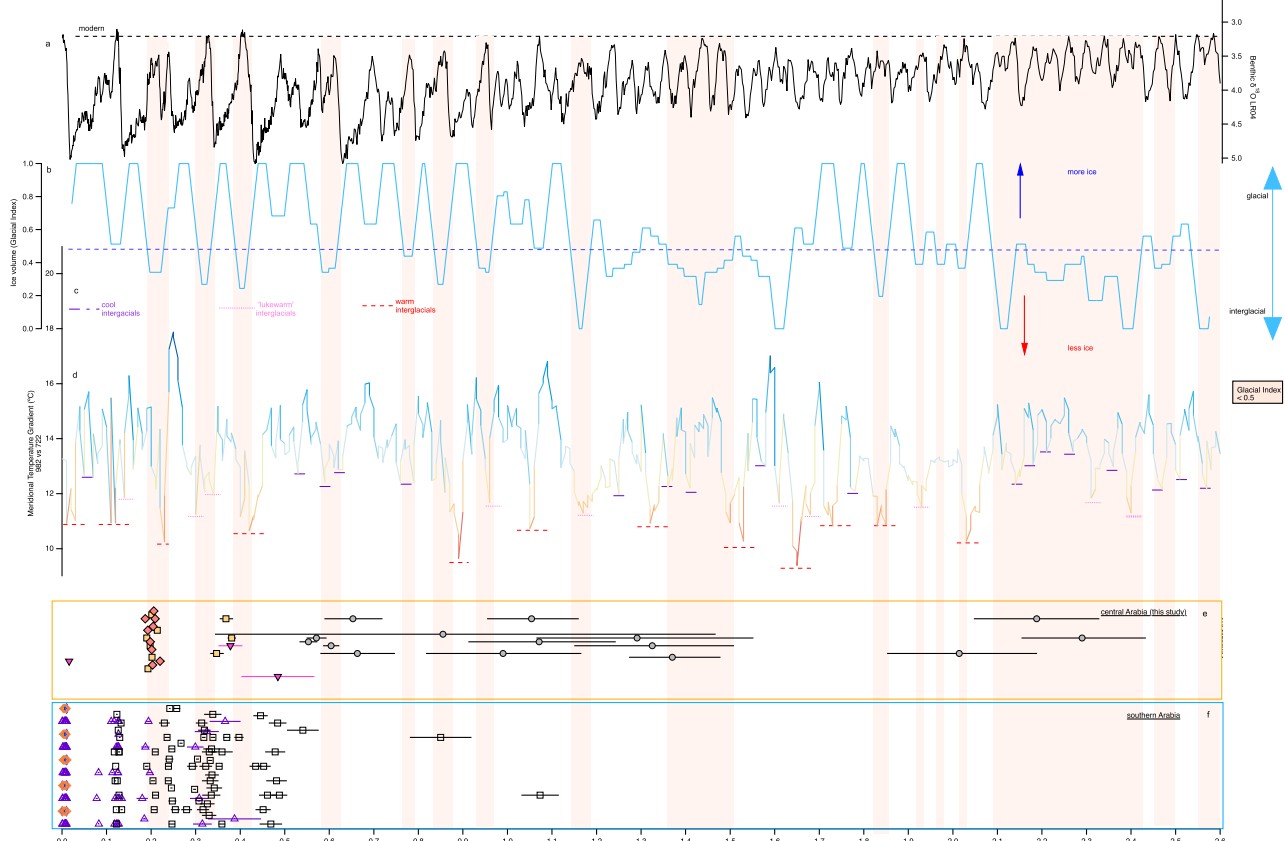

**Extended Data Fig. 8 | Pleistocene hydroclimate of the central Arabian desert and major climate variables.** a) Benthic LR04 $\delta^{18}$O (‰ VPDB) stack[85]. The modern values are indicated by the dashed black line. b) Glacial index, where 1 is a full glacial and 0 is a full interglacial, for full details on calculation see Koeler and van de Wal[84]. The blue dashed line is used as the cutoff (0.5) for significant interglacial phases as indicated by the red panels, where 1 indicates a full glacial endmember and 0 a full interglacial endmember. c) indicates the key for cool, lukewarm and warm interglacial phases. d) Meridional Temperature Gradients calculated from marine sedimentary cores ODP 982 and 722[17] where

warm (MTG < 11), lukewarm (11 > MTG < 12) and cool interglacials (MTG > 12) are identified by red dashed, pink dashed and purple lines, respectively. d) Negev desert speleothem ages[20]. e) Central Arabian speleothem ages (this study) where grey circles indicate U-Pb ages, yellow diamonds U-Th ages by solution MC-ICPMS and red squares U-Th ages by LA-MC-ICPMS. f) Southern Arabia speleothem ages from a compilation of cave sites[9,52,138–140]. The horizontal error bars of ages estimates refer to the 95% confidence interval uncertainty. The shaded background panels indicate periods where the glacial index is <0.5.

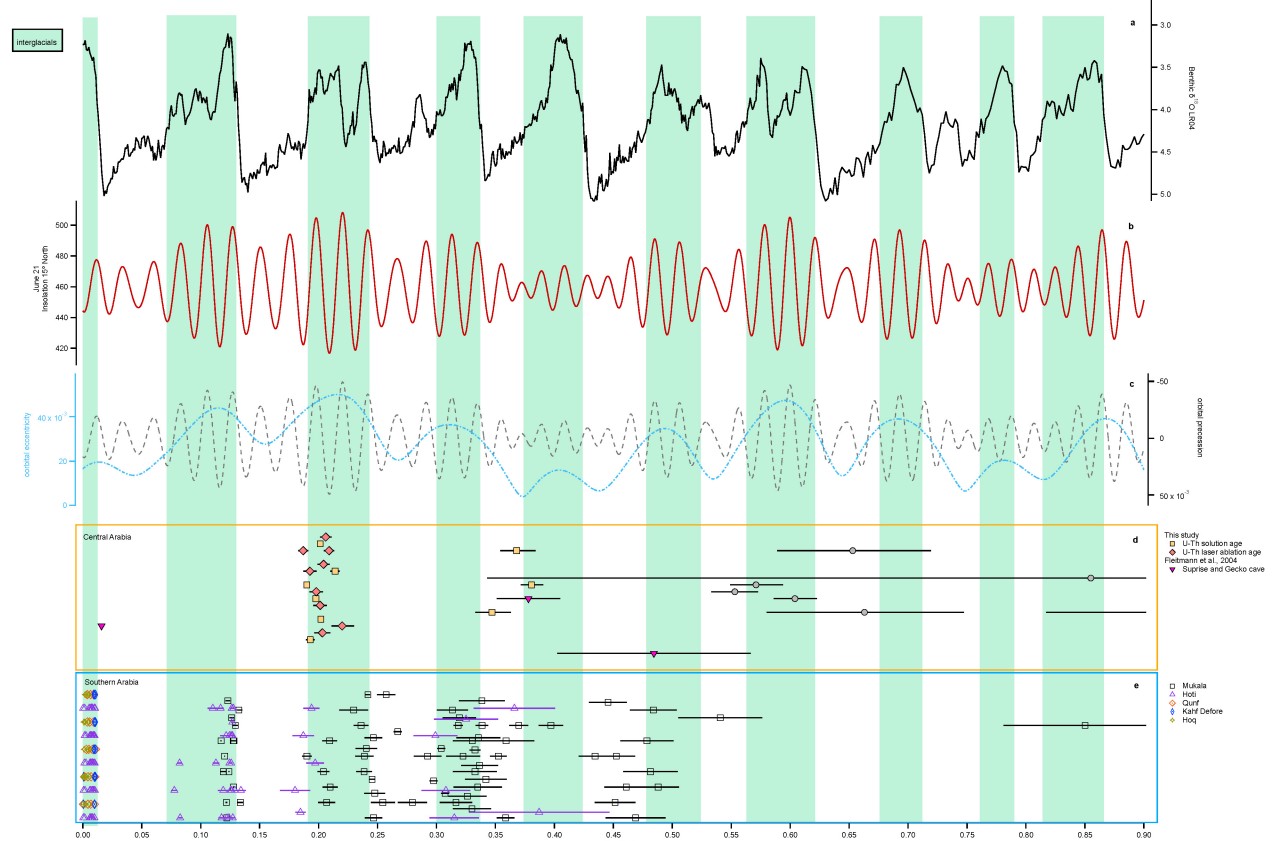

**Extended Data Fig. 9 | Arabian humid episodes and global climate cycles.** Panel a shows the LR04 benthic stack[85]; Panel b is the June 21 insolation anomaly at 15° N[141]; Panel c is the eccentricity anomaly (blue) and climate precession anomaly (black)[141]; Panel d shows the central Arabian U-Th and U-Pb dates as well as two ages reported by Fleitmann et al.[52], Panel e are the ages from a compilation of caves in southern Arabia[9,52,138–140]. The horizontal error bars of ages estimates refer to the 95% confidence interval uncertainty. The green background shading indicates interglacial phases.

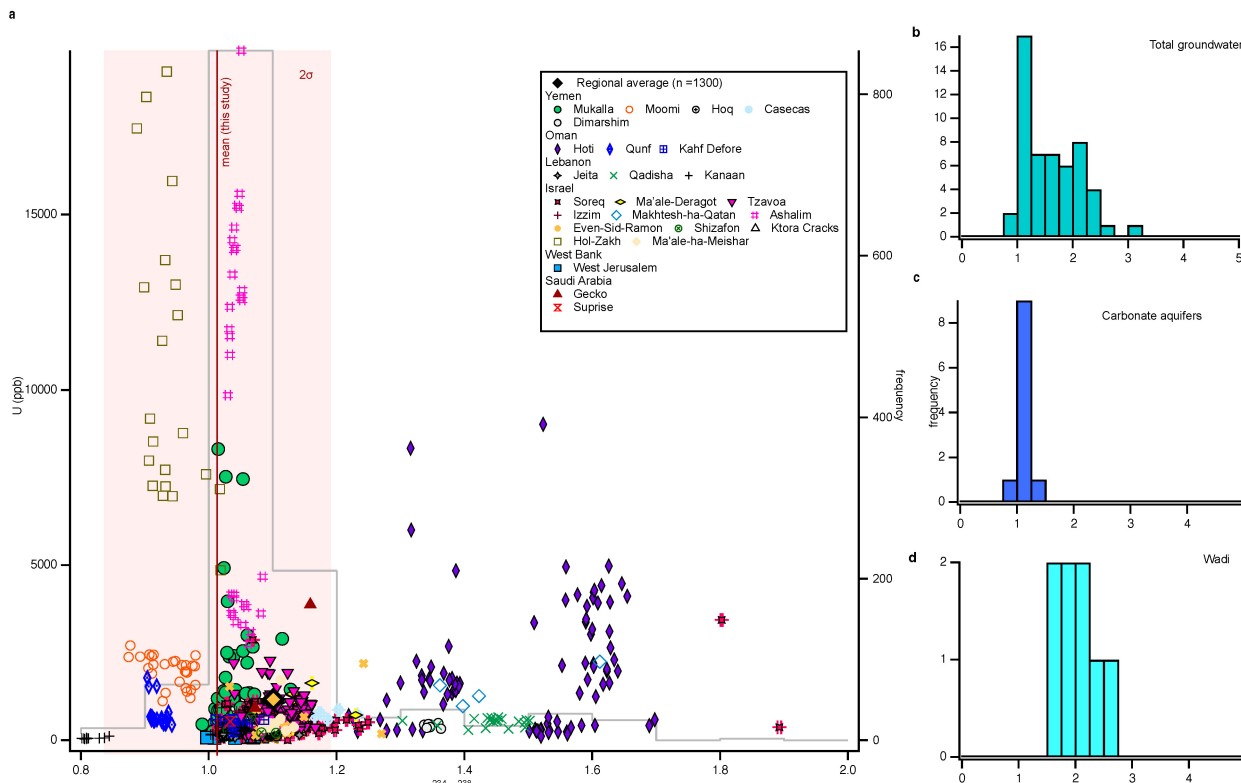

**Extended Data Fig. 10 | Distribution of regional speleothem $^{234}U/^{238}U_{initial}$ and U concentrations (ppb) and a histogram with the frequency of regional $(^{234}U/^{238}U)_{initial}$.** Panel a shows sites from Mukalla[9], Moomi[142], Hoq[140], Casecas[140], Dimarshim[139], Hoti[139,143], Qunf[139], Kahf Defore[138,139], Jeita[144], Qadisha[145], Kannan[146,147], Soreq[148–150], Ma'ale-Deragot[151], Ma'ale ha-Meishar[152], Tzavoa[151], Izzim[152], Makhtesh-ha-Qatan[152], Ashalim[152], Even-Sid-Ramon[152], Shizafon[152], Ktora Cracks[152], Hol-Zakh[152], West Jerusalem[153], Gecko[52], and Suprise[52]. U concentration data for caves Hartuv Quarry[150,154], Ma'ale-Efrayim[65,155], Small Ma'ale-Efrayim[65] Kana'im[155], Cave 81[155], Na'chal Ami'az[155] is unavailable and therefore are not shown in the graph, but the $(^{234}U/^{238}U)_{initial}$ values are used in the calculation of the regional average (n = 1300). Panels b-d show the regional groundwater $(^{234}U/^{238}U)$ for total groundwater (b), carbonate aquifers (c) and Wadis (d) in Saudi Arabia.

**Extended Data Table 1 | Results from ANOVA and post hoc Tukey tests for carbonate and fluid inclusion stable isotopes, p-values have 95% confidence intervals**

| Time Epoch | $\delta^{18}O_{carb}$ | $\delta^{18}O_{FI}$ | $\delta^2H_{FI}$ |
|---|---|---|---|
| Late Miocene and Early-to-Mid Pleistocene | 0.00 | 0.68 | 0.04 |
| Late Pleistocene and Early-to-Mid Pleistocene | 0.00 | 0.00 | 0.00 |
| Pliocene and Early-to-Mid Pleistocene | 0.00 | 0.77 | 0.69 |
| Late Pleistocene and Late Miocene | 0.00 | 0.00 | 0.00 |
| Pliocene and Late Miocene | 0.00 | 0.14 | 0.17 |
| Pliocene and Late Pleistocene | 0.00 | 0.00 | 0.00 |