## [Peer Review File · Nature]

Recurrent humid phases in Arabia over the past 8 million years

Corresponding Author: Dr Monika Markowska

Version 0:

Reviewer comments:

Referee #1

(Remarks to the Author)

This work by Markowska et al., details recurring humid geographical phases of the Arabian Peninsula through the last 8 Myr by way of proxy information from speleothems. Wetter periods are inferred from both the presence of speleothem growth, and by encapsulated H and O isotopic fractionation. Wetter times, referred to as Central Arabian Humid Periods (CAHPs), are inferred by the presence of speleothem growth, as speleothems require both moisture and the presence of organic respirators (such as plants) to produce acidic water for the secondary precipitation of calcite. The use of speleothem U-Pb ages to extend the paleoclimate record of the Arabian region is particularly inspired in this work. It provides a high-impact method of producing reliable ages much further into the geological past than traditional U-Th ages alone. Moreover, using laser ablation to produce U-Pb ages in this work helps cement this acquisition method's utility in the broader field. I have re-analysed the reported raw U-Th and U-Pb data and produced concordant ages, suggesting to me that the provided U-Th-Pb ages are of high fidelity. My only concern is their $^{234}\text{U}/^{238}\text{U}$ initial disequilibrium values applied to the U-Pb ages (outlined below). However, I do not think that this choice will significantly impact their CAHP interpretations. I see no reason from a speleothem age perspective that this work should not be accepted.

I believe this work is novel, of high importance, and broad interest to the wider field. The recurring "greening" of this region provides a useful timing for the land migration of fauna (including hominids) between Eurasia and Africa. This greening has been well-documented, in my opinion, by the speleothem record. I think this work should be accepted for publication following the address of the listed comments below. I would like to commend the authors on a well-written piece and for this impressive work.

Sincerely,
Dr. John Engel
(jengel@lanl.gov)

Comment 1) On Fig 3, the ^{18}O values seems to overlap (at least by eye) from 1Ma backwards. Can the authors please include the 2σ deviation with the average values. Have statistical tests been done to distinguish these groups to a significant level? I would suggest an ANOVA (one-way) of the oxygen isotopes (Fig 3g and Fig 4c) to test statistical significance between the CAHPs to see if their variances overlap. The omnibus ANOVA will test the effect of age on oxygen isotope fractionation (i.e., rainfall), and the post-hoc analysis will say which age groups are significantly different from each other.

Comment 2) Figure 2: Why were 250 Ka bin sizes chosen for U-Pb?

Comment 3) L317: "The largest temporal cluster of U-Pb ages around 3.5 Ma". Was this cluster of U-Pb ages from a single cave chamber or from multiple caves/chambers? If the majority of ages are from a single cave chamber, this cluster may represent a preservation bias, not necessarily a trend of the region's speleogenesis potential. For example, this single chamber (if that's the case) may have selectively survived this relatively-active geological area (<https://doi.org/10.3389/feart.2022.851737>).

Comment 4) L599: The use of younger U-Th values to estimate the $^{234}\text{U}/^{238}\text{U}$ initial disequilibrium of U-Pb ages is potentially problematic. First, please state the average U-Th based $^{234}\text{U}/^{238}\text{U} \pm 2\sigma$ here in the text. Additionally, in the

supplementary documents, please explicitly list the U-Th speleothems, and their $^{234}\text{U}/^{238}\text{U}$, used to calculate this value. Can you please explain your rationale for this decision? Initial disequilibrium is one of the chief impediments to accurate U-Pb calcite ages (in addition to very small analyte concentrations). Consider the hypothetical situation where 6 Myr ago, the source waters entering the karst were interacting with a local granite and extracting a very different $^{234}\text{U}/^{238}\text{U}$ than those recorded by the U-Th ages <700ka. I'm not saying this is necessarily the case, but I believe your applied value of 0.939 ± 0.054 (inferred from the supplementary file) is artificially small and may not incorporate this potential geological scatter. I would suggest two potential options. First, expand the uncertainty to a relatively large, arbitrary $^{234}\text{U}/^{238}\text{U}$ initial disequilibrium, say 1 ± 0.2 , and see if the CAHPs change from the amended U-Pb ages. My guess is that given the scale of Fig 3 x-axis, the CAHP's will not change significantly. Second, you might try our 2019 approach (<https://doi.org/10.1016/j.quageo.2019.101009>) where we used ^{235}U - ^{207}Pb isochrons (referenced to ^{208}Pb) as a way to bypass ^{234}U disequilibrium (at the cost of reduced precision). I see that ^{235}U was not an analyte measured during laser ablation, but you might estimate it using natural U composition from the ^{238}U signal. Tim Pollards' new software may help with this (<https://doi.org/10.5194/gchron-5-181-2023>)

U-Pb supplementary data

U-Pb calculates ages sheet

Comment 5) Can you please provide more information about which samples these data are coming from? For example, are SA17 ages 6, 5, 3, and 2 all from the same speleothem? This would allow readers to better interpret the CAHP periods based on U-Pb ages. For instance, SA17-2 and SA17-3 are within age uncertainty: if they're from the same sample, wouldn't this be "double dipping" of ages to increase the population sizes for the CAHPs? To help make this more clear, you could color-code the U-Pb ages in Figure 3 to match the oxygen isotopes plotted below them.

Comment 6) On point (6) please provide the 2σ uncertainty of the averaged $^{234}\text{U}/^{238}\text{U}$ calculated from the U-Th ages.

Comment 7) On point (8) I was not certain about the interpretation given. Is it perhaps supposed to be the added uncertainty (with (7) being the subtracted uncertainty) on applied initial $^{234}\text{U}/^{238}\text{U}$?

Sample raw data sheet

Comment 8) The error correlation value rho seems abnormally small in this work, especially when compared with previous laser ablation calcite studies (e.g. <https://doi.org/10.5194/gchron-3-35-2021>). Is this a unique consequence of your instrumentation? If so please provide context for why this is the case.

Comment 9) Please elaborate on your criteria for removing certain points from the U-Pb isochrons (e.g., SA10-6).

Referee #2

(Remarks to the Author)
Please see pdf attached.

Referee #3

(Remarks to the Author)

A. Summary of the key results

This paper presents a speleothem-based hydroclimatic record that documents recurrent humid phases in central Arabia over the last 8 Ma, potentially facilitating dispersals and biogeographical exchange at this crossroads between Africa and Eurasia.

B. Originality and significance:

The discovery, dating, and analysis of the speleothems in the heart of the Arabian Peninsula in modern-day Saudi Arabia is highly significant. As the authors note, it fills a critical gap in our understanding of the paleoclimatic history of the Arabian Peninsula between the Miocene and the Middle to Late Pleistocene, a gap of some 5 million years during which many dynamic climatic and evolutionary developments were taking place in Africa and Eurasia, but of which we had no information for Arabia. The discovery of a trend of progressively greater aridity in the $\delta^{18}\text{O}$ data is also very interesting. I think this is an important discovery that is relevant for our understanding of paleoclimatic changes that not only affected Arabia, but the entire Saharo-Arabian belt, and faunal (including hominin) exchange between Africa and Eurasia during the late Neogene.

C. Data & methodology

The speleothem data, and especially the U-Pb age dating, should be closely examined and reviewed by specialists in this field. The discussion and conclusions drawn regarding the interchange of large mammals between Africa and Eurasia, is however not very convincing and has some fundamental problems discussed in detail below. However I think it can largely be removed as the speleothem data is the main discovery here.

D. Appropriate use of statistics and treatment of uncertainties

In Fig 2, what are the error bars on the age samples in Fig. 2g? Are these 1 standard error, or standard deviation, or are they the 95% age confidence interval? I hope the latter, otherwise the actual 95% age estimates could end up spanning a million years or more. This is important because the humid periods in this figure are drawn against the maximum error bars of the individual age samples (which is in itself debatable). If the age uncertainty is too great, then there would be little statistical basis for arguing for discrete humid/arid periods.

E. Conclusions: robustness, validity, reliability

The conclusions seem fine to me, with the exception of the claim that "Taxa with African affinities that appear in Asia, and vice versa, are considered to reflect dispersals between these two regions following periods of climate amelioration in the Saharo-Arabian Desert." I think the analysis on which this conclusion is based is not credible for reasons discussed below. However, I also think it is unnecessary to the main point of the paper, and removing (or significantly toning down) these claims will not detract from the importance of the paper.

F. Suggested improvements: experiments, data for possible revision

Abstract: "Wetter conditions correspond with patterns of mammalian dispersals between the Afrotropical and Indomalayan realms, highlighting the role of Arabia as a key crossroad for continental-scale biogeographic exchanges." I do not think that you have demonstrated (or can with such small sample sizes, especially given the total lack of an Arabian record during the Pliocene – Early Pleistocene) that the identified humid intervals correspond to times or 'patterns' of dispersal across Arabia. Furthermore, it's not clear why you limit this to just Afrotropical and Indomalayan realms instead of all Eurasia. However, I think you are perfectly justified in changing this to "Wetter conditions could have facilitated mammalian dispersals between Africa and Eurasia". The record is simply not good enough to establish otherwise at the moment.

Line 78: The Saharo-Arabian belt, even when extended to the Thar Desert, has little to do with the boundary of the Indomalayan Realm, which is instead more significantly bounded by the Tibetan Plateau.

Also why is your reference (ref 10) for biogeographic realms Kreft & Jez's 2013 comment on the Holt et al. 2013 Science paper? I think you should either be following Kreft & Jez or Holt et al. 2013. Same for Fig. 1b.

94-96: Actually there is evidence aridity might not have increased in these savanna habitats. A drop in CO₂ is another main hypothesis for the spread of C₄ grasses, which might have nothing to do with desertification. See Blumenthal et al 2017 (also discussed by Faith et al 2018, eg their fig. 2A and 2C).

Also here and in Fig 3 you suggest there was an 'intensification' or an 'abrupt onset' of ice at around 2.6 Ma. You cite Westerhold et al. 2020 but this paper does not appear to state this so dramatically, but rather only confirms that ice sheets were in place by the Plio-Pleistocene boundary. The last 20 years of paleoclimatic data have more or less done away with the idea of a rapid or sudden onset of glaciation around 2.8-2.6, which rather seems to have been replaced by a more gradual model. See Trauth et al. (2021) for a review.

101: Is it really larger than the sandy parts of the Sahara?

106: Levant is Mediterranean. Not 'influenced'

109-110: 'vicariant agent': what is meant by this? If you mean climate changes geographically splitting/isolating species populations and thereby stimulating speciation, then that is quite different from patterns of biogeography and dispersal, and is not something that is followed up in this paper.

Fig2: 2d nicely shows the lack of a terrestrial vertebrate record (note 'fauna' also includes things like marine invertebrates, which is not what you mean here) between Baynunah and Nefud times. 2e and 2f however are problematic. First it is not clear why you chose to focus only on the large mammal record. If you are trying to establish Arabia as an arena of faunal dispersal, this should include a large-scale analysis of a wide range of taxa (what about fish for example, which provide a much better view of hydrographic networks?). Second, almost all the taxa discussed have very poor fossil records. Using a lack of data to speculate that these taxa dispersed during the identified humid periods is not convincing. This is not even to mention the dating uncertainties around many of these records. Vrba may have used this approach in 1995 to establish a hypothetical framework to be tested (the 'traffic-light model'), but this is hardly acceptable today and biogeographic comparisons should be much more quantitative. Additionally, to test your hypothesis that dispersals occurred across Arabia during specific humid periods, you need a much more resolved fossil record, one with an high resolution of first appearance datums for taxa in Africa, Arabia, and Eurasia, and which shows convincingly the chronological origins of taxa in one place, and their steady dispersal elsewhere (e.g. hominins in Africa -> Dmanisi -> Europe and Asia; or equids from North America to Eurasia). Since there is no terrestrial record between ~7 and 0.5 Ma in Arabia, that is simply not possible and the evidence is entirely circumstantial. In fact, determining locations of origin and directions of migration (especially for a genus or species) is exceedingly difficult even when using some of the best fossil records. You don't know if taxa dispersed through the Arabian Peninsula, or only along its edges (eg the Levant). It could also be that during humid periods Arabia was populated by taxa from nearby regions, without acting as a path for continental dispersal. Your review of the fossil record does not distinguish among these scenarios. For example, the fossil fishes of the Baynunah Formation provide evidence for a scenario of repeated colonization of Arabia, suggesting it could also have functioned as a kind of cul-de-sac (Forey & Young 1999; Otero 2022).

Furthermore there is probably nothing to keep fauna from dispersing through the Levant or along coasts and river networks during arid periods. Dispersal between Africa and Eurasia need not be restricted to Saharo-Arabian humid periods. It might have been more interesting to speculate to which degree the proposed arid periods could have acted as barriers to dispersal, or entirely prevented the establishment of faunas in Arabia. Since the Levant-Mesopotamian region is probably more important for the passage of faunas between Africa and Eurasia than the interior of the Arabian Peninsula, an arid Arabia might not have mattered much in many instances. All these general points should be considered in your framework.

There are additionally further problems in the details: The taxa chosen for the biogeographic analysis are largely poorly known and cannot be used to build a story of biogeographic exchange at specific points in time. As mentioned, Vrba did this in the 1980s and 1990s, but that was in a different time and context. For example:

Antilope sub torta: this is actually *Antilope aff. sub torta*. The aff. indicates this is not the same species as *A. sub torta* from the Siwaliks. This is furthermore represented only by two horn fragments from Member C in the Shungura Formation (Gentry 1985), which is around 2.6 Ma. The dashed line labelled 1 in Fig 2 makes it look like this is a well-established species between 2.8 and 2.5 Ma. This is misleading and in fact we know so little about the Shungura *Antilope* that it is not possible to know when and from where its immediate lineage originated, or how long it survived in Africa (is its poor record due to its short species lifespan or the result of rarity and poor preservation?). The same is true for *Makapania broomi*, as we have no idea what other Caprini it is related to, and cannot even guess as to when and from where its lineage entered Africa. It is likely that Caprini were well established in Africa and evolved there, but only rarely made it into the fossil record due to habitat preferences in places that were non-depositional (e.g. mountainous regions). The same is true for duikers (*Cephalophini*) which are widespread in African forests, but have almost no fossil record (same for chimpanzees and gorillas). Continuing down the list, it should not be a surprise that almost all the “Endemic Eurasian fauna in the Afrotropics” are poorly known taxa. *Parantidorcas latifrons*, *Budorcas churcheri*, etc. *Bouria angettyae* and probably *Nitidarcas asfawi*, though both described as a Caprini, are almost certainly Alcelaphini, and I wouldn’t advise hanging too much on these being caprins (Gentry 2010 hinted at this). Also, as already noted in this manuscript, it is just as likely that African fossil Caprini were parts of locally-evolving lineages and not episodic and short-lived arrivals from Eurasia. Fossil Caprini are simply so rare in Africa that it is not possible to sketch out a convincing biogeographic story. Ancient DNA / proteomes may one day address this question.

Also note that Vrba et al. 2015 found that the South Asian Reduncini were monophyletic and assigned to *Sivacobus*. They discuss possible relationships to the African *Kobus/Dorcadoxa aff. porrecticornis*. It’s not a simple matter of the latter taxon being an immigrant from Eurasia into Africa, or as claimed in Fig 2, of *Sivacobus* being an Afrotropical taxon in origin. There is no African *Sivacobus*. Also, even earlier African Reduncini are probably also present at Chorora (Suwa et al. 2015).

On the other hand, other examples / literature are missing. E.g. Boisserie & White (2004) describe an Asian hippopotamid in Ethiopia at about 2.5 Ma. Again here though, the ‘first appearance datum’ for this taxon is basically the only appearance datum there is. Such singleton (or single-interval) taxa are highly problematic and are actually usually left out of large-scale turnover analyses (eg Alroy 2010, or see Bibi & Kiessling’s 2015 discussion of this issue – Vrba’s turnover pulse at 2.8 Ma is basically based on singleton taxa). As such they should probably not be relied on for testing of large-scale biogeographic hypotheses either.

Additional questions continue for other taxa. For example, you list *Serengetilagus* as a Eurasian taxon, but I think you mean Leporidae generally. I believe *Serengetilagus* is not known from outside Africa. Also many of the taxa listed as dispersing from Africa to Eurasia are highly problematic, more so when one considers the lack of good dates for the Tatro/Pinjar faunas.

However, I think a solution is easily found. I do not believe the review and speculation on the sparse large mammal record here adds anything significant to the manuscript. It could be left out entirely. The main finding of speleotherms indicating humid periods between the Miocene and Pleistocene is the main finding. This evidence clearly implies that conditions could have been appropriate for the establishment of faunas like those of Baynunah and Nefud in Arabia at several intervals during the Pliocene and Pleistocene. It’s fair to propose that these humid periods might therefore have played an important role in expanding faunal exchange between Africa and Eurasia, even though this cannot be conclusively demonstrated with the current record (or without a much larger and more complex analysis).

296-308: a summary of the Baynunah fauna concludes that the environment was a wooded savannah there as at Toros Menalla in Chad. It’s not clear what this is supposed to indicate, or why the mention of *Sahelanthropus* here. ‘Wooded savannah’ represents such a diversity of habitats, and is basically a generic label for anything that is not a forest or pure woodland or grassland, and is therefore highly non-specific with regards to most taxa (including hominins).

An additional point. Regarding the cave names (1000 wings; Mossy; Friendly; Murrubeh; Scerake; Luxury; Hotel) – it seems strange that caves in an Arabic-speaking country were given English names (presumably with the exception of Murrubeh). I understand the need for convenient names while conducting fieldwork, but since you are here basically formalizing and fixing these names in print, would you consider giving them Arabic names? Perhaps these could simply be translations of the given English names (and you could retain the English names in brackets).

G. References: appropriate credit to previous work?
Seems fine.

H. Clarity and context: lucidity of abstract/summary, appropriateness of abstract, introduction and conclusions
Yes all fine.

Alroy, J. 2010. Fair sampling of taxonomic richness and unbiased estimation of origination and extinction rates. *Quantitative*

- Methods in Paleobiology. Paleontological Society Papers 16:55–80.
- Bibi, F., and W. Kiessling. 2015. Continuous evolutionary change in Plio-Pleistocene mammals of eastern Africa. *Proceedings of the National Academy of Sciences* 112:10623–10628.
- Blumenthal, S. A., N. E. Levin, F. H. Brown, J.-P. Brugal, K. L. Chritz, J. M. Harris, G. E. Jehle, and T. E. Cerling. 2017. Aridity and hominin environments. *Proceedings of the National Academy of Sciences* 114:7331–7336.
- Boisserie, J.-R., and T. D. White. 2004. A new species of Pliocene Hippopotamidae from the Middle Awash, Ethiopia. *Journal of Vertebrate Paleontology* 24:464–473.
- Faith, J. T., J. Rowan, A. Du, and P. L. Koch. 2018. Plio-Pleistocene decline of African megaherbivores: No evidence for ancient hominin impacts. *Science* 362:938–941.
- Forey, P. L., and S. V. T. Young. 1999. Late Miocene fishes of the Emirate of Abu Dhabi, United Arab Emirates; pp. 120–135 in P. J. Whybrow and A. P. Hill (eds.), *Fossil Vertebrates of Arabia, with Emphasis on the Late Miocene Faunas, Geology, and Palaeoenvironments of the Emirate of Abu Dhabi, United Arab Emirates*. Yale University Press, New Haven.
- Gentry, A. W. 2010. Bovidae; pp. 747–803 in L. Werdelin and W. J. Sanders (eds.), *Cenozoic Mammals of Africa*. University of California Press, Berkeley.
- Gentry, A. W. 1985. The Bovidae of the Omo Group deposits, Ethiopia (French and American collections); pp. 119–191 in Y. Coppens and F. C. Howell (eds.), *Les faunes Plio-Pléistocènes de la basse Vallée de l'Omo (Ethiopie); I: Perissodactyles-Artiodactyles (Bovidae)*. Cahiers de Paléontologie. CNRS, Paris.
- Holt, B. G., J.-P. Lessard, M. K. Borregaard, S. A. Fritz, M. B. Araújo, D. Dimitrov, P.-H. Fabre, C. H. Graham, G. R. Graves, K. A. Jönsson, D. Nogués-Bravo, Z. Wang, R. J. Whittaker, J. Fjeldså, and C. Rahbek. 2013. An update of Wallace's zoogeographic regions of the world. *Science* 339:74–78.
- Kreft, H., and W. Jetz. 2010. A framework for delineating biogeographical regions based on species distributions. *Journal of Biogeography* 37:2029–2053.
- Otero, O. 2022. Fishes from the Baynunah Formation; pp. 79–109 in F. Bibi, B. Kraatz, M. J. Beech, and A. Hill (eds.), *Sands of Time: Ancient Life in the Late Miocene of Abu Dhabi, United Arab Emirates*. Vertebrate Paleobiology and Paleoanthropology Springer International Publishing, Cham.
- Suwa, G., Y. Beyene, H. Nakaya, R. L. Bernor, J.-R. Boisserie, F. Bibi, S. H. Ambrose, K. Sano, S. Katoh, and B. Asfaw. 2015. Newly discovered cercopithecoid, equid and other mammalian fossils from the Chorora Formation, Ethiopia. *Anthropological Science* 123:19–39.
- Trauth, M. H., A. Asrat, N. Berner, F. Bibi, V. Foerster, M. Grove, S. Kaboth-Bahr, M. A. Maslin, M. Mudelsee, and F. Schäbitz. 2021. Northern Hemisphere Glaciation, African climate and human evolution. *Quaternary Science Reviews* 268:107095.
- Vrba, E. S., F. Bibi, and A. G. Costa. 2015. First Asian record of a late Pleistocene reduncine (*Artiodactyla*, Bovidae, Reduncini), *Sivacobus sankaliai*, sp. nov., from Gopnath (Miliolite Formation) Gujarat, India, and a revision of the Asian genus *Sivacobus* Pilgrim, 1939. *Journal of Vertebrate Paleontology* 35:e943399.
- Vrba, E. S. 1995. The fossil record of African antelopes (*Mammalia*, Bovidae) in relation to human evolution and paleoclimate; pp. 385–424 in E. S. Vrba, G. H. Denton, T. C. Partridge, and L. H. Burckle (eds.), *Paleoclimate and Evolution, with Emphasis on Human Origins*. Yale Univ. Press, New Haven.

Referee #4

(Remarks to the Author)

Summary of the Key results

In this manuscript, authors did generate a new hydroclimate record covering the last 8 million years from central Arabia region using speleothems record from 7 caves. To do so they combined U-Pb / U-Th dating with speleothem morphology/mineralogy analysis to inform on chronology of wet episodes. They then combine this with $\delta^{18}\text{O}$ of calcite to discuss general trend in rainfall activity over the studied period. Authors also used isotopic composition of fluid inclusions ($\delta^{18}\text{O}$ and $\delta^2\text{HFI}$) from the speleothems compared with those of various moisture sources in the region in order to track potential change in the geographic origin of moisture over the studied period of time. Based on this methodology they identified various wet periods over the late 8 Ma and compare the obtained chronology with the evolution of the meridional gradient of Sea Surface Temperature they derived from paleoceanographic records from high latitude and within the tropics. They used this to suggest that hydroclimate dynamics in the studied region is driven by periodic warming of northern hemisphere high latitudes and shrinkage of northern ice-sheets. Authors then related the identified wet periods (in an area that is at present-day hyperarid) to biogeographic record, and suggest that periodic environmental changes in this region at the crossroad of Eurasia and Africa have force dispersion dynamics of Fauna between the two subregions.

Originality and Significance

As a non-specialist of U-Th and U-Pb dating methods or speleothems I can only comment on the climate aspect of the paper. The new dataset provided in the study, is to my opinion a very important piece on the environmental evolution of a crucial region for both mammals & hominids dispersal but also important for the understanding of paleo-climate and climate dynamics of the Saharo-Arabian desert. This especially as data from this part of the world are very few when it comes to late Miocene-Pliocene records. If combined properly with recent records documenting the evolution of West Sahara region and Mesopotamia (eg. Crocker et al., 2022 ; Böhme et al., 2021 for the most recent ones), that roughly cover the same period of time, the new record could enable to draw a full picture of the large scale dynamics of the Saharo-Arabian region over time. Picturing the evolution of climate in this region and understanding forcing mechanisms and feedbacks is of major importance to better figuring out how hyper-arid regions will likely response to future climate change. The record presented here should therefore be of interest for various communities as it covers a period of time that is key for the understanding of hominids dispersal and climate dynamics in a world warmer than present day. Based on paleoclimate/climate dynamics of such record I would recommend considering this paper for publication after major improvement of the discussion.

Conclusions & Suggest improvements

The paper is well written/very clear, including the methodology (even for a non specialist) and figures are relevant, easy to read and to understand, which is a good point. General trend in hydro climate authors interpret from $\delta^{18}\text{O}_{\text{Carb}}$ in the speleothem are in line with progressive drying of the climate and shift in environment recorded from the late Miocene onward, alongside progressive cooling. Identified wet period seems more or less coherent with those identified further north by Böhme et al., 2021.

Few things though required clarification from the authors to my opinion :

- I note that it seems to be an incoherence between the 5 - 3 Ma wet period identified on Fig 2. and description in the text stating that the 6 - 3.5 Ma period is dry and suggest expansion of the arid belt, so it fits the Biogeographic exchange record. I should be better discuss also with regard to the timing of aridity expansion interpreted by Böhme et al., 2021 so it avoid confusing the reader.

- Despite the main finding being interesting and important I regret that the content in the main text is a bit light when it comes to climate mechanisms or biogeography. I had to get in the Supplementary information too many times to find informations of interest. There are for exemple a lot of crucial reasoning on how the authors related Meridional Gradient of Temperature and Humid periods they identified in the part 2.2 of the Supplementary information, that could be useful in the main text. Same thing is true for the part 2.3 of the SI that provide important informations about the relation between biogeography and climate. I think the manuscript could be largely improve by using part of the information in those two part of the SI to be clearer on the reasoning while relating timing of wet period identified in the speleothem records, climate dynamics and dispersal.

- The authors put all the focus on Meridional Temperature Gradient and ice-sheet forcing to explain the wet periods, while barely invoking insolation change as a major driver of hydroclimate. I agree that change in SST gradient should have an impact on ITCZ dynamics, as it controls the global atmosphere dynamics and expansion of Hadley cells as already largely discussed by authors. But previous work on northern African region also often relate oscillations in climate/environment to orbital forcing (such as this is the case during the well documented Green Sahara period of the Holocene/Pleistocene). I encourage authors to use available modeling work describing orbital effect on Northern African monsoon (see Kutzbach et al., 1981 for historic work, but more recent papers including Bosmans et al., 2015 for example are also insightful). While I am not aware of work focusing directly on Arabian region (but might have missed some), most of those studies display regional maps where Arabian region is represented. Most of those papers display simulations that modify the moisture flux and precipitation dynamics without modifying the extension of northern ice-sheet suggesting that change in regional Sea Surface Temperature are also somehow responsible for driving the signal of precipitation. This should be better discuss in the paper.

Another thing that is not discussed when it comes to explaining the global trend toward more aridity from the late Miocene to present-day is the potential effect of changes in geography. Zhang et al. 2014 for example do suggest that change in Paratethys extent or Iran plateau elevation (or rather the Iran-Anatolia topography) might have been responsible to aridification of the Sahara desert. And this seems that both of those geography features have been evolving during the late Miocene-Pliocene, though there direct effect of Arabian climate have not been extensively studied. I understand that the focus of this paper is more on identifying periodic humid periods and related it to biogeography, but this should at least be mentioned somewhere.

Minor comments :

- Please try to be consistent in the use of units : mm.a-1 vs. mm/a
- Fig 3c - Indonesian Throughflow instead of Indonesian followthrough

Zhang, Z. et al. Aridification of the Sahara desert caused by Tethys Sea shrinkage during the Late Miocene. *Nature* 513, 401-404 (2014). <https://doi.org:10.1038/nature13705>

Böhme, M. et al. Neogene hyperaridity in Arabia drove the directions of mammalian dispersal between Africa and Eurasia. *Communications Earth & Environment* 2, 85 (2021). <https://doi.org:10.1038/s43247-021-00158-y>

E. Kutzbach, Monsoon climate of the early Holocene: Climate experiment with the Earth's orbital parameters for 9000 years ago. *Science* 214, 59-61 (1981).

Bosmans et al., 2015. Response of the North African summer monsoon to precession and obliquity forcings in the EC-Earth GCM, *Climate dynamics*, 44, 279-297 (2015)

Version 1:

Reviewer comments:

Referee #1

(Remarks to the Author)

Thank you for your detailed responses to my queries - I believe you have fully addressed all of my concerns. Well done on an excellent piece of scientific research!

Referee #2

(Remarks to the Author)

I thank the authors for addressing all my comments, I realise this was not a straightforward task and I appreciate their thorough responses. In my opinion, the authors have revised their manuscript satisfactorily:

- By reassessing their U-Pb data and by adding more [234U/238U] measurements, the authors have removed my concerns about the initial [234U/238U] disequilibrium correction and about a previously apparent trendline of the initial [234U/238U] with age. From the newly added [234U/238U] measurements it is now indeed evident that there is no significant trend in the initial [234U/238U] through time. I appreciate that they use speleothem-specific, back-calculated initial [234U/238U] values for (a now increased set of) samples younger than 1.5 Ma and now also support the authors' choice to use an average for samples beyond that. I realise this was a major undertaking, but I hope the authors understand that understanding variation in the initial [234U/238U] is paramount for the precision and accuracy of speleothem U-Pb data and, ultimately, its interpretation. I appreciate that they authors recalculated their ages using the spine fit type and agree that the ages are now more robust. Through this revision, I believe the data has high fidelity and now has my full support. I also appreciate that the authors have gone the extra mile by collating a database of the average [234U/238U] of speleothems and data of the activity ratios in modern aquifers in the greater Arabian Peninsula region – this gives further confidence (and also provides interesting insights into the region's initial [234U/238U] signal).

- I appreciate that the authors have increased their U-Pb age data set. It is reassuring that the new ages fall within the previously identified humid phases. As the authors acknowledge, the sample size is still too small to identify discrete humid phases, and I appreciate it that the authors have removed references to the CAHP groupings. I believe the data are now more accurately presented and discussed.

- That brings me to my last point: by shifting the emphasise from humid phases being closely linked to faunal dispersals to the palaeoclimatic story, I believe this manuscript now has a much stronger narrative, and, in my opinion, the revised version is now also more strongly supported by the underlying chronology.

In summary, all points, and particularly my main concerns (the U-Pb age calculations, particularly the initial [234U/238U] disequilibrium correction and fit type for the U-Pb isochrons, sample size and preservation bias), have been satisfactorily addressed by the authors and are no longer concerns to me. I congratulate them on an impressive piece of work, and I now recommend this manuscript for publication.

Just a few minor comments/questions:

- The authors use an average initial 234U/238U activity ratio for samples >1.5 Ma. This is just something for the authors to consider whether an uncertainty-weighted average would be more appropriate.

- How was the 2 σ uncertainty (0.178) on the average initial 234U/238U activity ratio calculated?

- Figure 3f: what does the colour of the filled data points correspond to?

- Lines 276, 461 and 470 (in marked up version): probably the authors mean speleothem formation instead of speleogenesis (= cave formation, Fairchild & Baker 2012)?

- Lines 66 (in marked up version): I'd suggest changing the wording here and use "speleothem ages" (or similar) instead of "dates"

- Lines 1113-1116: just to be specific, probably better to explicitly say that "234U/238U" is an activity ratio. Also, I suggest adding to this sentence that samples younger than 1.5 Ma were calculated with speleothem-specific initial [234U/238U] values obtained from the solution U-Th measurements, and that only the samples older than 1.5 Ma have been corrected for initial disequilibrium using an average initial [234U/238U]

Referee #3

(Remarks to the Author)

The manuscript is much improved, and I find the expanded presentation of the speleothem data even more compelling. The authors addressed my previous comments in full in their revision (down to presenting cave names in Arabic, which I appreciate). The faunal biogeographic discussion has been appropriately shortened. I still have some concerns about the presentation of the faunal context, but these are minor at this point. Comments follow:

Line numbers refer to the tracked changes pdf.

169: the reader expects Fig 1 to show biogeography, but instead it shows rainfall

170: are refs 3 and 14 really the right ones here? It seems you are discussing modern biogeography, and for that a reference like Holt et al 2013 might be the right one (or Kraft & Jetz's comment, even though I still think the original paper should be cited).

FIG 1. B and C are hard to understand in terms of percentages. Would you consider converting these to mm/yr amounts?

211: Peninsula with capital P?

243: '8Ma BP' - BP is not necessary.

244-245: ≥ 100 mm/yr puts it in arid climate, not hyperarid (< 100). Or borderline, but not "one of the most hyperarid parts of modern central Arabia". From Fig 1 it's even a relatively humid part of the Peninsula considering most of it appears to receive 0 mm.

Fig 2. Could you place the panel titles a, b, c, etc on the left? It's a bit confusing on the right.

Also the apparent association of the data from d with the y-axis from c is confusing. I'd suggest more separation between panels vertically.

Are the dashed lines needed in b? The grey bars should be sufficient. Also check some of your hominin age ranges: Sahelanthropus is proposed to be ~ 7 . Ardipithecus is 4.4 to 5.6 Ma. Earliest Australopithecus is anamensis, around 4.1 Ma – why the dashed line going earlier? Also how is 'Early Homo' defined? If anything prior to Homo sapiens, then the bar should extend right to 300 ka or later.

Is the hominin timeline even relevant for this article? This is described as 'major junctions' in the caption, but it's basically a genus age range plot. Is there any correlation between the appearance/disappearance of hominin genera and the paleoclimate proxies shown? I think not.

Why the 'Old World Savanna' bubble at 6-8 Ma with an arrow pointing earlier? Bear in mind the Old World Savanna Paleobiome is alive and well in places like sub-Saharan Africa and South Asia today, it was just more extensive in the past. I assume you are highlighting the peak of the OWSP (Late Miocene to Early/Middle Pliocene) but that is not evident. Not sure this is relevant to this figure either.

Panel f is missing a callout in the caption. Also, please state what the whiskers extending from the grey circles represent – the response to reviewers states these are 95% confidence intervals. That should be clearly stated in the caption.

"One age with an uncertainty greater than > 1.5 Ma is not shown" - but one of your data points at ~ 4 Ma has an uncertainty > 2 Myr. Why is that one shown and the other left out? I would suggest not leaving any data out.

337-338: ~ 7.3 Ma until ~ 6.4 Ma... ~ 4.1 Ma until ~ 3.2 Ma – where are these numbers from? I suggest you do this quantitatively, by binning together all the ages including uncertainties of each humid interval, and reporting the 95% age ranges.

457: "centred around 2.1 and 1.0 Ma" - same here, provide a more quantitative value, like an average \pm uncertainty.

636 "temperature gradients shifted to approximately modern-day values" - I understand from the reference cited that this was a temporary dip in temperature from 7-5.4 Ma. You might want to be clearer about this as a 'temporary shift' or 'ephemeral dip', rather than a 'shift'.

713: "Biogeographical significance" - I still find this section too speculative and not contributing much to the main findings of the article. Comments follow.

729-735: the guenon, or primates in general, are negligible for habitat reconstruction in these assemblages. You already mentioned the hippos, giraffes, crocodiles, etc which indicate the presence of permanent water and grassland-woodland vegetation (ie 'savannas'). There is nothing that ties the Baynunah specifically to the Chadian sites, except that they are of similar age and had similar large mammals. The same could be said of any number of Late Miocene sites from China to South Africa. Certainly the guenon is not a point of similarity.

734: "enough precipitation to support vegetation suitable for hosting primates" - here too the emphasis on the primates is misplaced. Hippos, crocodiles, river turtles, and fish need a lot more water, and are present in far higher abundances. If anything, monkeys are generally more indicative of trees than water. Maybe you mean "suitable for hosting a high diversity of large mammals" or similar.

741-774: Quite right. Any argument for "a sustained arid barrier in northern Arabia (33° N) between 5.6 and 3.3 Ma" is using an absence of evidence as evidence of absence. There are simply no terrestrial deposits of that age in which to find fossils on the Peninsula (as far as we know) and that probably has more to do with regional tectonics (uplift / erosion / lack of deposition) than any kind of evidence for large mammals not being present. Your speleothem record is spectacular precisely for this reason.

780-781: reduction, rather than a breakdown. Also not just in Africa, but South Asia as well. And I think you mean genera Homo and Australopithecus, not 'early hominins' as those were also around in the Late Miocene. Though here too I fail to see the significance of the hominins to the argument.

782: "progressively more difficult to traverse and settle over time due to the gradual aridity trend" - a bit too speculative. More difficult for whom? Crocodiles maybe. But not oryx. And humans/hominins also had no problem occupying and presumably traversing the Peninsula at different times. Your data very nicely document a trend of increasing aridity, but what is the relationship between that trend and the ability of mammals to traverse the region? Aridity thresholds will be significantly different for different species, and can be fully modulated by seasonal waterways and lakes, especially on geological timescales. Bear in mind also that the Peninsula was probably always traversable along coastal / mountain routes, regardless of how dry the interior was.

783-789: I find these two sentences contradictory. The first states rightly that mammals were present in the dry Mid-Late Pleistocene, and the second still tries to claim that the new climatic record identifies 'constraints' on range expansions. It's

true the speleothem record is “providing insight into critical periods for which previously there was no data” but this is being conflated with data on fauna and biogeography, for which there is none. If anything, the sparse Arabian fossil record confirms that large mammals thrived there anytime we have the right deposits to find them in (Oligocene, Middle Miocene, Late Miocene, Mid-Late Pleistocene). As noted above, their absence is likely the result of geological preservation bias, not true absence.

Referee #4

(Remarks to the Author)

After getting across the manuscript and the rebuttal letter, it seems to me that the authors managed to address most of the criticisms from the reviewers regarding the analysis and statistical significance of the dataset. They did so by either doing suggested analysis or by reformulating the manuscript. I would however let a reviewer whose interpreting U/Th and U/Pb data is the specialty to comment on this aspect of the paper. Authors also removed most of the paragraph on biogeography while keeping the synthesis they performed in the Supplementary Material and they instead increased the climate implication of their dataset. Biogeography is now more an opening for the paper which I think is good due to strong limitations of biogeography records from the region pointed out by one of the reviewers.

One of the concerns raised was also that the ‘low’ number of samples would prevent identification of significantly humid periods. I do not think this should prevent the manuscript to be considered for publication as my recollection is that the size of the dataset is still impressive for this type of study in a place where barely any other records exist for the time period. Because of the latter, the results clearly have a strong importance for the understanding of climate evolution and climate dynamics of the entire northern Africa/Arabia region. In particular even with large uncertainty in the ages, the trends they identify in the $\delta^{18}O$ and δ^2H are by themselves interesting results, no matter the interpretation the authors have of them. Those could indeed motivate better integration of regional scale data by the paleo-climate community to gain insight into the regional Mio-Pliocene evolution of the ITCZ.

The effort the authors made in this version of the manuscript not to oversell too much the data should be acknowledged. They for example now better underline there are still gaps in the understanding of the regional climate and limitations to the dataset but still managed to draw an interesting story out of it. In particular, the authors reconsider the way to refer to the humid intervals they described by replacing the formal naming they initially gave (CAHPs) for ‘humid periods’ which emphasizes to my opinion the uncertainty in the duration or frequency of such events.

I however think that now that the authors shifted the discussion more toward climate implications of their results they need to be more precise in the description of the climate system processes before the paper could be published (see my comments below).

Comments :

L. 54. In the abstract the authors state that ‘Recent research places the emergence of this barrier at 11 million years (Ma) ago’, referring to Crocker et al. (2022) paper. This statement is incorrect: Crocker et al. published a record that started at 11Ma as the title of the paper states it, therefore the record does not provide any indication on the regional environment prior to 11 Ma. A more exact formulation would be that ‘Recent research suggests this barrier has been in place since at least 11 millions years ago’

L. 91. The authors now state that “Widespread aridification instigating C4 grass dominance in low and mid-latitude open environments has traditionally been linked to Quaternary cooling in the Northern Hemisphere (NH), triggered by the onset of glacial-interglacial cycles, which intensified at ~2.6 Ma.”. I might have misunderstood but I find this sentence confusing as the literature seems to place this transition way before the Quaternary, already in the late Miocene (see the Ceerling et al. (1997) and Herbert et al. (2016) reference authors cite in one of the previous sentence, though they have been new data published regarding this transition since then (see Tauxe and Feakins (2020) for review)). I therefore suggest the authors rephrase this part of the paragraph so it does not sound in opposition to the cited literature.

L. 94-95. ‘with fully arid conditions in the Sahara from 11 Ma’ : in line with the previous comment, I suggest the author to rephrase as ‘with fully arid conditions in the Sahara from at least 11 Ma’

Figure 1 : Could the authors specify which months are included as ‘Autumn-Winter’ and ‘Spring-Summer’ respectively ? This is because most of the time the definitions of seasons are not following rules in paleo-climate papers at least.

In addition, I strongly encourage the authors to add the main directions of the present day regional moisture transport they described lines 126-128 on Figure 1b and 1c. This is because this information is central for their discussion and having it display graphically would help the reader to understand.

L. 124-128. Because the authors decided to shift the paper discussion more toward climate significance of their findings, they need to be more precise in describing the moisture transport pathway in the region. For example, the text now reads “between westerlies to the north and east and the monsoon to the south and west receiving”; and while this is easy for the reader to understand where the winds comes from in the north and east part of the region, the reference to monsoon is a bit less straightforward and it would be more clear to provide additional information on the moisture provenance. I understand that there are character limitations, but there are some parts of the discussion that could be better organised to improve the fluidity of the manuscript and save space, as I mentioned in some of my comments below. I noticed that part of this

necessary piece of information is in the Supplementary Material but my opinion is that it belongs to the main text as soon as an emphasis is put on the climate implication of the records.

One might consider it not highly relevant from present day climate perspective but because the region the authors study is located at the border of 2 precipitation systems (Figure 1b-c), one might also imagine that northward or southward shifts of the winds belts might make it more sensitive either to westerlies or to winds coming from the Indian Ocean, modifying regional moisture and thereby precipitation dynamics. This is what the authors tentatively describe in the discussion of the paper (lines 235-235) but fail to integrate in the large-scale atmosphere dynamics that is brought out in a later paragraph (from line 247).

L. 154. paedogenic → pedogenic

Figure 2e) Please indicate the meaning of error bars also in the figure caption.

Figure 3. Arabian Sea SST especially from the Miocene are derived from Uk'37, with temperatures for the late Miocene that are close to the saturation limit of the proxy. Therefore, the gradient that is derived is likely to be under-estimated for the pre-late Miocene cooling part of the record. I think it would be good to acknowledge it somewhere, maybe in the SI, although it won't change anything in the interpretations as the cooling toward present day and increasing gradient is not a matter of debate.

L. 223. I suggest the authors find a different title to the paragraph so it relates better to the content. Indeed, they here mostly discuss sources of moisture rather than precipitation dynamics.

L. 232-233. The authors now state that 'The modern meteoritic local waterline is dominated by low precipitation amounts (Fig. 1a) from a mixture of summer and winter sources (Fig. 1b,c)'. The last part of the sentence is unclear. The authors should there specify what they mean by 'summer' and 'winter' sources respectively, especially because they suggest moisture source is changing over seasons. As mentioned in one of my previous comments, this would be easier to emphasize if the large-scale moisture pathway is identified on the figures 1b and c.

L. 237. - Fig. 4 → Fig 4b, so the reader knows directly where to look.

L. 237-238. "These data plot on a modern meteoric waterline derived from a tropical moisture source (Indian Ocean) (Fig. 4b)". It seems to me that tropical moisture source (Indian Ocean) is a reference to S-LMWL (Oman) in the figure. It would then be easier to read the figure and understand what the authors are explaining if they were also referring to that abbreviation directly in the text.

L. 238-239. Now the authors state "Our data reveal that the tropical rain-belt repeatedly reached our study area currently at 26° N, bringing precipitation of African/Indian origin". Africa and Indian regions seem to be two opposite directions relative to the site location. It is therefore not super clear what the authors mean here.

L. 235-245. My understanding is that a lot of different processes are usually involved in explaining shifts in water isotopes including but not limited to change in precipitation regimes and moisture source. Change in temperature can also affect the isotopic signature especially when comparing warm periods like the late Miocene and the Pleistocene or change in the isotopic composition of the source can also play a role without requiring change in source by itself. I think the argument would be stronger if the authors first expand a bit on all plausible explanations for isotopes shift they see in the record and explain why they retain the moisture source shift. Those hypothesis would be in addition probably easy to test with already published water-isotope enable simulations (eg. Knapp et al. 2022, <https://zenodo.org/records/6953979> as an example for the Pliocene).

In addition, this paragraph is appearing very confusing when reading the 'Meridional temperature gradient' paragraph (l.247) just after. Here, different climate phenomena are mixed here without much explanations and lack at this point of the text integration into the large scale atmosphere dynamics so the reader understands clearly the reasoning. For example authors first aligns their data points with different water meteoric lines and late Pleistocene record from further south to discuss about moisture source but then mentioned shift in precipitation regime for the Pliocene and Miocene, which are two different processes but that might probably both be related to Hadley circulation and ITCZ. I think the authors have a plausible explanation that relates the isotopes signal, regional atmosphere dynamics and global climate but they fail at properly accounting for it because the way the two paragraphs are structured.

The text would be improved by reformulating the paragraph on 'Reduced tropical precipitation' and better integrating it with the statements from the 'Meridional temperature gradient' part just below. This would enable to reduce text length and provide a more consistent picture integrating between large scale patterns and regional climate, which would help to better explain the isotopic data. For example mentioning the shift in ITCZ in Pliocene climate simulations (lines 253-255) straight after mentioning the very low $\delta^{18}O_{FI}$ and δ^2H_{FI} (lines 240-242) and Pleistocene contraction of Hadley cells following Pleistocene FI data (lines 243-245) would help.

L. 247. and following. I wonder why the authors do not make a better of other Arabian records (Böhme et al. (2021) or others) that they cite in the introduction of the paper and in the biogeography paragraph and that cover more or less the same period. Because those records are located in different types of regional climate zones at present day combining both could serve as an argument to infer change in the regional climate and link this to shift in large scale atmospheric patterns.

I note that the information provided in the paragraph that has been added l. 263-274 improved the manuscript.

L. 286-291. The paper cited here to justify potential impact of Mediterranean Sea and red sea shrinkage during the Messinian crisis here, Zhang et al. (2014), does not test a scenario for desiccation of the Mediterranean sea. I would rather refer to Ivanovic et al. (2014) that actually tested the hypothesis of Mediterranean Sea shrinkage on the climate, though it seems that the mean annual precipitation change they simulated over Arabia was insignificant.

In addition, and regarding Zhang et al. (2014) paper result, I would consider another possible (or complementary) explanation for the supposed late Miocene aridity in the region. There has been large variations in water body in the Paratethys region (eastern Europe) during the interval (eg. Krijgsman et al. 2010) that could have altered the dynamics of ITCZ in the same manner Zhang et al. (2014) described for the long term transition from early Miocene to present-day. I have not reviewed the literature on the topic but timing of such events seems to fit and I therefore do encourage the authors to look for appropriate literature on this.

L. 291. Mentioning 'Tethys shrinkage' is an incorrect (or incomplete) formulation even though Zhang et al. (2014) paper is making extensive reference to this. Zhang et al. (2014) actually accounted for two significant regional events in their geography 1) the closure of the Tethyan seaway between Mediterranean Sea and Indian Ocean - that it now mostly thought to have happen before and during the mid-Miocene though (see for example Bialik et al, 2019), and 2) the shrinkage of the Paratethys that is a more complicated event as there have been persistent lakes with episodes of large scale flooding event during the late Miocene/ early Pliocene (see for example Krijgsman et al. (2010) or Lazarev et al., (2021) among others). More exact formulation would therefore be 'the Paratethys shrinkage' or to a greater extent 'the Tethys Sea shrinkage'.

L. 291-293. Because the authors states that the mid-Pliocene is likely to have been more humid and this sentence conclude a paragraph on the late Miocene cooling period I would rephrase slightly the last part of the sentence so it does not sound that this shift what permanent, which contradict the finding of the paper.

L. 295. Humidity → humidity

L. 296 and following. While I think the addition of this section is valuable and relevant, I also think it could still be improved. In particular here the authors only quickly referred to effect of orbital changes on monsoon and totally neglect to refer to the link with possible moisture changes / detailed atmosphere dynamics that would be highly relevant for their study (eg. Lines 304-305 '...interglacial suggest a substantial increase in precipitation in the Arabian Peninsula in the order of 305 300 to 600 mm a-1, originating mainly from the North African monsoon'.)

I think being more precise and making use of the few papers that previously published modeling results centered on the region (even if it's only for the most recent interglacial period) would help to better integrate this part to the paper and probably make an easy link with the 'Reduced tropical precipitation' section. As an example I would refer to the Nicholson et al., (2020) paper on the last 1.1 Ma period in which they really well integrated the understanding of moisture transport and regional climate dynamics at orbital time scale to try explaining the records they have in the region.

L. 308. "Climate modelling of the Tethys suggests" → "Climate simulation investigating the effect of paleogeography changes in the Paratethys region suggest" or "Climate simulation investigating the effect of paleogeography changes in the Tethys Sea region suggest" would be a more correct formulation.

L. 308-309. "a high sensitivity of the African monsoon to orbital forcing began after the Tethys closure by the Late Miocene". The reference to Tethys closure is confusing as I mentioned above but directly link to what is written in Zhang et al. (2014). The main problem with Zhang et al. paper is that reader would understand from it that Paratethys disappeared during the late Miocene which is more or less correct as there was still megalakes in the region with highly variable size during the Mio-Pliocene (eg. Krijgsman et al. 2010) or that the Tethyan seaway did close during the late Miocene which is not true if one believe recent literature (eg. Bialik et al., 2019). I think this reference is useful when it comes to understanding the effect of paleogeography changes on climate because it uses paleogeography sensitivity tests but becomes less relevant when it comes to timing of such events when considering more recent research on this topic. More correct sentence would therefore only state that 'those simulations show increased sensitivity of orbital forcing owing to changes in the land-sea mask in the region.'

L. 311-312. "may potentially be driven by orbital insolation cycles which are more result in more effective precipitation in a 'warmer' versus 'ice-age' world.". This part of the sentence read weird and is therefore difficult to understand.

Figure 4a. The red-filled circle described as central Arabia in the legend is confusing. I assumed after closer look to the figure it refers to the colored circle colors by age (from blue to red). I suggest the authors to rather use a blank-filled circle to avoid confusion and to move the lave just above the age color scale so it's easier to read. The size of the age color scale in the figure could be reduced to find space.

In addition, the symbols for the Northerly and Southerly source regions are very difficult to see on this figure.

L 371-372. Because the authors decided to remove the biogeography focus of the paper I am not sure this conclusion holds anymore and I suggest changing it.

Ref 53 is incomplete and it seems there is an issue with reference numbering between Main text and Methods part.

S.Fig 9. I suggest the authors to at least add the coordinates in the caption for the oceanic sites that are used to derive gradients so one can directly understand what those gradients represent. Adding a small map with site location would be a better alternative though.

References

Crocker, A. J. et al. Astronomically controlled aridity in the Sahara since at least 11 380 million years ago. *Nature Geoscience* 15, 671-676 (2022).

Tauxe, L., and S. J. Feakins. "A reassessment of the chronostratigraphy of late Miocene C3–C4 transitions." *Paleoceanography and Paleoclimatology* 35.7 (2020): e2020PA003857.

Zhang, Z. et al. Aridification of the Sahara desert caused by Tethys Sea shrinkage 547 during the Late Miocene. *Nature* 513, 401-404 (2014).

Ivanovic, Ruza F., et al. "Modelling global-scale climate impacts of the late Miocene Messinian Salinity Crisis." *Climate of the Past* 10.2 (2014): 607-622.

Krijgsman, W., et al. "Rise and fall of the Paratethys Sea during the Messinian Salinity Crisis." *Earth and Planetary Science Letters* 290.1-2 (2010): 183-191.

Lazarev, S., et al. "Five-fold expansion of the Caspian Sea in the late Pliocene: New and revised magnetostratigraphic and 40Ar/39Ar age constraints on the Akchagylian Stage." *Global and Planetary Change* 206 (2021): 103624.

Bialik, Or M., et al. "Two-step closure of the Miocene Indian Ocean Gateway to the Mediterranean." *Scientific reports* 9.1 (2019): 8842.

Nicholson, Samuel L., et al. "Pluvial periods in Southern Arabia over the last 1.1 million-years." *Quaternary Science Reviews* 229 (2020): 106112.

Version 2:

Reviewer comments:

Referee #3

(Remarks to the Author)

I thank the authors for addressing all my comments. I have no further concerns.

Minor points:

I would just suggest checking the hominin genera age ranges in Fig 2 one more time. Make sure you distinguish between the age range of actual occurrences of the taxon, and uncertainty in dating. For example, I believe *Sahelanthropus* is known from ~3 specimens that come from a single site, and possibly a single geological unit. Whether it's Vignaud et al.'s estimate of 7-6 or Lebatard et al.'s estimate of 6.8-7.2 (largely considered unreliable, the technique has found no further applications on similar sediments), these are the age uncertainties of the assemblage, not the age range of the assemblage. 6.8-7.2 would better be represented as a point at 7 with uncertainty extending 0.2 Ma in each direction. I believe the same applies to *Orrorin* and possibly others. (Bear in mind there are proposals that *Sahelanthropus*, *Orrorin*, and *Ardipithecus* are all the same genus. I think for many people the jury is still out on that one.)

Lines 110-114: Ages of samples should include \pm uncertainty. Ages of sample clusters would more clearly be written as median \pm uncertainty, rather than ranges.

Referee #4

(Remarks to the Author)

I first want to apologize for the delay replying. The authors have done a tremendous work with this paper and think really improved the significance of the manuscript over the revision process. The records they provide here really fill a gap in the knowledge of past hydroclimate and I hope will trigger new research to better understand Miocene hydroclimate change. I believe the authors addressed all of my concerns and think the climatic interpretation of the data is much more stronger now.

Minor comments :

L 181 : 'However, show a gradual trend towards higher values (Fig. 3g).' → it seems to me that the sentence is uncomplete.

L. 202 : 'A steady decline of the poleward (northward) extent' → This sounds a bit weird to refer to change in geographic position by 'decline'.

L. 232 : 'Regional evidence for terminal Miocene aridification can be drawn from the Messinian Salinity Crisis interval when palaeoclimate data from both the Mediterranean and Red seas document significant desiccation between 6.0 and 5.3 Ma' —> I suggest the authors to slightly rephrase as right now it read as the desiccation is caused by aridity while it's most likely the other way around. I think the next sentence provide sufficient information, providing the author include the 'MSC' into it.

L.265 : 'Our data, which highlights in situ climatic sensitivity' —> I suggest the authors to change the term 'climate sensitivity' by 'hydroclimate sensitivity'. Main reason for this is that 'climate sensitivity' commonly refer (at least in climate community) to the degree of warming per degree of CO₂, which is not what the authors refer to here. This might be confusing.

L. And 327 and 330 - Figure 3 caption : "differences between Arabian Sea core 722 and North Atlantic core 982' Sahara dust flux from North Atlantic marine core 659'. I believe the convention to name the drilling site is to use the drilling program the core have been retrieved through. In that case probably 'site ODP 722' instead of 'marine core 722' and so on

L. 757-759 / 777-778 . See comment above. Convention is to give the drilling program in addition to site number.

L. 329 - 'Indonesian flowthrough' —> 'Indonesian throughflow'

L 760 - reflect —> reflects

Authors response to reviewer comments:

“Recurrent humid phases in Arabia over the past 8 million years”

We firstly thank the editor and the four reviewers for taking the time to provide constructive and insightful feedback and giving us practical suggestions to improve this manuscript. We are now pleased to say that the major issues raised by the reviewers, particularly regarding the U-Pb age calculation and the $^{234}\text{U}/^{238}\text{U}$ correction have been addressed in the revised MS, including through the generation of new data, and a more robust estimation of the uncertainties. A brief overview on the major points addressed in the revised manuscript are as follows:

(1) Analyses of additional U-Pb ages and additional speleothems, additional $^{234}\text{U}/^{238}\text{U}$ measurements and calculation and a re-evaluation of all the ages in our dataset to include a more robust estimation of the initial $^{234}\text{U}/^{238}\text{U}$ disequilibrium. Firstly, we expanded the number of $^{234}\text{U}/^{238}\text{U}$ calculations to include our younger (<1.5 Ma) U-Pb material, and now include an additional 11 data points. Further, we analysed an additional 14 $^{234}\text{U}/^{238}\text{U}$ by solution, which allowed us to calculate an additional 9 initial $^{234}\text{U}/^{238}\text{U}$ values. This yielded a new average of 1.013 ± 0.178 (2SD). With this new dataset we established that there was no significant trendline of $^{234}\text{U}/^{238}\text{U}$ through time. We then use a speleothem specific $^{234}\text{U}/^{238}\text{U}$ initial correction for all samples <1.5 Ma and the average value for samples >1.5 Ma, where an appreciable difference in $^{234}\text{U}/^{238}\text{U}$ is no longer discernible. For the samples where we use an average value, we have now employed a much large uncertainty window ($\pm 2\text{SD}$) and also calculated 50,000 Monte Carlo simulations of the initial in the disequilibrium age calculation using the DQPB software. This results in a much more robust estimate of the age uncertainty, addressing the concerns of both reviewer 1 and 2. In addition, we collated a database of all the average $^{234}\text{U}/^{238}\text{U}$ of speleothems in the greater Arabian Peninsula region (n =1300) and calculated the average $^{234}\text{U}/^{238}\text{U}$ initial values and range as well as compiling data of the activity ratios in modern aquifers in the region. Our calculated initial values fit very well within the regional estimates, which predominately fall between 1 and 1.1. This suggests that our estimates of the initial are reasonable and regionally broadly representative.

2) Preservation bias and representativeness of samples in regard to the broader population. We have addressed this by producing more U-Pb ages (an additional 17 U-Pb ages) and including four additional speleothems to our dataset. This has resulted in almost twice the number of U-Pb ages compared to the original submission, which is more than the total number of ages in many papers. This subsequently also included adding previously undated speleothems to the paper, including 4 speleothem samples; SA01, SA18, SA19, and SA26. Adding such a large proportion of new data and material may have potentially resulted in different interpretations of the new dataset. However, despite the substantial re-evaluation of our initial ages as well as the addition of new ages, the main messages and general climate interpretations remain unchanged in the new version and the additional samples align well with the pre-existing dataset from our original submission.

3) Faunal dispersal inferences and discussion. To address this, we taken the advice of Reviewer 3 and removed text inferring potential dispersals from first appearance dates between different biogeographic realms. We have significantly toned down the dispersal implications, while still retaining sections about the specific fauna fossils found in Arabia and the potential implications. This has shifted the focus towards the climate story (as suggested by Rev. 3) whilst also retaining some brief discussion about the fauna.

Regarding specific reviewer comments, please find our responses to reviewer comments below. Reviewer comments are given in **bold black text** and our responses in plain black text. Where appropriate, sections of the revised manuscript are copied here and new text is shown in **plain green text**. Line numbers correspond to the manuscript with track changes document.

Yours Sincerely,

Monika Markowska on behalf of all authors

Referees' comments

Referee #1 (Remarks to the Author):

This work by Markowska et al., details recurring humid geographical phases of the Arabian Peninsula through the last 8 Myr by way of proxy information from speleothems. Wetter periods are inferred from both the presence of speleothem growth, and by encapsulated H and O isotopic fractionation. Wetter times, referred to as Central Arabian Humid Periods (CAHPs), are inferred by the presence of speleothem growth, as speleothems require both moisture and the presence of organic respirators (such as plants) to produce acidic water for the secondary precipitation of calcite.

Thank you for this review. We believe this summary accurately reflects our paper content.

The use of speleothem U-Pb ages to extend the paleoclimate record of the Arabian region is particularly inspired in this work. It provides a high-impact method of producing reliable ages much further into the geological past than traditional U-Th ages alone. Moreover, using laser ablation to produce U-Pb ages in this work helps cement this acquisition method's utility in the broader field.

Thank you for this comment, and in particular, we wish to also emphasise that our paper does indeed provide novel chronological information for Arabia owing to application of U-Pb dating methods to a speleothem archive spanning 8 Ma and highlights the utility of laser ablation as a robust method which will encourage its future wider application in the field.

I have re-analysed the reported raw U-Th and U-Pb data and produced concordant ages, suggesting to me that the provided U-Th-Pb ages are of high fidelity. My only concern is their $^{234}\text{U}/^{238}\text{U}$ initial disequilibrium values applied to the U-Pb ages (outlined below). However, I do not think that this choice will significantly impact their CAHP interpretations. I see no reason from a speleothem age perspective that this work should not be accepted.

Thank you, Rev. 1, for these points about dating. We appreciate your conclusion that our ages are of "high fidelity" and that from a speleothem age perspective that there is no reason why this work should not be accepted. With respect to the concerns about the initial disequilibrium, we have re-evaluated all the U-Pb ages in our dataset, as well as analysing additional ages. We have calculated the initial $^{234}\text{U}/^{238}\text{U}$ disequilibrium of either the speleothem specific initial $^{234}\text{U}/^{238}\text{U}$ or an average value of all speleothems, using 2σ deviations. We have also expanded our initial dataset and compiled a regional dataset of initial $^{234}\text{U}/^{238}\text{U}$ for the entire Arabian region to assess how our data aligned with the regional averages.

I believe this work is novel, of high importance, and broad interest to the wider field. The recurring "greening" of this region provides a useful timing for the land migration of fauna (including hominids) between Eurasia and Africa. This greening has been well-documented, in my opinion, by the speleothem record. I think this work should be accepted for publication following the address of the listed comments below. I would like to commend the authors on a well-written piece and for this impressive work.

Thank you for your positive assessment indicating that our new speleothem record provides important insight into the "greening" of Arabia, as well as your positive recommendation concerning the publication of our paper.

Sincerely,
Dr. John Engel
(jengel@lanl.gov)

Comment 1) On Fig 3, the 18O values seems to overlap (at least by eye) from 1Ma backwards.

Can the authors please include the 2σ deviation with the average values. Have statistical tests been done to distinguish these groups to a significant level? I would suggest an ANOVA (one-way) of the oxygen isotopes (Fig 3g and Fig 4c) to test statistical significance between the CAHPs to see if their variances overlap. The omnibus ANOVA will test the effect of age on oxygen isotope fractionation (i.e., rainfall), and the post-hoc analysis will say which age groups are significantly different from each other.

We agree that a statistical evaluation on the grouped isotope values in Fig. 3 is necessary to determine whether they are statistically distinguishable at a significant level. We have since removed the “CAHP” groupings due to a comment from Reviewer 2, however, we retain the distinction of the isotope composition of different time periods in both figures (late Miocene, Pliocene, Early-to-Mid Quaternary, and Late Quaternary). A new section has been added that includes one-way ANOVA and post hoc Tukey tests that have now been performed and we have additionally also applied the statistical evaluation on the fluid inclusion dataset. The results from the statistical evaluation can be found in Section 2.4 in Supplement 1 and are summarized below:

Lines 299 -313:

“Statistical tests

To determine whether the isotopic composition of the carbonate and fluid inclusions statistically differed in d¹⁸O mean and variance, one-way ANOVA and post hoc Tukey tests were performed using the “dplyr” package in the “R” software. Isotope values were split into four groups based on geological time epochs: Late Miocene, Pliocene, Early-to-Mid Pleistocene, and Late Pleistocene. Early-to-Mid Pleistocene (2.56 – 1 Ma) and Late Pleistocene (1 – 0 Ma) were defined by the start of the mid-Pleistocene transition at ~1 Ma. Of the four epochs investigated, all showed statistically significant differences in the carbonate d¹⁸O values (p<0.05; Table x). The fluid inclusion isotope data showed that the Late Miocene, Pliocene and Early-to Mid-Pleistocene were all statistically different to the Late Pleistocene, which had significantly higher d¹⁸O and d²H values. Additionally, the Late Miocene was also significantly different to the Early-to-Mid Pleistocene.”

ANOVA and Tukey Honest Significant Differences	p-value (95% confidence intervals)		
Time Epoch	d¹⁸O_{carb}	d¹⁸O_{FI}	d²H_{FI}
Late Miocene and Early-to-Mid Pleistocene	0.00	0.68	0.04
Late Pleistocene and Early-to-Mid Pleistocene	0.00	0.00	0.00
Pliocene and Early-to-Mid Pleistocene	0.00	0.77	0.69
Late Pleistocene and Late Miocene	0.00	0.00	0.00
Pliocene and Late Miocene	0.00	0.14	0.17
Pliocene and Late Pleistocene	0.00	0.00	0.00

Comment 2) Figure 2: Why were 250 Ka bin sizes chosen for U-Pb?

In the original submission, 250 ka bin sizes were chosen as that was the size of the approximate median uncertainty of the U-Pb individual ages. Since the original submission, we have expanded the dataset, adding additional age measurements and more robust disequilibrium age uncertainty calculations. As such, the new median uncertainty is approximately 300 ka and the bins have accordingly been adjusted to reflect this value.

Comment 3) L317: “The largest temporal cluster of U-Pb ages around 3.5 Ma”. Was this cluster of U-Pb ages from a single cave chamber or from multiple caves/chambers? If the majority of ages are from a single cave chamber, this cluster may represent a preservation bias, not necessarily a trend of the region’s speleogenesis potential. For example, this single chamber (if that’s the case) may have selectively survived this relatively-active geological area (<https://doi.org/10.3389/feart.2022.851737>).

This is an important point. The described cluster of U-Pb ages at ~3.5 Ma was from several different speleothem samples (n = 7) from five individual caves (Friendly Cave, Hotel Cave, EP19.8 Cave, Luxury Cave and Mossy Cave). Consequently, this does not suggest the cluster is an artefact of preservation bias from one single chamber or cave, but rather reflects the overall wetter conditions allowing speleothems to form in several individual cave systems. The individual caves where the speleothems were sampled are indicated in Supplementary data file 3 and Supplement 1 section 1.1.

Comment 4) L599: The use of younger U-Th values to estimate the $^{234}\text{U}/^{238}\text{U}$ initial disequilibrium of U-Pb ages is potentially problematic. First, please state the average U-Th based $^{234}\text{U}/^{238}\text{U}$ +/- 2σ here in the text.

This has been amended in the text as suggested. The average $^{234}\text{U}/^{238}\text{U}$ has been recalculated on an expanded dataset based on the comments from Reviewers 1 and 2 and is now 1.013 ± 0.178 (2σ).

Lines 1113-1117: “The initial $^{234}\text{U}/^{238}\text{U}$ were estimated from the average measured $^{234}\text{U}/^{238}\text{U}$ on younger (<1.5 Ma) U-Th and U-Pb dated samples to calculate the initial $^{234}\text{U}/^{238}\text{U}$, based on the isotope decay constants yielding an average value of 1.013 ± 0.178 (2σ). Disequilibrium ages were calculated using DQPB software¹ and uncertainties are given in Extended data File 2.

Additionally, in the supplementary documents, please explicitly list the U-Th speleothems, and their $^{234}\text{U}/^{238}\text{U}$, used to calculate this value.

This has been added as an extra worksheet entitled “initial 234/238 measured” in Supplementary data file 2.

Can you please explain your rationale for this decision? Initial disequilibrium is one of the chief impediments to accurate U-Pb calcite ages (in addition to very small analyte concentrations). Consider the hypothetical situation where 6 Myr ago, the source waters entering the karst were interacting with a local granite and extracting a very different $^{234}\text{U}/^{238}\text{U}$ than those recorded by the U-Th ages <700ka. I’m not saying this is necessarily the case, but I believe your applied value of 0.939 ± 0.054 (inferred from the supplementary file) is artificially small and may not incorporate this potential geological scatter.

Thank you for raising this important point. We have re-evaluated all the ages in our dataset and significantly adjusted the initial disequilibrium value and uncertainty used in our uncertainty propagation of error. Further, we have collated all the regional speleothem $^{234}\text{U}/^{238}\text{U}$ data for the Arabian Peninsula (n = 1300) and calculated the initial $^{234}\text{U}/^{238}\text{U}$, yielding average values consistent with those from our dataset. See further details in our responses to comments directly below.

I would suggest two potential options. First, expand the uncertainty to a relatively large, arbitrary $^{234}\text{U}/^{238}\text{U}$ initial disequilibrium, say 1 ± 0.2 , and see if the CAHPs change from the amended U-Pb ages. My guess is that given the scale of Fig 3 x-axis, the CAHP’s will not change significantly. Second, you might try our 2019 approach (<https://doi.org/10.1016/j.quageo.2019.101009>) where we used ^{235}U - ^{207}Pb isochrons

(referenced to 208Pb) as a way to bypass 234U disequilibrium (at the cost of reduced precision). I see that 235U was not an analyte measured during laser ablation, but you might estimate it using natural U composition from the 238U signal. Tim Pollards' new software may help with this (<https://doi.org/10.5194/gchron-5-181-2023>)

Thanks for the detailed suggestions and practical amendments to improve our estimation of our U-Pb age uncertainty due to the initial $^{234}\text{U}/^{238}\text{U}$. We have since expanded the uncertainty as you suggested and followed "option 1". As per Reviewer 2's suggestions (see below) we have also expanded the number of $^{234}\text{U}/^{238}\text{U}$ measurements to calculate an average $^{234}\text{U}/^{238}\text{U}$ initial disequilibrium value and 2σ . Our revised average initial disequilibrium is 1.013 ± 0.178 , consistent with Rev. 1's suggestion of implementing an initial of 1 ± 0.2 . We further accepted your recommendation to calculate the disequilibrium ages based on the DQPB software (Pollard et al., 2023) and used Monte Carlo simulations (x50,000) of the initial $^{234}\text{U}/^{238}\text{U}$ to more robustly account for the uncertainty arising from the initial $^{234}\text{U}/^{238}\text{U}$. To assess whether our data is also in line with regional estimates, we collated a database for all speleothem $^{234}\text{U}/^{238}\text{U}$ in the Arabian Peninsula and found remarkable consistency, where most values fell between 1.0 and 1.1 (see SFig 12).

U-Pb supplementary data

U-Pb calculates ages sheet

Comment 5) Can you please provide more information about which samples these data are coming from? For example, are SA17 ages 6, 5, 3, and 2 all from the same speleothem?

This would allow readers to better interpret the CAHP periods based on U-Pb ages. For instance, SA17-2 and SA17-3 are within age uncertainty: if they're from the same sample, wouldn't this be "double dipping" of ages to increase the population sizes for the CAHPs? To help make this more clear, you could color-code the U-Pb ages in Figure 3 to match the oxygen isotopes plotted below them.

SA17 related to a single speleothem and suffixes following i.e. "-1" relate to the individual date from that speleothem. We have dated multiple segments to capture all the growth phases. For example, sample SA09 has ages: 0.360, 2.188, 2.290, 4.098 Ma. To make this clearer we have colour-coded the U-Pb ages to match the carbonate box plots, as suggested. Further, these ages are clearly outlined on diagrams of the speleothem thick sections in Supplement 1.1. In relation to Fig. 3 we have grouped the isotope box-plots by speleothem and only separated this into individual box plots when the uncertainty of the ages are not overlapping. For example, where there are two dates from a single speleothem that are overlapping, we present the carbonate isotope box plot as one sample. In the case of SA09 where the uncertainty of the ages do not overlap we present these individual growth stages as separate box plots.

Comment 6) On point (6) please provide the 2σ uncertainty of the averaged $^{234}\text{U}/^{238}\text{U}$ calculated from the U-Th ages.

We have provided this in the Methods section in the text and in Supplementary 2 tab “Initial $^{234}\text{U}/^{238}\text{U}$ measured”.

Line 1113-1117: “The initial $^{234}\text{U}/^{238}\text{U}$ were estimated from the average measured $^{234}\text{U}/^{238}\text{U}$ on younger (<1.5 Ma) U-Th and U-Pb dated samples to calculate the initial $^{234}\text{U}/^{238}\text{U}$, based on the isotope decay constants yielding an average value of 1.013 ± 0.178 (2σ). Disequilibrium ages were calculated using DQPB software¹ and uncertainties are given in Extended data File 2.”

Comment 7) On point (8) I was not certain about the interpretation given. Is it perhaps supposed to be the added uncertainty (with (7) being the subtracted uncertainty) on applied initial $^{234}\text{U}/^{238}\text{U}$?

We have updated our Supplementary 2 data file and clarified the notes as follows and have indicated the columns which contain the “relative” and “absolute” uncertainty.

- “(1) Number of analyses considered.
- (2) Spine width, or the median absolute deviation of weighted residuals.
- (3) Upper 95 % confidence bound of spine width.
- (4) Concordia equilibrium ages calculated using the DQPB software, using the spine algorithm and including decay constant uncertainties and the natural U^{238}/U^{235} uncertainties.
- (5) Indicates whether a sample or average value were used to calculate the initial $^{234}U/^{238}U$.
- (6) For samples <1.5 Ma the $^{234}U/^{238}U$ was measured using solution chemistry.
- (7) Calculated initial $^{234}U/^{238}U$ value used in the disequilibrium age calculation
- (8) Disequilibrium age. Calculated using the DQPB software, using the spine fit and including decay constant uncertainties and the natural $^{238}U/^{235}U$ uncertainties. Calculated using 50,000 Monte Carlo simulations for the initial. For samples where the average initial value was used, an uncertainty of 2 standard deviations was used (0.178).
- (9) Lower 95% confidence interval relative uncertainty.
- (10) Upper 95% confidence interval relative uncertainty.
- (11) Lower 95% confidence interval absolute uncertainty.
- (12) Upper 95% confidence interval absolute uncertainty.
- (13) An equilibrium age was assumed for this gypsum sample.”

Sample raw data sheet

Comment 8) The error correlation value rho seems abnormally small in this work, especially when compared with previous laser ablation calcite studies (e.g. <https://doi.org/10.5194/gchron-3-35-2021>). Is this a unique consequence of your instrumentation? If so please provide context for why this is the case.

The rho of the $^{207}Pb/^{206}Pb$ vs $^{206}Pb/^{238}U$ is calculated following Schmitz & Schoene², Noda³ and Ludwig⁴. While the rho of $^{207}Pb/^{235}U$ vs $^{206}Pb/^{238}U$ is commonly around 0.5-0.99, that of the $^{238}U/^{206}Pb$ - $^{207}Pb/^{206}Pb$ is not well correlated, thus close to 0. We have found multiple studies do not report the rho ($^{207}Pb/^{206}Pb$ vs $^{206}Pb/^{238}U$), for example in the paper cited by the reviewer, the BGS lab only reported $^{207}Pb/^{235}U$ vs $^{206}Pb/^{238}U$. We have observed that only labs using Iolite data processing software (such as the UCSB lab in the above paper) report higher error correlation rho values for $^{207}Pb/^{206}Pb$ vs $^{206}Pb/^{238}U$. We speculate that it is because Iolite calculates rho as Pearson sample correlation coefficient. Consequently, we disagree that our rho is ‘abnormally small’.

Comment 9) Please elaborate on your criteria for removing certain points from the U-Pb isochrons (e.g., SA10-6).

This is a helpful observation – thank you. In our age calculations, the points which have been omitted were those with negative ^{207}Pb counts. All other points have been included in the age calculation. There were a few points in the previously submitted version, for example in samples SA10-6, which were removed as outlier points. To address this comment, we have now also included these previously excluded datapoints and instead used the spine algorithm in the DQPB software (as suggested to us by Rev. 1) to recalculate the disequilibrium ages, as the spine algorithm is better for dealing with outliers.

Referee #2 (Remarks to the Author):

The manuscript by Markowska et al. presents an extensive data set from Saudi-Arabian speleothems including new U-Th and U-Pb chronology, stable isotope ($\delta^{18}\text{O}$) and fluid inclusion data together with a thorough literature study of first appearances dates of mammals in Eurasia and Africa. Based on the timing of the absence and presence of speleothem growth, the authors defined five humid periods in Central Arabia over the last 8 million years that seem to coincide with faunal dispersals between Africa and Eurasia.

We thank the reviewer for their positive summary of our work in the original manuscript.

The speleothem $\delta^{18}\text{O}$ and fluid inclusion data set are impressive. They found a clear increasing trend in their $\delta^{18}\text{O}$ data suggesting that this region became progressively drier towards the Pleistocene. $\delta^{18}\text{O}$ and $\delta^2\text{H}$ fluid inclusion analyses were then used to identify the atmospheric moisture source. This is very interesting palaeoclimatological data providing new insights into the climatic conditions during the Miocene to Pleistocene in Central Arabia – an area for which previously limited palaeoclimatological information from terrestrial records was available, particularly for the Miocene to Early Pleistocene.

We thank reviewer 2 for their positive assessment of our carbonate and fluid inclusion record which together provides a unique, extraordinarily long record of continental hydroclimate in an area with no previous data.

The main implications of this manuscript, though, are linked the five identified humid periods based on the speleothem ages. The authors report an extensive collection of mammalians first appearances dates which seem to coincide with the Central Arabian humid periods. They ultimately suggest that due to periodic climatic ameliorations, the Saharo-Arabian desert acted as an important corridor for faunal dispersals between Africa and Eurasia. While the implications of this narrative are important and of immediate interest to people from several disciplines, they stand or fall with the quality of the chronology and the statistical significance of the defined humid periods – and, unfortunately, there are issues with the U-Pb data set.

We thank both reviewers 1 and 2 for their detailed review of the ages in the original manuscript. As noted, we have re-evaluated all the dates in the manuscript following the advice of reviewers 1 and 2. We have also undertaken a rigorous assessment on the $^{234}\text{U}/^{238}\text{U}$ uncertainty of the ages (see detailed evaluation below) as well as expanding our dataset by adding additional speleothems as well additional dates on speleothems presented in the original manuscript. We believe that we have robustly addressed all the issues raised by both reviewers 1 and 2 about the U-Pb dataset.

The two major issues are: 1) the sample size for ages beyond ~500 ka is only 22 and seems too small to identify five humid periods over such a long period of time (8 million years) with enough confidence,

Thank you, Reviewer 2, for this comment. In response, we have removed the identification of 5 distinct humid periods within the manuscript and now refer to 'humid episodes', making it clear that we do not consider periods of overlapping ages as million-year long humid phases. Further we have also produced new ages to further bolster the sample size as well as adding 4 new samples (just measured) previously not in the original manuscript.

and 2) the disequilibrium correction using the initial $^{234}\text{U}/^{238}\text{U}$ activity ratios to calculate the final U-Pb ages was not done adequately, which could drastically change the timing of the identified humid periods. Both these issues undermine the main conclusions of this paper, i.e., that the humid phases coincided with major faunal dispersals between Africa and Eurasia.

We thank reviewer 2 for this comment and agree that the $^{234}\text{U}/^{238}\text{U}$ uncertainty is often the largest in speleothem U-Pb dating. We have reevaluated our entire dataset, accordingly, and produced further $^{234}\text{U}/^{238}\text{U}$ measurements for younger speleothems (<1.5 Ma) and addressed reviewer 2's major concern of a trendline in $^{234}\text{U}/^{238}\text{U}$ through time which could have potentially significant implications for the ages

calculated. In our expanded dataset we find no evidence for a trend in $^{234}\text{U}/^{238}\text{U}$ through time. Further we have compiled a regional dataset of all measured $^{234}\text{U}/^{238}\text{U}$ for Arabia (n= 1300) as well as compiling groundwater data for the region. We find the average values that we apply for the $^{234}\text{U}/^{238}\text{U}$ correction are in line with regional values and in our expanded dataset we report an average $^{234}\text{U}/^{238}\text{U}$ close to 1. We have expanded the uncertainty on the $^{234}\text{U}/^{238}\text{U}$ correction in our age calculations. For further details please see our specific responses to comments in the below sections.

I support the palaeoclimatological interpretations based on the identified trend in the $\delta^{18}\text{O}$ and regarding the moisture source based on fluid inclusion data. I am very positive about this data and believe they are of immediate interest to the palaeoclimate community and beyond. The chronological data set is impressive in the sense that, until this data set, only a handful of speleothem ages older than 500 ka existed from this region, and furthermore, this study is the first to report ages for this area beyond the Pleistocene.

We thank Reviewer 2 for their positive assessment of the impact our study has for the paleoclimate community and beyond, as well as in terms of its important role in filling a critical gap in our knowledge of the climate in this region beyond 500 ka.

However, due to the small sample size and issues with the disequilibrium corrections of the U-Pb ages, I cannot recommend this manuscript for publication in Nature in the current form. I will explain the abovementioned issues in more detail below and provide recommendations to the authors on how to address them and improve their manuscript.

We thank Reviewer 2 for the recommendations provided below and we have followed these and provided detailed responses in the respective sections. We believe we have a much more robust U-Pb dataset as a result.

Major issues:

Sample size

1. For the conclusions and implications of this paper, the absence of speleothems is just as important as their presence (because, simply put, present = wet, absent = dry). The U-Pb ages from this area are novel and very interesting, but from a statistical point of view, the sample size is too small to identify statistically significant humid phases. 8 million years is a very long time – do you have any statistical means to evaluate whether it is true what you see, or what you don't see (=absence of speleothems)? How confident are the authors that the 22 U- Pb ages truly represent the speleothem growth phases in this area? A recent speleothem-based study (Weij et al. 2020) investigated this issue and from their simulations it is obvious that 22 ages are not enough to distinguish between randomness and the true signal. The authors have not discussed the effect of sample size on the statistical significance of the identified speleothem growth periods, but this should really be addressed.

We thank the reviewer for the observation, and stress that we do not suggest that all the potential growth phases are covered in our current dataset as statistically this is not possible. Nonetheless, to address this comment, we have increased our sample size by 17 new U-Pb dates and added 4 additional speleothems to our record.

The identification of 5 discrete humid phases, may be misleading as it could lead the reader to assume these are continuous. Consequently, we have removed the specific sections where we identify "5" discrete humid phases and have emphasised that periods of speleothem growth are episodic wetter intervals, and rather focus on the climatic interpretation of the clear trend that can be seen through time of these wetter episodes. We have directly identified this in the text and made it clear that gaps in our record cannot be interpreted directly as climate information (i.e. caption in Figure 2). The generation of the 150+ ages suggested by Weij et al.⁵ is only possible with access to an even larger sample size than presently available, and usually from broken rubble speleothems. Instead, we have chosen to highlight the present limitations of the study (despite being one of the largest speleothem U-Pb datasets to date). We have also toned down the discussion on the terminal Miocene period and specifically outline that

we cannot statistically interpret the gap in our record. Instead, we acknowledge it is there, and discuss other regional phenomena at that time. Further details are provided below in our specific responses.

2. Another issue is a potential preservation bias of speleothems, i.e., destruction of speleothems with time. Both Scroxton et al. (2016) and Weij et al. (2020) discuss an exponential decay or “half-life” of speleothems with age, meaning that the further back in time, the less likely a speleothem would have been preserved. Therefore, the absence of speleothems does not necessarily mean that there really weren’t any speleothems growing at that time. What does that mean for the speleothem data set presented in this study?

We are aware of the preservation bias of speleothems whereby ‘natural attrition’ means that in some cases one is less likely to see speleothem preservation in deeper time as the caves may have experienced downward erosion of the karst terrain, cave collapse, in-cave erosional processes, in-cave sedimentation and speleothem precipitation covering stalagmites. We do not think this process is as relevant in dryland cave systems, compared to the tropics, and have included an additional paragraph in the Supplement 1 Section 2.5 discussing this.

Supplement 1 Lines 340-357: “Preservation bias due to ‘natural attrition’ processes results in less speleothem preservation further back in time. This is due to processes either removing or covering older material such as; downward erosion of the karst terrain, cave collapse, in-cave erosional processes, in-cave sedimentation and speleothem precipitation covering older stalagmites. Scroxton et al.⁶ focusing on a tropical site in Sulawesi, Indonesia, suggest stalagmites follow an exponential relationship of decreasing numbers through time. Whilst this is likely very relevant for this site in a tropical setting which is not periodically water limited, the opposite trend is observed in central Arabia, whereby more speleothem material by volume has been deposited in the late Miocene and Pliocene, than recently. This was also reflected in an earlier sampling campaign⁷ where most of the samples found were beyond the U-Th dating technique (>500ka). We suggest that dryland environments do not experience high rainfall erosivity like the tropics, thus may be more suitable for speleothem archive preservation in deeper time. For example, the Nullarbor speleothem record in arid southern Australia (i.e. Woodhead et al.,⁸) shows a similar trend to our data in that most of the speleothems in those cave localities formed in the Miocene and Pliocene, with little formation at all in the Quaternary. Further, a similar story is emerging from stalagmites collected from caves in the Negev desert, Israel⁹. Therefore, the fact that we see most speleothems preserved in these deeper time intervals gives us confidence that potential gaps in our data such as the Late Miocene are unlikely to be a result of natural attrition.”

Both these aspects should be addressed together with why the authors believe that the signal seen in the speleothems presented here is in fact the true absence of speleothems due to unfavourable climatic conditions.

We have addressed both of these comments, with further details in the below sections.

U-Pb age calculation

1. Initial $^{234}\text{U}/^{238}\text{U}$ disequilibrium correction of the U-Pb ages: the authors have not directly measured the initial $^{234}\text{U}/^{238}\text{U}$ activity ratio in their U-Pb dated samples – why not?

We thank reviewer 2 for this comment. We have, in a response, directly measured the initial $^{234}\text{U}/^{238}\text{U}$ activity ratios in all our samples where the initial $^{234}\text{U}/^{238}\text{U}$ activity ratio is distinguishable from equilibrium (<1.5 Ma). To make this clearer in the revised version we have provided this data in the same spreadsheet as the U-Pb dating information. We further have provided additional measurement of initial $^{234}\text{U}/^{238}\text{U}$ activity ratios to expand the fidelity our dataset. Specific details can be found below in our responses to Reviewer 2’s detailed review on the initial $^{234}\text{U}/^{238}\text{U}$ activity ratio correction.

The final disequilibrium correction is crucial and can affect the final age quite significantly. For samples older than 3-4 Ma, there probably won’t have any measurable $^{234}\text{U}/^{238}\text{U}$, but samples between 0.5 and 3 Ma probably do. In general, it is best practise to have corresponding residual $^{234}\text{U}/^{238}\text{U}$ measurements for U-Pb dated samples to back calculate the initial $^{234}\text{U}/^{238}\text{U}$ activity ratio. For any older samples without measurable $^{234}\text{U}/^{238}\text{U}$, one could then

use a weighted average or trendline-based estimate from the younger U-Pb dated samples, depending on whether a trend with age is observed or not. Particularly for a paper like this with important implications that entirely depend on the chronology, I would expect that the U-Pb ages have a corresponding $^{234}\text{U}/^{238}\text{U}$ measurement. Instead, the authors have used a weighted average initial $^{234}\text{U}/^{238}\text{U}$ activity ratio obtained from the solution U-Th dated samples. The problem with this approach is that there is an obvious trend in initial $^{234}\text{U}/^{238}\text{U}$ with U-Th age, as seen in the plot below:

We thank Reviewer 2 for this suggestion and have consequently taken their advice and back calculated the initial $^{234}\text{U}/^{238}\text{U}$ activity ratio for all our younger samples (1.5 Ma). Samples older than 1.5 Ma show no significant deviation from 1 (Fig. 1). For samples that are older than 1.5 Ma we have used an average initial $^{234}\text{U}/^{238}\text{U}$ value, calculated from our expanded dataset. However, this is following the re-evaluation of all our dataset to assess whether the trendline identified by Reviewer 2 was still significant in our expanded dataset. We found this was not the case and thus used an average value for samples where the U-Pb age was greater than 1.5 Ma. In additional we further increased the uncertainty of the average initial $^{234}\text{U}/^{238}\text{U}$ to 2 standard distributions. Details for all these changes can be found in our specific responses below.

[FIGURE REDACTED]

Fig 1. Measured activity ratios (via solution MC-ICPMS) vs age (ka).

This trend suggests that the initial $^{234}\text{U}/^{238}\text{U}$ could have been much higher for older U-Pb ages than the weighted average currently used to perform the disequilibrium correction. It follows that the U-Pb ages then are actually younger than currently reported. This is very problematic. To demonstrate why, I have fitted a linear trendline through the data (excluding the one outlier) and recalculated the U-Pb disequilibrium ages using a trendline-based estimate initial $^{234}\text{U}/^{238}\text{U}$ activity ratio (in DQPB software, Pollard et al. 2023). I was able to reproduce the isochrons and most of the equilibrium ages with their errors and MSWD using a model 1 fit type (see table on the final page of this document). The authors should check why some ages/age errors/MSWDs are different from the ones recalculated, perhaps some typos?

We thank the reviewer for pointing out this potential problem with the dataset. We have recalculated all the dates in our dataset using the aforementioned software (DQPB) for robust evaluation of the $^{234}\text{U}/^{238}\text{U}$ corrections. We have updated this in the Supplementary dataset and expanded the Supplement 3 dataset to provide further detail on the calculation fit method. As our $^{234}\text{U}/^{238}\text{U}$ calculated and average values are very close to one and there is no significant trend line through time, there is no evidence that the initial $^{234}\text{U}/^{238}\text{U}$ was much larger further back in time. Consequently, the concern that the calculated ages in our dataset could be much younger is not valid. Details for all these changes can be found in our specific responses below.

More importantly, the disequilibrium U-Pb ages I recalculated using a trendline-based value for initial $^{234}\text{U}/^{238}\text{U}$ are drastically lower than calculated by the authors (again see table one the final page of this document). This means that the humid periods will shift and do not coincide so well anymore with NH cooling periods nor with faunal dispersals, see figure below. If this is true, then the conclusions of this manuscript will have to be revisited and may not hold anymore.

Again, we thank Reviewer 2 for their detailed review and suggestions for revision. The trendline in the expanded dataset has a reduced slope (0.0001) compared Reviewer 2's trendline (0.00038775) and there is more scatter in the expanded dataset. We do not observe a meaningful trend through time in the $^{234}\text{U}/^{238}\text{U}_{\text{initial}}$. The apparent trend is mainly due to the high sampling of the U-Th dated samples (SA31 and SA30; Fig. y) at ~200 ka (x 6 dates and initial measurements) which were all very tightly constrained with low $^{234}\text{U}/^{238}\text{U}_{\text{initial}}$. Removing these data results in a non-statistically significant relationship between $^{234}\text{U}/^{238}\text{U}_{\text{initial}}$ and age ($p < 0.05$). Further, our data are not normally distributed (Shapiro test $p = 0.03$). Consequently, using a linear regression line trend would be inappropriate in this instance. We further suggest that extrapolating any trend over a 6.5 Ma period without a solid mechanism to explain such a trend could introduce further uncertainty in the age calculations. Although not specifically requested by Reviewer 2, we also collated all the regional data for Arabia to assess whether there was a statistically significant trend through time in the $^{234}\text{U}/^{238}\text{U}_{\text{initial}}$ ($n=1300$; Fig. y). This revealed no trend in the initial ($R^2 = 0.00$; $p > 0.05$), consistent with what we report in our dataset. Details for all these changes can be found in our specific responses below as well as our new recalculated trendline.

2. Fit-type U-Pb isochrons: first, the authors did not specify which fit-type model was used to calculate the U-Pb isochrons – this should be included for reproducibility of the ages. After recalculating U-Pb equilibrium ages, I was able to reproduce them using Model 1 fit type, so I assume it is this one the authors have used?

Yes we used Model 1. We have since reevaluated our entire dataset at Reviewer 2's suggestion and used the spine fit type suggested in the comment below. We have outlined the fit method used in the Supplementary as well as in the Methods section in the manuscript.

Model 1 result different age uncertainties than for example the spine fit type. The spine fit type is generally more robust than the classical model 1 fit type (see Powell et al. 2022 and Pollard et al. 2023). When using the model 1 or spine fit type, the ages themselves differ only within a few percent of each other, so that is fine. However, since the age uncertainties will be different for model-1 or spine ages, this will affect the timing of the humid periods if they are defined by their age uncertainties. The choice of fit type is ultimately up to the authors, but the issue

above should be considered and addressed, and their choice of fit type should be specified and explained.

We have taken onboard Reviewer 2's comments about the fit method used to calculate the ages. In the initial submission we used the Model 1 fit. As Reviewer 2 rightly mentioned, the spine method is more robust, so we have implemented this model for the calculation of all our ages with the DQPB software¹. We have updated the notes in the Supplementary data file 2 specifying clearly the fit used as follows:

“(8) Disequilibrium age. Calculated using the DQPB software, using the spine algorithm and including decay constant uncertainties and the natural ²³⁸U/²³⁵U uncertainties. Calculated using 50,000 Monte Carlo simulations for the initial. For samples where the average initial value was used, an uncertainty of 2 standard deviations was used (0.178).”

Humid phases and the terminal Miocene dry period:

- 1. I'm wondering how exactly the humid periods defined? In the text, they seem to be defined by their mean ages, but in the figures by their age uncertainties. The authors should be consistent and also explain their choice in the text (main or methods).**

Humid periods are defined in the text by their mean ages. The CAHPS have been removed due to comments from Rev. 1 and 2. In the figures we instead include frequency plots of ages (Fig. 2h) with corresponding background shading reflecting the frequency of ages using a bin width of the average radiometric uncertainty. We have added a caveat about erroneous interpretations in the caption of Fig. 2:

Lines 332-333: “...As not all speleothem layers were dated it should not be used as a record of paleoclimate.”

- 2. The small sample size and issues with the disequilibrium correction of the U-Pb ages have both raised the following question: how sure are the authors be that the speleothem growth phases we see are discrete events modulated wet climatic conditions? And that the absence of speleothems really means that there were no speleothems growing at that time? They could be absent in the dataset because they really weren't there, or simply missed because of a small sample size, or destroyed/buried due to a preservation bias.**

We have now addressed this issue in the revised paper. Please see response to Major issue 2 above (page 11).

- 3. This leads to the section addressing “*the Terminal Miocene dry period*”: again, the underlying chronological data aren't very strong to support the conclusions drawn from it. The dataset includes 22 speleothem ages, of which 10 are older than 3 Ma. What makes the authors sure that there really weren't any speleothems growing during the terminal Miocene period? Also, if the final disequilibrium-corrected ages will change, then this interpreted “dry period” will not coincide with the terminal Miocene anymore.**

We appreciate that an absence of evidence is not always evidence of absence. We have addressed this in the following way:

- We removed the CAHPS and the colour shading of distinct humid periods on both Fig. 2 and 3.
- We have clarified that the periods of speleothem growth that we have identified reflect episodic humid phases, rather than a single continuous ‘humid period’ (which we were never suggesting but appreciate that the figures may have given this impression).
- We further have added an additional 17 U-Pb radiometric ages to the paper.

- We have significantly toned down the reference to the Terminal Miocene Dry Period and removed this as a stand-alone section, reducing it to several sentences. We explicitly state that whilst most, if not all, gaps in our record are too small to exclude sampling bias as the cause for their appearance, the largest gap in our dataset, from ~6.3 to ~4.1 Ma is contemporaneous with both regional and global evidence for increased NH low-latitude aridity.

The new Terminal Miocene discussion points are as follows:

Lines 618-648: “The 8 Ma speleothem record demonstrates recurrent wetter intervals, reflecting the alternating nature of wet-dry phases in central Arabia. Whilst most, if not all, gaps in our record are too small to exclude sampling bias as the cause for their appearance, the largest gap in our dataset, from ~6.3 to ~4.1 Ma is contemporaneous with both regional and global evidence for increased NH low-latitude aridity. Late Miocene Cooling (6 to 5 Ma), when global temperatures and meridional temperature gradients shifted to approximately modern-day values¹⁰ in conjunction with NH ephemeral glaciation (6.0-5.5 Ma)¹¹ and evidence of ice-rafted debris in the North Atlantic¹² (Fig. 3a) likely led to increased aridity on the Arabian Peninsula. Regional evidence for terminal Miocene aridification can be drawn from the Messinian Salinity Crisis interval when palaeoclimate data from both the Mediterranean and Red seas document significant desiccation between 6.0 and 5.3 Ma¹³⁻¹⁵. The catastrophic shrinkage of the Mediterranean and Red seas¹⁵ is likely to have contributed to regional aridity by reducing the amount of moisture available as a direct result of changes in regional land-sea distributions. The resulting energy balance change would have decreased surface heat fluxes and net energy in the atmosphere, causing an equatorial retreat of the ITCZ, as observed in global climate models simulating similar reductions in surface water due to the Tethys Shrinkage¹⁶. Higher meridional SST gradients (Fig. 3) and regional hydroclimate change both provide mechanisms for an equatorial shift of the ITCZ away from the Arabian Peninsula in the terminal Miocene.

Recommendations

This manuscript has great potential, but with the current chronological data set, it does not have the statistical foundation to support the manuscript’s main claims. I suggest the following options:

- - **Disequilibrium correction of the U-Pb ages: the authors are strongly encouraged to measure the residual $^{234}\text{U}/^{238}\text{U}$ in their U-Pb dated samples using U-Th solution methods, at least for the samples between 0.5 and 3 Ma, and back calculate the initial $^{234}\text{U}/^{238}\text{U}$ for these samples. They should then correct the U-Pb samples with their corresponding initial $^{234}\text{U}/^{238}\text{U}$ value. The authors should then plot those values against age to look for any trends. If there is a trend, I suggest fitting a trendline through this data and, via extrapolation, using trendline-based initial $^{234}\text{U}/^{238}\text{U}$ estimates to correct the U-Pb samples for which no residual $^{234}\text{U}/^{238}\text{U}$ could be measured. If there is no trend, then an average weighted initial $^{234}\text{U}/^{238}\text{U}$ of the U-Pb samples, or possibly the entire data set, could be used.**

We appreciate Review 2’s thorough review of our age calculations and the uncertainty from the initial $^{234}\text{U}/^{238}\text{U}$. We have undertaken a thorough re-evaluation of all the initial uncertainty assumptions used for our age calculations. Briefly, we reevaluated all our ages and the $^{234}\text{U}/^{238}\text{U}_{\text{initial}}$ uncertainty based on Reviewer 1’s suggestion to increase the uncertainty on our $^{234}\text{U}/^{238}\text{U}_{\text{initial}}$ uncertainty of 0.2 (in this case that is $\sim 2\sigma$) and reviewer 2’s suggestions to expand our dataset and include the U-Pb dated samples where the $^{234}\text{U}/^{238}\text{U}$ activity ratios are still measurable. Further, we also measured additional samples and included 3 datapoints from previously published work in the same cave region⁷. With this expanded dataset we found a larger scatter in the $^{234}\text{U}/^{238}\text{U}_{\text{initial}}$ and a reduced slope, suggesting no trend through time. Further we also collated all regional data in Arabia and calculated the $^{234}\text{U}/^{238}\text{U}_{\text{initial}}$ and found no trend through time and statistically insignificant relationship. Expanded information of the aforementioned summary is detailed below.

Linear trendline

We appreciate that if there was a linear trend in the $^{234}\text{U}/^{238}\text{U}_{\text{initial}}$ through time, that this could have large implications for our age corrections.

Initial calculations for U-Pb samples where the $^{234}\text{U}/^{238}\text{U}_{\text{initial}}$ could be measured. We appreciated review 2's suggestion to add further $^{234}\text{U}/^{238}\text{U}_{\text{initial}}$ data for samples which were U-Pb dated, and the activity ratio of $^{234}\text{U}/^{238}\text{U}$ was still measurable by solution. We have taken a conservative approach of including samples 1.5 Ma and younger as after about 1.5 Ma most sample activity ratios cluster around secular equilibrium (Fig. 1). We added 11 new datapoints and three additional datapoints from a previous publication⁷. Further, we analysed an additional 14 $^{234}\text{U}/^{238}\text{U}$ by solution, which allowed us to calculate an additional 9 initial $^{234}\text{U}/^{238}\text{U}$ values. This yielded a new average of 1.013 ± 0.178 (2SD).

$^{234}\text{U}/^{238}\text{U}_{\text{initial}}$ trendline with new expanded dataset

Fig. 2 shows the new calculated trendline with the now expanded dataset, alongside the initial trendline calculated by Reviewer 2. The trendline in the expanded dataset has a reduced slope (0.0001) compared to Reviewer 2's trendline (0.00038775) and there is more scatter in the data. We do not observe a meaningful trend through time in the $^{234}\text{U}/^{238}\text{U}_{\text{initial}}$ (Fig. 2). The apparent trend is mainly due to the high sampling of the U-Th dated samples (SA31 and SA30; Fig. 2) at ~200 ka (x 6 dates and initial measurements) which were all very tightly constrained with low $^{234}\text{U}/^{238}\text{U}_{\text{initial}}$. Removing these data results in a non-statistically significant relationship between $^{234}\text{U}/^{238}\text{U}_{\text{initial}}$ and age ($p < 0.05$). Further, our data are not normally distributed (Shapiro test $p = 0.03$). Consequently, using a linear regression line trend would be inappropriate in this instance. We further suggest that extrapolating any trend over a 6.5 Ma period without a solid mechanism to explain such a trend could introduce further uncertainty in the age calculations. Although not specifically requested by Reviewer 2, we also collated all the regional data for Arabia to assess whether there was a statistically significant trend through time in the $^{234}\text{U}/^{238}\text{U}_{\text{initial}}$ ($n=1300$; Fig. 3). This revealed no trend in the $^{234}\text{U}/^{238}\text{U}_{\text{initial}}$ ($R^2 = 0.00$; $p > 0.05$), consistent with what we report in our dataset.

For the above-mentioned reasons, we do not think it is robust to use a trendline approach, but we do appreciate Reviewer 2 raising this as a potential issue. However, we have calculated the maximum potential impact this trendline approach would have if used to estimate the initial and implement it in the age calculation. The oldest sample (and hence most affected and our "worst case scenario end member") is sample SA10-3 with a disequilibrium age of 7.444 Ma. If we used the trendline approach and the trendline equation calculated from the new dataset ($y=0.0001x+0.9527$), an initial value of 1.7027 is yielded. If we use this initial assumption in the age calculation, that yields a disequilibrium age of 7.186 Ma. This is a total shift of 0.258 Ma, opposed to ~ 1 Ma suggested from Reviewer 2's trendline. Importantly, we wanted to demonstrate that potential large shifts in calculated ages from using a trendline $^{234}\text{U}/^{238}\text{U}_{\text{initial}}$ approach are not valid, and we do not see this impacting the major assumptions of our manuscript.

Fig. 2. Trendline of $^{234}\text{U}/^{238}\text{U}$ initial against time with new expanded dataset.

Further $^{234}\text{U}/^{238}\text{U}$ initial considerations

To assess the regional spread in $^{234}\text{U}/^{238}\text{U}$ initial in the region, and how representative our calculated central Arabian speleothem $^{234}\text{U}/^{238}\text{U}$ initial is, we collated and calculated (where the initial information was not provided) all the regional speleothem $^{234}\text{U}/^{238}\text{U}$ initial data for Arabia ($n = 1300$), as well as published modern regional groundwater $^{234}\text{U}/^{238}\text{U}$ measurements (Fig. 3). The regional average $^{234}\text{U}/^{238}\text{U}$ initial is 1.1, which is very similar to our average initial value which sits centrally in the distribution of regional values (Fig. 3). Furthermore, the modern activity ratios of the carbonate aquifers proximal to our cave site show similarly low $^{234}\text{U}/^{238}\text{U}$, which resembles the speleothem values. We have added this additional information to the Supplement 1.

Fig 3. Panel a distribution of regional speleothem $^{234}\text{U}/^{238}\text{U}_{\text{initial}}$ and U concentrations (ppb) as well as a histogram with the frequency of regional $^{234}\text{U}/^{238}\text{U}_{\text{initial}}$ with an $n=1300$. Panels b-d show the regional groundwater $^{234}\text{U}/^{238}\text{U}$ activity ratios for total groundwater (b), carbonate aquifers (c) and Wadis (d) in Saudi Arabia.

Redating samples at the limit of the U-Th technique. We were concerned that one potential source of the apparent initial trend found by Reviewer 2 was that some of the samples used in the calculation of the trend were close to the limit of the U-Th technique (samples >400 ka) and could be potentially older. To discount this as a potential cause of the linear trend, we re-dated samples between 400-500 ka close to the limit of the U-Th method also with the U-Pb method (x 3 samples). We found that two of these samples were older than the original U-Th dates between 400 and 500 ka. For example, the top section above the hiatus of sample SA05 was initially dated to 425 ka using U-Th, but when we dated this with U-Pb we calculated a much older age of 3200 ka. This was also the case for sample SA14 which was originally dated to 497.1 ka and later dated with U-Pb at 3180 ka. This has now been updated in our paper. We have re-evaluated all the U-Th data and have assessed that all the ages >400 ka in our dataset are too close to secular equilibrium to provide an age. Consequently, we do not report an age for these samples, rather we indicate them as 'out of range' but include the measurements in the Supplement 1 data file for completeness.

Additionally, we had to remove one sample (SA36) from our dataset as further dating and subsequent thick section analyses has revealed potential problems with recrystallisation. Although the data fits within the narrative of our paper, we have concerns that the isotope data may have been impacted by recrystallisation processes and therefore may not be original and reflect the isotopic composition at the time of formation, leading to potentially spurious results. We initially included this speleothem, despite our concerns, as to not "cherry pick" our material and include everything we have analysed, but upon further reflection and U-Pb and U-Th analyses, it is clear there has been some recrystallisation. We have not included it in the finalised dataset as further work is required.

- Sample size: unfortunately, there may be no quick solution for this issue if samples aren't readily available. To identify humid periods over this time scale with confidence, I think the sample size should be much larger (triple, quadruple, more?). The current 22 U-Pb ages were measured in one day, so if there are more samples, then the nowadays' rapid laser ablation

dating techniques would allow to quadruple the data set over three nights. This, of course, also depends on the authors' timeline and financial means.

As per Reviewer 2s comments, we have expanded our dataset and measured an additional 20 U-Pb ages, which has almost doubled the number of ages in our original dataset. We now have a total of 39 U-Pb ages. Due to time constraints and machine breakdowns further analyses were not feasible. This number of U-Pb ages is consistent or arguably greater than recent publications of speleothems in high-impact journals (i.e. Pickering et al.¹⁷, *Nature*, 2019 (n = 29); Engel et al.,¹⁸ *Geology*, 2020 (n = 19), Weij et al.,¹⁹ 2022, *Nature Communications Earth and Environment* (n =13)). We particularly also note that we used a combined approach of U-Th and U-Pb, including 33 U-Th dates and additional solution measurements of $^{234}\text{U}/^{238}\text{U}_{\text{initial}}$ to constrain the correction of our U-Pb ages. This combination makes this summation of work considerable. We have additionally added a further 4 speleothems to our dataset.

- - **I think the current U-Pb data set is too small to interpret the speleothem ages as discrete humid periods with enough confidence and therefore cannot strongly be linked to faunal dispersals. However, the interpretations based on the oxygen isotopes and fluid inclusions are very interesting and could stand on their own. The authors could decide to leave out the part regarding humid periods linked to faunal dispersals, refocus the manuscript on the palaeoclimatological aspects and resubmit this data – whether to Nature or to a different journal is at the editor's discretion.**

From the comments here as well as those from Reviewer 3, we have significantly removed and toned down our comparisons and inferences of the 'humid periods' identified in our data and the direct relationship to faunal dispersals. Further information regarding specific changes can be seen in our response to Reviewer 3's comments below on pages 28-35. We have also removed the "Central Arabia Humid Periods" identified in our original submission and the phraseology of these being discrete phases. We have refocused our discussion towards the palaeoclimatological implications as per the suggestions from Reviewer 2 and 3.

General comments:

Line 87: "*dispersals out of Africa*", preferable to use different wording, for example "*dispersals from Africa to Eurasia*", or something similar.

We have changed this as suggested.

Line 174: "...earliest evidence of hominin dispersals from Africa to Eurasia...."

Lines 127-129 "*desert speleothems*" and "(1) regional precipitation greater than ± 300 mm/a": not sure if this hard number of 300mm/year is necessary and whether this generalization is true for all deserts across the globe. This number was based on a study in Israel, how sure are the authors that this number also applies to Saudi Arabia? And other deserts? It also depends on the seasonality of rain fall, for example, if it mostly rains in summer, then most of it will evaporate. I suggest reframing this statement and perhaps restricting this statement either to central Arabian deserts, or using a more generalized statement saying something along the lines that sufficient regional precipitation is required.

We have made this a more generalised statement now, stating that sufficient rainfall is required and clarifying that previous studies have put this in the order of 300 mm/a.

Lines: 274-274 "... (1) sufficient regional precipitation, with previous estimates based on the distribution of active and inactive speleothems in the Negev Desert being in the order of ~ 300 mm a⁻¹ ..."

Lines 137-138: while it is true that U-Pb dating of speleothems is less frequently applied than the U-Th chronometer (arguably, partly due to a preservation bias of older speleothems in the

geological record), there certainly have been other pioneering studies utilizing the speleothem U-Pb dating technique, published in high profile journals, e.g.: Polyak et al. 1998 (Science), 2008 (Science); Meyer et al. 2009 (QSR), 2011 (Geology); Bajo et al. 2012 (Quaternary Geochronology), 2020 (Science), Pickering et al. 2018 (Nature), Engel et al. 2020 (Geology), Weij et al. 2022 (Nature Communications Earth & Environments). The current number and choice of citations here doesn't seem to do justice to the years of work the abovementioned studies have put into U-Pb dating of speleothems. The manuscript's chronological data set is novel, but the dating technique is itself not; it is well-established and has been routinely applied.

We appreciate there has been a lot of work on U-Pb dating of speleothems in the last decade and there are well-established methods. However, we disagree that U-Pb dating is "routinely" applied. This can certainly be argued for U-Th dating, which is currently *the* routinely applied method for speleothems, but not U-Pb in the wider community. We have amended the phrasing to reflect this and instead noted that this potential has only more recently been exploited this potential in speleothems. Further, we added several extra references as the reviewer requested including Bajo et al., 2020 and Weij et al., 2022.

Lines 282-284: "...Although the uranium-lead (U-Pb) chronometer is regularly used for precise age determinations in other archives, its routine application has only been recently exploited in speleothems^{8,19,21-23}..."

Line 142: are these 76 ages from 76 different growth phases or do some growth phases within a speleothem have multiple ages?

In many cases the ages are from different growth phases. During our selection of speleothem sections, our strategy was to sample as many different growth phases as possible. Multiple dates from a single speleothem were conducted particularly if there were clear hiatuses separating different phases (for example SA09, which growth intermittently from the Early Pliocene to Mid-Pleistocene). This is clearly outlined in Supplementary 1.1.

Lines 146 Figure 2 panel b "African palaeo-environments": perhaps a better label would be "hominin evolution" or something similar.

This has been changed on Fig. 2 as suggested.

Line 154: First Appearance Dates in figure 2, does each bar represent an age range of FADs reported in different studies from different sites?

That's correct, it represents the age date and uncertainty (or age range). We have removed this FADs from this figure due to comments from Reviewer 3, so this is no longer applicable.

Line 158: please define red diamonds and yellow squares as in figure 3.

Thanks for spotting this omission, we have defined this in the Fig. 3 caption.

"This study; U-Pb derived ages (filled circles) and U-Th derived ages using LA-MC-ICP-MS (filled diamonds) and solution MC-ICP-MS (filled squares). Ages of gypsum 'crust' samples indicated by *."

Lines 159-160: I appreciate the attempt to account for varying age uncertainties, however, since the resolution of the identified humid periods are similar within the U-Th and U-Pb domains, it might be better to keep a similar bin width for the U-Th ages. That way it's more obvious for the reader that the data density within the U-Th domain is much higher and it won't really affect the conclusions drawn from this figure.

Thanks for the comment. We have changed the bin width of the U-Th ages to the same number as the U-Pb (300 ka) as suggested (see revised Fig. 2).

Lines 160-161: I suggest removing the 2-point binomial smoothed average line for the U-Pb ages unless the U-Pb age errors is considered. However, I suspect that when the errors are taken into account, it will result into quite large uncertainties in the timing of the peaks.

We have removed the binomial smoothed average line as suggested (see revised Fig. 2).

Line 171: if this period is resolved at glacial-interglacial scale, what implications does that have for the previous humid periods, for which less data are available? Possibly, there were many more speleothems growing then, especially with a progressive aridity trend over time, but due to a preservation bias, we don't find these older speleothems as readily available as younger speleothems.

As mentioned above in our responses on page 10, we are aware of the preservation bias of speleothems whereby 'natural attrition' means that in some cases you are less likely to see speleothem preservation in deeper time as the caves may have experienced downward erosion of the karst terrain, cave collapse, in-cave erosional processes, in-cave sedimentation and speleothem precipitation covering stalagmites. However, as mentioned above, we see the opposite in our trend in our field site, whereby more speleothem material by volume has been deposited in the late Miocene and Pliocene, than more recently. We suggest that dryland environments do not experience high rainfall erosivity like the tropics, thus may be more suitable for speleothem archive preservation in deeper time similar to other sites with a similar phenomenon like the Nullarbor and the Negev Desert.

We have removed the words "two discrete CAHPS" and identified that there are humid period centred around the mean U-Pb ages:

Lines 339-457: "Further speleothem humid episodes were identified in the Early Pleistocene, centred around 2.1 and 1.0 Ma (Fig. 2)."

Lines 177-178 "*despite increased recharge to groundwater reservoirs, only a slight increase in local precipitation occurs*": this statement seems a bit premature here or needs more explanation. Based on what information could the reader know that recharge to groundwater reservoirs was increased, and that local precipitation only increased slightly?

We have clarified the text to explain the differences in formation mechanisms of gypsum versus calcite. Whereas both require water, calcite forms via degassing and therefore requires a CO₂ and a soil zone for dissolution of limestone to occur. Gypsum requires far less water and is mainly a result of supersaturation driven by evaporation, therefore it is uncommon to see both forming at the same time as they have different formation mechanisms.

Lines 463-470: "...A switch to gypsum and calcite precipitation occurred over 0.529 – 0.062 Ma with only gypsum overgrowth found after MIS 7, covering older calcite speleothem layers (SFig. 10). Unlike calcite which forms via degassing and typically requires vegetation and a soil zone for the formation of carbonic acid which dissolves CaCO₃, gypsum speleothem formation is mostly controlled by evaporation and typically occurs in arid and semi-arid regions. This suggests that despite increased recharge to groundwater reservoirs, only a slight increase in local precipitation occurred, which was insufficient to promote calcite speleogenesis²⁴ in central Arabia."

Line 181: please define *aridity*: are we looking at precipitation, soil moisture, humidity, run-off, precipitation-evaporation? There are many definitions, and they all mean something slightly different in terms of wetness vs. dryness.

We have clarified as follows:

Line 472: "A progressive aridity trend with decreasing water availability..."

Line 182: I suggest referring to figure 3 here already so that one can immediately look at the values when reading a statement like this.

We have added the reference to the figure as suggested:

Line 474: "...towards higher values (Fig. 3)."

Lines 190-198: I appreciate this link to speleothem fabric – often overlooked even though it can convey important information about drip rate and water supply, which are ultimately linked to the hydroclimate above the cave.

We appreciate the comment.

Line 198 "*caused by a decrease in precipitation*": and what about the effect of evaporation?

We have changed this to "reduced effective groundwater recharge" as this encompasses both precipitation and evaporation.

Line 513-515: "The synchronous shifts in $d^{18}O_{carb}$ and speleothem fabrics suggests reduced effective groundwater recharge in the Arabian interior towards the Late Pleistocene."

Line 201 Figure 3 panel d "*increasing aridity*": what is this blocky line? It now looks like the thickness of the little blocks are related to the peaks in dust, but it's probably just the line pattern? I suggest using a thinner line with a clearer arrow.

We have changed the arrow as suggested. There is now a thinner solid line with a clearer arrow. See amended Fig. 3 below:

Line 213: please add a similar description of the U-Th ages to figure 2.

We have already made this amendment in a previous comment above.

Line 234: “Fig 4”: Are you referring to panel b here? Would be useful to add that to the figure reference. “modern meteoric waterline derived from a tropical moisture source (Indian Ocean)”: which one in your figure 4?

Changed as suggested:

Line 566: “...(Indian Ocean) (Fig. 4b)...”

Line 235 “26 degrees N”: is this where the study site is? It looks like it in your figure 1. If so, I suggest adding a few words to this sentence to clarify that.

Changed as suggested:

Line 567: "...reached our study area currently at 26° N..."

Lines 251-252: I think it reads better if you refer to figure 3 after "during periods of lower meridional SST gradients", otherwise it reads like there should be a plot of the global climate model simulations.

Changed as suggested:

Line: 595 "...During periods with lower meridional SST gradients (Fig. 3).."

Line 254-255: we are now in the "glacial-interglacial world" when dust transport was mainly increased during glacial periods. For long, the palaeoclimate community believed that an increase in dust meant more arid conditions. However, Muhs (2013), for example, also showed that other factors play a role: increased source areas due to more continental exposure and more shrub-like vegetation, stronger winds, greater production of dust-sized particles. Therefore, I'm not entirely convinced that increased dust transport from the Sahara Desert is strong evidence the Saharo-Arabian aridification.

We appreciate this comment. We agree that the dust fluxes are controlled by a combination of continental aridity (source) and atmospheric winds (transport). For our interpretation of the marine records we have leaned on the interpretation of the authors of Crocker et al.²⁵ for their marine record; they state that "The transition to stage III, which covers the past ~2.25 Myr (early Pleistocene to Recent) of African climate history, is characterized by a shift towards more arid conditions (decreasing[Al + Fe]/[Si + K + Ti] centred around 3.1 Ma) followed, ~400 kyr later, by increased mean values of ln[Zr/Rb] and dust fluxes (which also show higher amplitude variability), closely contemporaneous with the intensification of glacials as revealed by benthic oxygen isotope records (Fig. 1b). The highest dust fluxes in our records during stage III are consistent with the suggestion that the growth of large continental ice sheets in the Northern Hemisphere promoted the development of more arid and dusty conditions on Africa through steepening latitudinal temperature gradients and strengthening winds".

We have clarified this extensively and made a clearer distinction between what the marine records are showing (dustier glacials) with what our speleothem data have revealed, which is progressively drier humid periods (interglacials) over the last 8 Ma, particularly after the onset of the Pleistocene and mid-Pleistocene transition. As our speleothems effectively select for only the humid phases, we have highlighted the significance of this finding which suggests aridification or drier wetter phases through time. We have clarified further the multiple factors influencing the dust record.

Lines: 605-616: "Changes in atmospheric circulation patterns in Saharo-Arabia are also evidenced by increased dust transport from the Sahara Desert from 2.3 Ma²⁵ (Fig. 3d) and greater terrigenous input into the Arabian Sea²⁶. Enhanced dust transport has been attributed to the Pleistocene intensification of glacials, associated with increased aridity, less vegetation, lower soil moisture, changes in atmospheric wind direction and intensity coupled with lower precipitation²⁷. The production of fine-grained sediments from source areas such as Mega Lake Chad during interglacials is followed by desiccation, deflation and their transport in glacials, acting as dust-producing 'hot spots'²⁵. Importantly, however, as our speleothems are selective records of interglacial humid intervals (opposed to glacial aridity), the drying trend observed in the speleothem record may occur in tandem with higher dust transport in glacial phases. Further, during the globally warmer and higher atmospheric CO₂ world of the Pliocene and Late Miocene^{28,29} the wetter humid episodes in central Arabia are likely associated with lower wind speeds, more vegetation and higher recharge."

Line 274: Figure 4 panel a and c “Fossil Water Central Arabia”: where is this plotted in panel a and c?

We have updated the caption as suggested.

Lines 696-706: **“Fig. 4. Isotopic composition of central Arabian speleothem fluid inclusion waters.** Panel a: Late Miocene-Pleistocene isotope composition of fluid inclusion fossil water within the speleothem calcite lattice compared to Riyadh precipitation waters, back trajectory analysis indicating rainfall sources which originate from the Arabian Gulf (AG), Mediterranean Sea (MS), Arabian Sea (AS) and Red Sea (RS) and the weighted mean rainfall for Riyadh (yellow diamond). Meteoric water lines are presented for the Global Meteoric Water Line (GMWL), the Riyadh Local Meteoric Water Line (LMWL) and coastal Bahrain LMWL (further details are provided in the Methods section). Panel b: Modern LMWLs for northern (N) and southern (S) sources for Arabia compared to fossil water isotope data from speleothems from Hoti Cave, Oman (open symbols)³⁰, from Mukalla Cave, Yemen (open symbols)²¹ and this study (filled symbols). Panel c is the central Arabia speleothem $\delta^{18}\text{O}_{\text{FI}}$ fossil water with respect to time.”

Line 325 “the aforementioned absence of speleothems”: again, how sure are you that this absence is statistically true?

We have already addressed this in a previous comment on page 10 of this document.

Line 551: are those 28 sub-samples for U-Th solution work? From how many speleothem samples? And are they from distinct growth phases separated by hiatuses or are some growth phases dated multiple times?

Yes. We analysed 68 U-Th samples, including by solution (46) and laser (22). This is clarified in the text:

Line 1049-1050: “A total of 68 U-Th measurements (46 via solution and 22 via laser ablation) were determined for to produce 35 ages for 22 speleothems (Extended Data File 1)”.

The specific sample locations are indicated in diagrams in Supplementary 1.1. and data with depth and speleothem specimen information in Supplementary Data File 1.

Line 559: what is the error on the $^{232}\text{Th}/^{238}\text{U}$ weight ratio of 3.8 and was this error propagated to the final age uncertainty?

The assumed error in the $^{232}\text{Th}/^{238}\text{U}$ weight ratio of 3.8 is - as usually - 50 % (1.9), and the uncertainty is propagated to the final age uncertainty. We have clarified this in the text as follows:

Lines 1059-1060: “and corrected for detrital contamination assuming a $^{232}\text{Th}/^{238}\text{U}$ weight ratio of 3.8 ± 1.9 (50%) for the detritus which is propagated into the final age uncertainty and..”

Line 562: again, from how many speleothem samples? And are they from distinct growth phases separated by hiatuses or are some growth phases dated multiple times?

This has been addressed previously in a response to reviewer 1. We have dated multiple segments to capture all the growth phases. For example, sample SA09 has ages: 0.360, 2.188, 2.290, 4.098 Ma. To make this clearer we have colour-coded the U-Pb ages and corresponding isotope box plots in Fig.3. Further, these ages are clearly outlined on diagrams of the speleothem thick sections in Supplement 1.1. Additionally, we have added Supplement 1.1 reference here as well.

Line 1075: “...diamonds; Extended data File 1; Supplement 1.1)...”

Lines 575-576: see comment about line 559.

We addressed this in the previous text in line 599 and have referenced that the same correction was used here.

Lines 1086-7: “Detrital contamination was corrected for as described above for the solution-based ²³⁰Th/U-dating.”

Lines 582-583: from the supplementary file I gather that this is done at Frankfurt University, but please specify where the analyses were done.

Added as requested.

Line1094: “...ICP-MS) at FIERCE, Goethe University Frankfurt was used...”

Line 596: I assume the ages were calculated in Isoplot? Which fit type was used and could you briefly elaborate on the choice of fit type? This information should be added to the Methods.

This has already been amended due to a previous comment by Reviewer 1, Comment 7, pages 5-6.

Lines 599-600: please provide a plot of the initial ²³⁴U/²³⁸U ratios against age like I have done in the main comments.

We have included this whilst addressing the previous comments on page 11.

Line 605: I assume this is done at the MPIC but please specify where the an³¹alyses were done.

Added as requested.

Line 1121: “...preparation device at the Max Planck Institute for Chemistry, Mainz...”

Lines 610 and 614: I see the $\delta^{13}\text{C}$ data in the supplementary file, but it is not discussed in the manuscript. Could you briefly explain why not?

An in-depth analysis of the $\delta^{13}\text{C}$ data and the mechanisms controlling its variability is beyond the scope of this MS, but for completeness, as we have conducted the $\delta^{13}\text{C}$ and $\delta^{18}\text{O}$ analyses simultaneously, we thought it would be prudent to include this data in Supplementary data file 1. Speleothem $\delta^{13}\text{C}$ is frequently attributed to vegetation and soil processes, which is not the focus of this MS. Future work will include the analyses of pollen grains in the speleothems and compound specific carbon isotope data to deconvolve $\delta^{13}\text{C}$, giving us a more complete picture of the environmental and vegetation characteristics history in central Arabia. We hesitate to make any premature interpretations of the data at this point as $\delta^{13}\text{C}$ is a complicated signal, controlled by multiple different processes simultaneously (i.e. review by Fohlmeister et al.) and interpretations are often complex. We look forward to examining this more robustly in future work.

Line 625: again, I assume this was done at the MPIC but please specify in the Methods.

Added as requested.

Line 1148: “...Protocol⁷³ at the Max Planck Institute for Chemistry, Mainz...”

Supplementary Methods

Lines 173-174: I suggest referring to SFig 10 here.

Added as requested.

Line 217: "...interglacial phases within a given period and the frequency of humid phases (SFig. 10)..."

Lines 206-208: this could be true, but again I'm wondering about the effect of your sample size and speleothem preservation?

We have addressed this in previous comments on page 11 (preservation bias) and pages 14-15 and 19 (sample size).

Line 214 "the blue dashed line ... used as the cutoff for significant interglacial phases": please specify what significant means here. Is it statistically significant? Or is the position of this line the author's subjective choice? Please briefly explain.

Significant interglacial here are defined as land ice distributions from the binary glacial index calculated by Koeler and van der Wal³² where a glacial = 1 and an interglacial = 0. The dashed line at 0.4 indicates more i.e. more interglacial conditions than glacial. We refer to Koeler and van der Wal³² in the caption for a full explanation on the index calculation.

Lines 334-335: this is good to point out: the importance of understanding the absence of something. Here it is related to the hiatuses in animal movement, but more important for this paper is to first understand the meaning of the absence of speleothems. So again, please consider and discuss speleothem preservation and the sample size. Why are the authors sure that the speleothem growth phases are true?

We have addressed this in previous comments on page 11 (preservation bias) and pages 14-15 and 19 (sample size).

Supplementary U-Pb data

U-Pb calculated ages sheet

Footnote #1: please specify which software was used to calculate the ages and what fit type was used.

The DQPB software was used in the revised version of the MS from the reviewers recommendation and the spine fit was used. This has been updated in the footnotes in the "U-Pb calculated ages" spreadsheet.

"(4) Concordia equilibrium ages calculated using the DQPB software, using the spine algorithm and including decay constant uncertainties and the natural ²³⁸U/²³⁵U uncertainties."

Footnote #6: please also add the uncertainty of the initial 234/238 activity ratio here.

This has been added in a new sheet called "Initial U234_238 measured" in Supplementary data file 2. All the measured activity ratios and calculated initial ratios clearly marked.

Footnotes #7 and #8: I'm not following here. Where does this activity ratio of 0.885 come from and what is it used for?

We have since added more data and recalculated the average initial value, so this is no longer relevant.

Sample raw data sheet

I see several measurements were excluded. After replotting the data and examining the isochrons, I see that most of these excluded measurements are obvious outliers, however,

some are not. Could you please add a sentence explaining the reader why some measurements were excluded?

We have added an additional footnote to explain excluded data in Supplementary data file 2:

“^eDenotes excluded values from the age calculation including samples that were clear outliers and those where the ²⁰⁷Pb cps <0.”

Other comments:

The figures in the manuscript are well done, but could you please check the colour schemes for suitability for colour blind people?

We have made every effort to include both colour and symbol differentiation of data as well as labelled on specific proxy data on the figures according to guidelines for how best to present data for colour blind persons.

The supplementary files contain “raw data”, however, with “raw data” I’m thinking of the unprocessed numbers that come off the mass spectrometer. The data presented in the supplementary files are corrected for systematic and/or statistical uncertainties, so are technically not “raw”. I suggest using different wording here.

We have changed this to “^{U-Pb LA-ICPMS data pt1}” and “^{U-Pb LA-ICPMS data pt2}” in Supplementary data file 2.

Referee #3 (Remarks to the Author):

A. Summary of the key results

This paper presents a speleothem-based hydroclimatic record that documents recurrent humid phases in central Arabia over the last 8 Ma, potentially facilitating dispersals and biogeographical exchange at this crossroads between Africa and Eurasia.

We agree with this summary of our manuscript.

B. Originality and significance:

The discovery, dating, and analysis of the speleothems in the heart of the Arabian Peninsula in modern-day Saudi Arabia is highly significant. As the authors note, it fills a critical gap in our understanding of the paleoclimatic history of the Arabian Peninsula between the Miocene and the Middle to Late Pleistocene, a gap of some 5 million years during which many dynamic climatic and evolutionary developments were taking place in Africa and Eurasia, but of which we had no information for Arabia. The discovery of a trend of progressively greater aridity in the $\delta^{18}O$ data is also very interesting. I think this is an important discovery that is relevant for our understanding of paleoclimatic changes that not only affected Arabia, but the entire Saharo-Arabian belt, and faunal (including hominin) exchange between Africa and Eurasia during the late Neogene.

We thank Reviewer 3 for highlighting the significance of our research and agree that this is an important piece of work that is relevant for our understanding of paleoclimatic changes that not only affected Arabia, but the entire Saharo-Arabian belt as well as faunal exchange between Africa and Eurasia during the late Neogene.

C. Data & methodology

The speleothem data, and especially the U-Pb age dating, should be closely examined and reviewed by specialists in this field. The discussion and conclusions drawn regarding the interchange of large mammals between Africa and Eurasia, is however not very convincing and has some fundamental problems discussed in detail below. However I think it can largely be removed as the speleothem data is the main discovery here.

We thank the reviewer for their critical review of the interchange of large mammals between Africa and Eurasia in our discussion. We further agree with suggestion to largely remove this speculative discussion and focus on the speleothem and climatological interpretations. We have amended the manuscript accordingly with specific details for this below.

D. Appropriate use of statistics and treatment of uncertainties

In Fig 2, what are the error bars on the age samples in Fig. 2g? Are these 1 standard error, or standard deviation, or are they the 95% age confidence interval? I hope the latter, otherwise the actual 95% age estimates could end up spanning a million years or more. This is important because the humid periods in this figure are drawn against the maximum error bars of the individual age samples (which is in itself debatable). If the age uncertainty is too great, then there would be little statistical basis for arguing for discrete humid/arid periods.

These are 95% confidence intervals. This has been made clearer in the Supplementary data file 2 with the ages.

E. Conclusions: robustness, validity, reliability

The conclusions seem fine to me, with the exception of the claim that “Taxa with African affinities that appear in Asia, and vice versa, are considered to reflect dispersals between these two regions following periods of climate amelioration in the Saharo-Arabian Desert.” I think the analysis on which this conclusion is based is not credible for reasons discussed below. However, I also think it is also unnecessary to the main point of the paper, and removing (or significantly toning down) these claims will not detract from the importance of the paper.

We thank the reviewer again for their detailed and expert analyses of this section. We agree with these conclusions and have removed and significantly toned down all discussion regarding the dispersals of taxa through Arabia. We have amended the manuscript accordingly with specific details for this below.

F. Suggested improvements: experiments, data for possible revision

We have implemented the suggested improvements from Reviewer 3, please see our detailed responses below.

Abstract: “Wetter conditions correspond with patterns of mammalian dispersals between the Afrotropical and Indomalayan realms, highlighting the role of Arabia as a key crossroad for continental-scale biogeographic exchanges.” I do not think that you have demonstrated (or can with such small sample sizes, especially given the total lack of an Arabian record during the Pliocene – Early Pleistocene) that the identified humid intervals correspond to times or ‘patterns’ of dispersal across Arabia. Furthermore, it’s not clear why you limit this to just Afrotropical and Indomalayan realms instead of all Eurasia. However, I think you are perfectly justified in changing this to “Wetter conditions could have facilitated mammalian dispersals between Africa and Eurasia”. The record is simply not good enough to establish otherwise at the moment.

Thank you for highlighting this point. We have changed the last sentence of the abstract accordingly.

Line 156-157: “Wetter conditions likely facilitated mammalian dispersals between Africa and Eurasia, with Arabia acting as a key crossroad for continental-scale biogeographic exchanges.”

Line 78: The Saharo-Arabian belt, even when extended to the Thar Desert, has little to do with the boundary of the Indomalayan Realm, which is instead more significantly bounded by the Tibetan Plateau.

We are citing previously published literature here about the distribution of biogeographic realms and state the neighbouring realms to the Saharo-Arabian transition zone. We reference existing literature that establishes Arabia as a location which hosts complex faunal admixtures that exhibit African,

Eurasian, and South Asian affinities. We have toned down the emphasis on dispersals and the transition zone and removed this from Figure 1 as a focal point. We have redrawn Figure 1 to focus on the climatic interpretations of our results as suggested.

Also why is your reference (ref 10) for biogeographic realms Kreft & Jez's 2013 comment on the Holt et al. 2013 Science paper? I think you should either be following Kreft & Jez or Holt et al. 2013. Same for Fig. 1b.

The reference is for the Kreft and Jez's³³ comment on the Holt et al.,³⁴ paper as they describe the distinction and importance of the Sahara-Arabian transition zone, showing that this geographic region is a permeable zone, rather than a straight line between zones like in Kreft and Jez³³. It represents the natural gradation between different biogeographic realms and reflects that during different climate phases there be fluxes across this area. We have removed Fig. 1b to address Reviewer 4's comments about shifting the focus away from dispersals and towards paleoclimate.

94-96: Actually there is evidence aridity might not have increased in these savanna habitats. A drop in CO₂ is another main hypothesis for the spread of C₄ grasses, which might have nothing to do with desertification. See Blumenthal et al 2017 (also discussed by Faith et al 2018, eg their fig. 2A and 2C).

We agree and have already amended this due to a previous comment from a Reviewer above. We also note that the CO₂ mechanism is also contentious as Osbourne³⁵ also points out that increasing seasonality of rainfall is linked to changes in the relative abundance of grasses.

Lines 191-203: "...Most strikingly, ecosystems transitioned from those dominated by plants following the C₃ photosynthetic pathway (e.g., trees, herbs, shade-loving grasses) to those dominated by C₄ plants (e.g. arid-adapted grasses)³⁶ (Fig. 2). **Widespread aridification instigating C₄ grass dominance in low and mid-latitude open environments** has traditionally been linked to Quaternary cooling in the Northern Hemisphere (NH), triggered by the onset of glacial-interglacial cycles, which intensified at ~2.6 Ma³⁷. However, recent research suggests drying may have begun much earlier, with fully arid conditions in the Sahara from 11 Ma²⁵ and hyper-aridity in the northern Arabian margins starting at 9 Ma³⁸. **Further, increasing distributions of arid-adapted vegetation and, consequently, C₄ mammalian grazers, may be a result of C₄ plants outcompeting C₃ in many parts of the world under decreasing atmospheric CO₂ levels where C₄ photosynthesis is physiologically advantageous³⁹. This leaves the driving mechanisms in what appears to be a key Plio-Pleistocene aridity trend unresolved.**"

Also here and in Fig 3 you suggest there was an 'intensification' or an 'abrupt onset' of ice at around 2.6 Ma. You cite Westerhold et al. 2020 but this paper does not appear to state this so dramatically, but rather only confirms that ice sheets were in place by the Plio-Pleistocene boundary. The last 20 years of paleoclimatic data have more or less done away with the idea of a rapid or sudden onset of glaciation around 2.8-2.6, which rather seems to have been replaced by a more gradual model. See Trauth et al. (2021) for a review.

In figure 3 and the supporting caption we have replaced 'intensification' and 'abrupt onset' with "Gradual Northern Hemisphere Glaciation" with a box indicating this process spanned 3.5 to 2.5 Ma according to the suggested reference "Trauth et al., 2021", which we have additionally added.

101: Is it really larger than the sandy parts of the Sahara?

The Empty Quarter/Rub al-Khali is the largest area of continuous sand in the world. We recognise that the Sahara Desert, however, is some 14 times the size of the Empty Quarter/Rub al-Khali. We have clarified this in the text to avoid any confusion.

Line 206: “ ..the **Empty Quarter** (the largest **area of continuous** sand on Earth)..”

106: Levant is Mediterranean. Not ‘influenced’

We have removed the word “-influenced” according to the reviewer comment.

Line 211: “...and in the Mediterranean Levant..”

109-110: ‘vicariant agent’: what is meant by this? If you mean climate changes geographically splitting/isolating species populations and thereby stimulating speciation, then that is quite different from patterns of biogeography and dispersal, and is not something that is followed up in this paper.

We have decreased the emphasis of paleo biogeography as requested above and as a result we have removed the words ‘vicariant agent’ and replaced it with the following:

Lines 212-214 “...The lack of direct hydroclimatic information from the continental interior **prior to the Middle Pleistocene**, renders the climatic modulation of the Arabian Peninsula and **the degree of permanence of its hyper arid interior** largely unknown.”

Fig2: 2d nicely shows the lack of a terrestrial vertebrate record (note ‘fauna’ also includes things like marine invertebrates, which is not what you mean here) between Baynunah and Nefud times. 2e and 2f however are problematic. First it is not clear why you chose to focus only on the large mammal record. If you are trying to establish Arabia as an arena of faunal dispersal, this should include a large-scale analysis of a wide range of taxa (what about fish for example, which provide a much better view of hydrographic networks?). Second, almost all the taxa discussed have very poor fossil records. Using a lack of data to speculate that these taxa dispersed during the identified humid periods is not convincing. This is not even to mention the dating uncertainties around many of these records. Vrba may have used this approach in 1995 to establish a hypothetical framework to be tested (the ‘traffic-light model’), but this is hardly acceptable today and biogeographic comparisons should be much more quantitative. Additionally, to test your hypothesis that dispersals occurred across Arabia during specific humid periods, your need a much more resolved fossil record, one with an high resolution of first appearance datums for taxa in Africa, Arabia, and Eurasia, and which shows convincingly the chronological origins of taxa in one place, and their steady dispersal elsewhere (e.g. hominins in Africa -> Dmanisi -> Europe and Asia; or equids from North America to Eurasia). Since there is no terrestrial record between ~7 and 0.5 Ma in Arabia, that is simply not possible and the evidence is entirely circumstantial. In fact, determining locations of origin and directions of migration (especially for a genus or species) is exceedingly difficult even when using some of the best fossil records. You don’t know if taxa dispersed through the Arabian Peninsula, or only along its edges (eg the Levant). It could also be that during humid periods Arabia was populated by taxa from nearby regions, without acting as a path for continental dispersal. Your review of the fossil record does not distinguish among these scenarios. For example, the fossil fishes of the Baynunah Formation provide evidence for a scenario of repeated colonization of Arabia, suggesting it could also have functioned as a kind of cul-de-sac (Forey & Young 1999; Otero 2022).

We thank reviewer 3 for the detailed comments and summary here. We have significantly reduced the text regarding dispersals through Arabia (see manuscript Lines: 713-789 and Supplement 1 Lines: 359-511), instead highlighting the region as a possible region where intercontinental dispersal could have taken place in light of our new climate record. We agree that the fossil record is problematic due to the poor age constraints and the lack of fossils over periods of time like the Pliocene in Arabia.

In Fig. 2 we have removed all the FADs and any discussion of these in the text. Instead, we focus on climate change and its role in strengthening/weakening faunal interconnectivity between Africa and Eurasia, such as the breakdown of the ‘Old World Savannah Biome’ and subsequent growing endemism in Africa.

In the text where we discuss the Baynunah Formation fossils, we have also included the reference for evidence of a 'cul-de-sac'⁴⁰.

Lines 724-728: "...The rich diversity of bovid and giraffid fossils which exhibit mixtures of Afrotropical, Indomalayan (Siwalik), and minor Palearctic affinities, and the broad absence of endemic species, substantiates Arabia as a dispersal corridor or mixing cul-de-sac⁴⁰ between Asia and Africa⁴¹. It also suggests a level of selective filtering between neighbouring biogeographic regions⁴², driven by the frequency of humid episodes in the Saharo-Arabian Desert..."

Furthermore there is probably nothing to keep fauna from dispersing through the Levant or along coasts and river networks during arid periods. Dispersal between Africa and Eurasia need not be restricted to Saharo-Arabian humid periods. It might have been more interesting to speculate to which degree the proposed arid periods could have acted as barriers to dispersal, or entirely prevented the establishment of faunas in Arabia. Since the Levant-Mesopotamian region is probably more important for the passage of faunas between Africa and Eurasia than the interior of the Arabian Peninsula, an arid Arabia might not have mattered much in many instances. All these general points should be considered in your framework.

We do not suggest that movements were "restricted" to Saharo-Arabia but nonetheless agree with the reviewer that movements may, and probably did, take place along other routes such as through the Levant-Mesopotamian region. Commenting on which routes were more important is at present difficult given the fossil record, and it is possible the the Saharo-Arabian region represented an as important, or perhaps even more important, area for faunal exchanges. Nonetheless, following the reviewer's suggestions here and below, the arguments have been reframed to present the Arabian Peninsula as a possible place of bigeographical exchanges during the Neogene and Quaternary.

There are additionally further problems in the details: The taxa chosen for the biogeographic analysis are largely poorly known and cannot be used to build a story of biogeographic exchange at specific points in time. As mentioned, Vrba did this in the 1980s and 1990s, but that was in a different time and context. For example:

Antilope subtorta: this is actually Antilope aff. subtorta. The aff. Indicates this is not the same species as A. subtorta from the Siwaliks. This is furthermore represented only by two horn fragments from Member C in the Shungura Formation (Gentry 1985), which is around 2.6 Ma. The dashed line labelled 1 in Fig 2 makes it look like this is a well-established species between 2.8 and 2.5 Ma. This is misleading and in fact we know so little about the Shungura Antilope that is it not possible to know when and from where its immediate lineage originated, or how long it survived in Africa (is its poor record due to its short species lifespan or the result of rarity and poor preservation?). The same is true for Makapania broomi, as we have no idea what other Caprini it is related to, and cannot even guess as to when and from where its lineage entered Africa. It is likely that Caprins were well established in Africa and evolved there, but only rarely made it into the fossil record due to habitat preferences in places that were non-depositional (e.g. mountainous regions). The same is true for duikers (Cephalophini) which are widespread in African forests, but have almost no fossil record (same for chimpanzees and gorillas). Continuing down the list, it should not be a surprise that almost all the "Endemic Eurasian fauna in the Afrotropics" are poorly known taxa. Parantidorcas latifrons, Budorcas churcheri, etc. Bouria angettyae and probably Nitidarcas asfawi, though both described as a Caprini, are almost certainly Alcelaphini, and I wouldn't advise hanging too much on these being caprins (Gentry 2010 hinted at this). Also, as already noted in this manuscript, it is just as likely that African fossil Caprini were parts of locally-evolving lineages and not episodic and short-lived arrivals from Eurasia. Fossil Caprini are simply so rare in Africa that is not possible to sketch out a convincing biogeographic story. Ancient DNA / proteomes may one day address this question.

We thank the reviewer for their detailed and informative response here. We have removed the fauna FAD's from the figure and have significantly reduced the related text in the SI, which should satisfy most of the reviewers concerns. We have retained a brief qualitative analysis in the SI. This is not meant to be exhaustive, as the article's focus is on the speleothem record, but to simply to demonstrate to the

readers that African-Eurasian faunal exchanges did take place during the Neogene and Quaternary and that conditions in Arabia would have at times been favourable for such exchanges.

Regarding *Antilope sub torta*, we have amended the text to reflect this *affins* taxonomic assignment and uncertainties around its origin. According to McDougall and colleagues (2013), the Shungura Member C dates to between ca. 3.0–2.6 Ma based on $^{40}\text{Ar}/^{39}\text{Ar}$ dates on from Tuff C at the base of Member C at Shungura and the Burgi Tuff from Koobi Fora and we retain the original (though slightly amended) dates. The amended text now reads:

Supplement Lines 421-423: "...and potentially another antilopin which is recorded in the Late Pliocene Shungura Member C deposits, Ethiopia, dated to ~3.0–2.6 Ma^{43,44}, and may be closely related to the South Asian *Antilope sub torta*"

Regarding members of Caprini (i.e., *M. broomi*, *Bu. churcheri*, *Bo. Anngettyae*, *N. asfawi*), their mention is now restricted to the brief discussion of Vrba's 'traffic light' hypothesis, which, as some more recent scholars have suggested (e.g., Bibi, 2011), may be supported by taxa like *Makapania broomi*. We had already mentioned the possibility of *in situ* speciation among Caprini within Africa and have now added a short bit highlighting just how poor the fossil record is in some cases:

Supplement 1 Lines 456-458 "Poor dating and a patchy fossil record makes constraining the timing of these movements difficult, with taxonomic uncertainties and ambiguities complicating matters even further."

Following the reviewer's suggestion, Arabia as a region of biogeographic exchange is now presented as a hypothesis to be tested, and we summarize as follows:

Supplement 1 Lines: 506-511: "In summary, it appears that many of the abovementioned taxa would have been well-suited for life in Arabia during the wet interglacials phases identified in our speleothem record. The scant fossil record aside, we suggest that Arabia likely acted as a hitherto unrecognized but important crossroad for biogeographic exchange between Africa and Eurasia over the past 8 Ma. The precise nature of these exchanges, and the exact role of Arabia in these, may only be elucidated with an improved fossil record, better dating, and phylogenetic studies using ancient DNA and proteomics."

Also note that Vrba et al. 2015 found that the South Asian Reduncini were monophyletic and assigned to *Sivacobus*. They discuss possible relationships to the African *Kobus/Dorcadoxa aff. porrecticornis*. It's not a simple matter of the latter taxon being an immigrant from Eurasia into Africa, or as claimed in Fig 2, of *Sivacobus* being an Afrotropical taxon in origin. There is no African *Sivacobus*. Also, even earlier African Reduncini are probably also present at Chorora (Suwa et al. 2015).

We have removed mention of *Sivacobus*.

On the other hand, other examples / literature are missing. E.g. Boissarie & White (2004) describe an Asian hippopotamid in Ethiopia at about 2.5 Ma. Again here though, the 'first appearance datum' for this taxon is basically the only appearance datum there is. Such singleton (or single-interval) taxa are highly problematic and are actually usually left out of large-scale turnover analyses (eg Alroy 2010, or see Bibi & Kiessling's 2015 discussion of this issue – Vrba's turnover pulse at 2.8 Ma is basically based on singleton taxa). As such they should probably not be relied on for testing of large-scale biogeographic hypotheses either.

Again, the analysis was not intended to be exhaustive, though we have now included the hippopotamid example, as hippos are becoming increasingly well-represented in the Arabian fossil record:

Supplement 1 Lines: 438-439: "while the hippopotamid *Hexaprotodon bruneti*, with strong affinities to the South Asian hexaprotodons, was discovered in the Middle Awash of Ethiopia, dating to ~2.5 Ma (Boissarie and White, 2004)."

Additional questions continue for other taxa. For example, you list *Serengetilagus* as a

Eurasian taxon, but I think you mean Leporidae generally. I believe *Serengetilagus* is not known from outside Africa. Also many of the taxa listed as dispersing from Africa to Eurasia are highly problematic, more so when one considers the lack of good dates for the Tatrot/Pinjor faunas.

Yes correct. We have amended the text as follows:

Supplement 1 Lines: 435-436: “Additionally, leporids make their first appearance in Africa at Late Miocene sites in Ethiopia and Chad, dating to ~7.5–6.8 Ma (Lopez-Martinez et al., 2007; Suwa et al., 2015).”

However, I think a solution is easily found. I do not believe the review and speculation on the sparse large mammal record here adds anything significant to the manuscript. It could be left out entirely. The main finding of speleothems indicating humid periods between the Miocene and Pleistocene is the main finding. This evidence clearly implies that conditions could have been appropriate for the establishment of faunas like those of Baynunah and Nefud in Arabia at several intervals during the Pliocene and Pleistocene. It’s fair to propose that these humid periods might therefore have played an important role in expanding faunal exchange between Africa and Eurasia, even though this cannot be conclusively demonstrated with the current record (or without a much larger and more complex analysis).

We thank again Reviewer 3 for their detailed and critical appraisal of our terrestrial vertebrate taxa data. Following their suggestions, we have removed the analysis of the FADs and drastically reduced the text with regards to dispersals. We have retained a brief discussion of the fossil record in the SI to highlight to readers that faunal exchanges between Africa and Eurasia did occur during the Neogene and Quaternary, opening the possibility that the Arabian Peninsula may have played a key role in some of these movements.

296-308: a summary of the Baynunah fauna concludes that the environment was a wooded savannah there as at Toros Menalla in Chad. It’s not clear what this is supposed to indicate, or why the mention of *Sahelanthropus* here. ‘Wooded savannah’ represents such a diversity of habitats, and is basically a generic label for anything that is not a forest or pure woodland or grassland, and is therefore highly non-specific with regards to most taxa (including hominins).

We have clarified that we are talking about wooded savannah in the broadest sense, in comparison to today’s environmental context where vegetation is xeric/grasslands. We have clarified below:

Lines: 728-735: “...Further, the Baynunah Formation also contains the earliest primate guenon (old world monkeys of the tribe Cercopithecini) outside of Africa⁴⁵, suggesting an environment comprising savannahs and gallery forests⁴⁶. This is similar to the inferred environments at Lake Chad, central Africa, over a similar time interval, where the oldest known hominin remains (*Sahelanthropus tchadensis*, dated to ~7 Ma⁴⁷) have been recovered. In the Saharo-Arabian Desert, periods of the Late Miocene likely received enough precipitation to support vegetation suitable for hosting primates, where today this is lacking, with our data suggesting that subsequent humid episodes were drier.”

An additional point. Regarding the cave names (1000 wings; Mossy; Friendly; Murrubeh; Scerake; Luxury; Hotel) – it seems strange that caves in an Arabic-speaking country were given English names (presumably with the exception of Murrubeh). I understand the need for convenient names while conducting fieldwork, but since you are here basically formalizing and fixing these names in print, would you consider giving them Arabic names? Perhaps these could simply be translations of the given English names (and you could retain the English names in brackets).

We have updated this to the Arabic names as suggested.

G. References: appropriate credit to previous work?
Seems fine.

Thanks.

H. Clarity and context: lucidity of abstract/summary, appropriateness of abstract, introduction and conclusions
Yes all fine.

Thanks.

- Alroy, J. 2010. Fair sampling of taxonomic richness and unbiased estimation of origination and extinction rates. *Quantitative Methods in Paleobiology. Paleontological Society Papers* 16:55–80.
- Bibi, F., and W. Kiessling. 2015. Continuous evolutionary change in Plio-Pleistocene mammals of eastern Africa. *Proceedings of the National Academy of Sciences* 112:10623–10628.
- Blumenthal, S. A., N. E. Levin, F. H. Brown, J.-P. Brugal, K. L. Chritz, J. M. Harris, G. E. Jehle, and T. E. Cerling. 2017. Aridity and hominin environments. *Proceedings of the National Academy of Sciences* 114:7331–7336.
- Boisserie, J.-R., and T. D. White. 2004. A new species of Pliocene Hippopotamidae from the Middle Awash, Ethiopia. *Journal of Vertebrate Paleontology* 24:464–473.
- Faith, J. T., J. Rowan, A. Du, and P. L. Koch. 2018. Plio-Pleistocene decline of African megaherbivores: No evidence for ancient hominin impacts. *Science* 362:938–941.
- Forey, P. L., and S. V. T. Young. 1999. Late Miocene fishes of the Emirate of Abu Dhabi, United Arab Emirates; pp. 120–135 in P. J. Whybrow and A. P. Hill (eds.), *Fossil Vertebrates of Arabia, with Emphasis on the Late Miocene Faunas, Geology, and Palaeoenvironments of the Emirate of Abu Dhabi, United Arab Emirates*. Yale University Press, New Haven.
- Gentry, A. W. 2010. Bovidae; pp. 747–803 in L. Werdelin and W. J. Sanders (eds.), *Cenozoic Mammals of Africa*. University of California Press, Berkeley.
- Gentry, A. W. 1985. The Bovidae of the Omo Group deposits, Ethiopia (French and American collections); pp. 119–191 in Y. Coppens and F. C. Howell (eds.), *Les faunes Plio-Pléistocènes de la basse Vallée de l’Omo (Ethiopie); I: Perissodactyles-Artiodactyles (Bovidae)*, . Cahiers de Paléontologie. CNRS, Paris.
- Holt, B. G., J.-P. Lessard, M. K. Borregaard, S. A. Fritz, M. B. Araújo, D. Dimitrov, P.-H. Fabre, C. H. Graham, G. R. Graves, K. A. Jønsson, D. Nogués-Bravo, Z. Wang, R. J. Whittaker, J. Fjeldså, and C. Rahbek. 2013. An update of Wallace’s zoogeographic regions of the world. *Science* 339:74–78.
- Kreft, H., and W. Jetz. 2010. A framework for delineating biogeographical regions based on species distributions. *Journal of Biogeography* 37:2029–2053.
- Otero, O. 2022. Fishes from the Baynunah Formation; pp. 79–109 in F. Bibi, B. Kraatz, M. J. Beech, and A. Hill (eds.), *Sands of Time: Ancient Life in the Late Miocene of Abu Dhabi, United Arab Emirates*, . Vertebrate Paleobiology and Paleoanthropology Springer International Publishing, Cham.
- Suwa, G., Y. Beyene, H. Nakaya, R. L. Bernor, J.-R. Boisserie, F. Bibi, S. H. Ambrose, K. Sano, S. Katoh, and B. Asfaw. 2015. Newly discovered cercopithecoid, equid and other mammalian fossils from the Chorora Formation, Ethiopia. *Anthropological Science* 123:19–39.
- Trauth, M. H., A. Asrat, N. Berner, F. Bibi, V. Foerster, M. Grove, S. Kaboth-Bahr, M. A. Maslin, M. Mudelsee, and F. Schäbitz. 2021. Northern Hemisphere Glaciation, African climate and human evolution. *Quaternary Science Reviews* 268:107095.
- Vrba, E. S., F. Bibi, and A. G. Costa. 2015. First Asian record of a late Pleistocene reduncine (Artiodactyla, Bovidae, Reduncini), *Sivacobus sankaliai*, sp. nov., from Gopnath (Millioli Formation) Gujarat, India, and a revision of the Asian genus *Sivacobus* Pilgrim, 1939. *Journal of Vertebrate Paleontology* 35:e943399.
- Vrba, E. S. 1995. The fossil record of African antelopes (Mammalia, Bovidae) in relation to human evolution and paleoclimate; pp. 385–424 in E. S. Vrba, G. H. Denton, T. C. Partridge, and L. H. Burckle (eds.), *Paleoclimate and Evolution, with Emphasis on Human Origins*. Yale Univ. Press, New Haven.

Referee #4 (Remarks to the Author):

Summary of the Key results

In this manuscript, authors did generate a new hydroclimate record covering the last 8 million years from central Arabia region using speleothems record from 7 caves. To do so they combined U-Pb / U-Th dating with speleothem morphology/mineralogy analysis to inform on chronology of wet episodes. They then combine this with $d_{18}O$ of calcite to discuss general trend in rainfall activity over the studied period. Authors also used isotopic composition of fluid inclusions ($d_{18}O$ and d_2HFI) from the speleothems compared with those of various moisture sources in the region in order to track potential change in the geographic origin of moisture over the studied period of time. Based on this methodology they identified various wet periods over the late 8 Ma and compare the obtained chronology with the evolution of the meridional gradient of Sea Surface Temperature they derived from paleoceanographic records from high latitude and within the tropics. They used this to suggest that hydroclimate dynamics in the studied region is driven by periodic warming of northern hemisphere high latitudes and shrinkage of northern ice-sheets. Authors then related the identified wet periods (in an area that is at present-day hyperarid) to biogeographic record, and suggest that periodic environmental changes in this region at the crossroad of Eurasia and Africa have force dispersion dynamics of Fauna between the two subregions.

We thank Reviewer 4 for their detailed summary of our manuscript.

Originality and Significance

As a non-specialist of U-Th and U-Pb dating methods or speleothems I can only comment on the climate aspect of the paper. The new dataset provided in the study, is to my opinion a very important piece on the environmental evolution of a crucial region for both mammals & hominids dispersal but also important for the understanding of paleo-climate and climate dynamics of the Saharo-Arabian desert. This especially as datas from this part of the world are very few when it comes to late Miocene-Pliocene records. If combined properly with recent records documenting the evolution of West Sahara region and Mesopotamia (eg. Crocker et al., 2022 ; Böhme et al., 2021 for the most recent ones), that roughly cover the same period of time, the new record could enable to draw a full picture of the large scale dynamics of the Saharo-Arabian region over time. Picturing the evolution of climate in this region and understanding forcing mechanisms and feedbacks is of major importance to better figuring out how hyper-arid regions will likely response to future climate change. The record presented here should therefore be of interest for various communities as it covers a period of time that is key for the understanding of hominids dispersal and climate dynamics in a world warmer than present day. Based on paleoclimate/climate dynamics of such record I would recommend considering this paper for publication after major improvement of the discussion.

We thank Reviewer 4 for recognising the potential contribution of our work to the field and its role in enabling us to draw a full picture on the evolution of climate in this region. Further, that understanding climate in these regions has clear knock-on effects for the evolution of future climate in dryland areas. We thank the Reviewer for recommending that our manuscript be published, after making the amendments suggested, which we have since done. We have made the requested changes.

Conclusions & Suggest improvements

The paper is well written/very clear, including the methodology (even for a non specialist) and figures are relevant, easy to read and to understand, which is a good point. General trend in hydro climate authors interpret from $d_{18}O_{Carb}$ in the speleothem are in line with progressive drying of the climate and shift in environment recorded from the late Miocene onward, alongside progressive cooling. Identified wet period seems more or less coherent with those identified further north by Böhme et al., 2021.

We thank reviewer 4 for indicating that our manuscript is easy to read and understand for a generalist audience. We agree that there is coherence with our record and that of Bohme et al., 2021 and we have provided further discussion on how these records agree and where they disagree further in the text.

Few things though required clarification from the authors to my opinion :

- I note that it seems to be an incoherence between the 5 - 3 Ma wet period identified on Fig 2. and description in the text stating that the 6 - 3.5 Ma period is dry and suggest expansion of the arid belt, so it fits the Biogeographic exchange record. I should be better discuss also with regard to the timing of aridity expansion interpreted by Böhme et al., 2021 so it avoid confusing the reader.

This is a good point. Our findings in the Pliocene in central Arabia contradict the findings in Northern Arabia by Bohme et al.,³⁸. However, there may be several explanations for this. Firstly, their site is located in Mesopotamia and quite a distance further North (33 degrees) than our site (26 degrees) in central Arabia, so different climatic regimes likely control each respective location. This can be seen if we look at the present-day precipitation maps where the Bohme et al.³⁸ site is dominated by predominately winter derived rainfall (Fig. 1b), whereas our site receives very little rainfall from *both* summer and winter seasons (monsoon vs westerlies, respectively) and sits on the transition zone between these two dominated rainfall regimes. Secondly, Bohme et al.³⁸ present a terrestrial sedimentary record which is related to the fluvial inflows of the Paleo-Tigris River, which was already established in a SE direction by the Late Miocene. Consequently, the origin of the water (and by extension aridity) could be reflected in distal climatic changes in the headwaters, which today lie in modern Turkey. By contract, our record has the advantage of recording the climatic conditions directly in situ. Thirdly, in the Negev desert (i.e. Chaldekas et al.,⁴⁸) intermittent vadose zone speleothem formation occurs (30 degrees N) from 6 Ma until the Quaternary, suggesting increases in rainfall in the North of the Arabian Peninsula and Levant. We have added the following text to address the incoherence:

Lines 738-774: “Previous work in northern Arabia (33° N) provided an explanation of Neogene hyperaridity resulting in a sustained arid barrier spanning 5.6 and 3.3 Ma, impeding dispersals and consequently allowing African mammalian faunas to endemically diversify to their present-day clades³⁸. Our data, representing *in situ* climatic sensitivity, reveal that this aridity did not extend to the entire Arabian Peninsula and that the Pliocene contained episodes of higher than modern precipitation that could support and facilitate a range of species that are currently absent from the Peninsula.”

- Despite the main finding being interesting and important I regret that the content in the main text is a bit light when it comes to climate mechanisms or biogeography. I had to get in the Supplementary information too many times to find informations of interest. There are for exemple a lot of crucial reasoning on how the authors related Meridional Gradient of Temperature and Humid periods they identified in the part 2.2 of the Supplementary information, that could be useful in the main text. Same thing is true for the part 2.3 of the SI that provide important informations about the relation between biogeography and climate. I think the manuscript could be largely improve by using part of the information in those two part of the SI to be clearer on the reasoning while relating timing of wet period identified in the speleothem records, climate dynamics and dispersal.

We have added further detail on the relevant climate mechanisms. We have reduced the emphasis on biogeography and fauna in our responses to Review 3's comments above. See our below responses for our extended discussion on orbital controls. We cannot extend this further due to word limit restriction and note that the extended discussion we now include does take the MS slightly over the word count (2500). For further discussion we still include a more detailed summary in the Supplementary Information .

We have also provided further detail in our section on the MTGs as follows:

Lines:“ Meridional temperature gradients
Meridional temperature gradients

Lines 576-616: “A gradual decline of the poleward (northward) extent of the tropical rain belt during humid episodes over the last 8 Ma is consistent with increases in the meridional sea surface temperature (SST) gradient between the Arabian Sea and the northern Atlantic Ocean over the last 8 Ma (Fig 3b and see Supplementary Section 3.1 for expanded text). Higher meridional SST gradients are associated with the Pleistocene and Late Miocene cooling intervals and are strongly coupled with NH polar ice extent (Fig. 3). During periods with lower meridional SST gradients (Fig. 3), like the mid-Pliocene, global climate model simulations show increased precipitation in Arabia during boreal summer, due to a weaker Hadley Circulation and reduced tropical convection⁴⁹. Conversely, in the Miocene-Pliocene boundary and the Pleistocene, when the gradient is high the Hadley Circulation strengthened and contracted, bringing largescale subsidence and prolonged dry conditions over Arabia. NH cooling is reflected in our record by the gradual oxygen isotope shift to higher $\delta^{18}\text{O}$ values, both in speleothem calcite and fluid inclusion water, which provides evidence for shifting atmospheric circulation patterns, and a reduced effective precipitation trend during interglacials, particularly after the MPT.

Changes in atmospheric circulation patterns in Saharo-Arabia are also evidenced by increased dust transport from the Sahara Desert from 2.3 Ma²⁵ (Fig. 3d) and greater terrigenous input into the Arabian Sea²⁶. Enhanced dust transport has been attributed to the Pleistocene intensification of glacials, associated with increased aridity, less vegetation, lower soil moisture, changes in atmospheric wind direction and intensity coupled with lower precipitation²⁷. The production of fine-grained sediments from source areas such as Mega Lake Chad during interglacials is followed by desiccation, deflation and their transport in glacials, acting as dust-producing ‘hot spots’²⁵. Importantly, however, as our speleothems are selective records of interglacial humid intervals (opposed to glacial aridity), the drying trend observed in the speleothem record may occur in tandem with higher dust transport in glacial phases. Further, during the globally warmer and higher atmospheric CO₂ world of the Pliocene and Late Miocene^{28,29} the wetter humid episodes in central Arabia are likely associated with lower wind speeds, more vegetation and higher recharge. “

- The authors put all the focus on Meridional Temperature Gradient and ice-sheet forcing to explain the wet periods, while barely invoking insolation change as a major driver of hydroclimate. I agree that change in SST gradient should have an impact on ITCZ dynamics, as it controls the global atmosphere dynamics and expansion of Hadley cells as already largely discussed by authors. But previous work on northern African region also often relate oscillations in climate/environment to orbital forcing (such as this is the case during the well documented Green Sahara period of the Holocene/Pleistocene). I encourage authors to use available modeling work describing orbital effect on Northern African monsoon (see Kutzbach et al., 1981 for historic work, but more recent papers including Bosmans et al., 2015 for example are also insightful). While I am not aware of work focusing directly on Arabian region (but might have missed some), most of those studies display regional maps where Arabian region is represented. Most of those papers display simulations that modify the moisture flux and precipitation dynamics without modifying the extension of northern ice-sheet suggesting that change in regional Sea Surface Temperature are also somehow responsible for driving the signal of precipitation. This should be better discuss in the paper.

We thank reviewer 3 for pointing this out. Whilst we initially did not include much on orbital-scale time frames, as our dataset doesn't have the dating resolution to resolve different orbital phasing, we appreciate that reference to other literature and climate modelling output is of value to add context to our overall findings and implications. Consequently, as suggested, we have added an additional paragraph discussing precipitation dynamics and fluxes including orbital forcing mechanisms:

Lines 650-693: **“Orbital-driven humidity**

“In the subtropical drylands, like southern Arabia, humid episodes are often only pulsed and short-lived (several to tens of thousands of years)^{21,30}. Astronomical forcing, particularly orbital precession cycles, induce large changes in seasonal insolation, up to 100 W m⁻², increasing the tropical insolation gradient and resulting in a poleward extension of the African monsoon⁵⁰. In Arabia, solar radiation is principally influenced by variations in precession, modulated by

eccentricity⁵¹ which is often linked to high-latitude climate variability⁵². However, direct low-latitude insolation appears responsible for obliquity-paced Indian monsoonal variability in the Arabia Sea⁵³ as well as meridional shifts in the African monsoon belt⁵⁰. On land, global climate models from the last interglacial suggest a substantial increase in precipitation in the Arabian Peninsula in the order of 300 to 600 mm a⁻¹, originating mainly from the North African monsoon⁵⁴. Whilst we cannot resolve orbital-scale forcing in most of our record, dates cluster around interglacial phases suggesting orbital cycles play a role, but only when background conditions, such as reduced NH ice cover, are suitable. Climate modelling of the Tethys suggests a high sensitivity of the African monsoon to orbital forcing began after the Tethys closure by the Late Miocene¹⁶. In low NH ice conditions, such as the Pliocene, the high density of pulsed humid phases we observe in Arabia (Fig. 3f) may potentially be driven by orbital insolation cycles which are more result in more effective precipitation in a 'warmer' versus 'ice-age' world."

Another thing that is not discussed when it comes to explaining the global trend toward more aridity from the late Miocene to present-day is the potential effect of changes in geography. Zhang et al. 2014 for example do suggest that change in Parathetys extent or Iran plateau elevation (or rather the Iran-Anatolia topography) might have been responsible to aridification of the Sahara desert. And this seems that both of those geography features have been evolving during the late Miocene-Pliocene, though their direct effect of Arabian climate have not been extensively studied. I understand that the focus of this paper is more on identifying periodic humid periods and related it to biogeography, but this should at least be mentioned somewhere.

This is a good point. However, Zhang et al.¹⁶, suggest that by the Late Miocene the closure of the Tethys is already established. This closure they then postulate lead to the enhanced sensitivity of the African monsoon to orbital cycles. We have included a sentence to this effect as follows:

Lines 663-690: "Climate modelling of the Tethys suggests a high sensitivity of the African monsoon to orbital forcing began after the Tethys closure by the Late Miocene¹⁶".

Minor comments :

- Please try to be consistent in the use of units : mm.a-1 vs. mm/a

Thanks for noticing this. We have changed all the units to mm a-1 for consistency.

Supplement 1

Line 92: "...(<100 mm a⁻¹)..."

Main text

Line 245: "...(~104 mm a⁻¹; Fig. 1)..."

Line 3274-275: "...~300 mm a⁻¹..."

- Fig 3c - Indonesian Throughflow instead of Indonesian followthrough

This has been changed to "Indonesian Throughflow" as suggested.

Zhang, Z. et al. Aridification of the Sahara desert caused by Tethys Sea shrinkage during the Late Miocene. *Nature* 513, 401-404 (2014). <https://doi.org:10.1038/nature13705>

Böhme, M. et al. Neogene hyperaridity in Arabia drove the directions of mammalian dispersal between Africa and Eurasia. *Communications Earth & Environment* 2, 85 (2021). <https://doi.org:10.1038/s43247-021-00158-y>

E. Kutzbach, Monsoon climate of the early Holocene: Climate experiment with the Earth's orbital parameters for 9000 years ago. *Science* 214, 59-61 (1981).

Bosmans et al., 2015. Response of the North African summer monsoon to precession and obliquity forcings in the EC-Earth GCM, *Climate dynamics*, 44, 279-297 (2015)

References:

- 1 Pollard, T. *et al.* DQPB: software for calculating disequilibrium U–Pb ages. *Geochronology* **5**, 181-196 (2023). <https://doi.org:10.5194/gchron-5-181-2023>
- 2 Schmitz, M. D. & Schoene, B. Derivation of isotope ratios, errors, and error correlations for U-Pb geochronology using ²⁰⁵Pb-²³⁵U-(²³³U)-spiked isotope dilution thermal ionization mass spectrometric data. *Geochemistry, Geophysics, Geosystems* **8** (2007). <https://doi.org:https://doi.org/10.1029/2006GC001492>
- 3 Noda, A. A new tool for calculation and visualization of U–Pb age data: UPbplot.py. *BULLETIN OF THE GEOLOGICAL SURVEY OF JAPAN* **68**, 131-140 (2017). <https://doi.org:10.9795/bullgsj.68.131>
- 4 Ludwig, K. R. Isoplot/Ex Version 3.75: A Geochronological Toolkit for Microsoft Excel. 1-75. (Berkeley Geochronology Center, , Berkley, USA, 2012).
- 5 Weij, R., Woodhead, J., Hellstrom, J. & Sniderman, K. An exploration of the utility of speleothem age distributions for palaeoclimate assessment. *Quaternary Geochronology* **60**, 101112 (2020). <https://doi.org:https://doi.org/10.1016/j.quageo.2020.101112>
- 6 Scroxton, N. *et al.* Natural attrition and growth frequency variations of stalagmites in southwest Sulawesi over the past 530,000years. *Palaeogeography, Palaeoclimatology, Palaeoecology* **441**, 823-833 (2016). <https://doi.org:https://doi.org/10.1016/j.palaeo.2015.10.030>
- 7 Fleitmann, D., Matter, A., Pint, J. J. & Al-Shanti, M. A. The Speleothem Record of Climate Change in Saudi Arabia. (Saudi Geological Survey, Jeddah, Saudi Arabia, 2004).
- 8 Woodhead, J. D. *et al.* The antiquity of Nullarbor speleothems and implications for karst palaeoclimate archives. *Scientific Reports* **9**, 603 (2019). <https://doi.org:10.1038/s41598-018-37097-2>
- 9 Vaks, A., Bar-Matthews, M., Ayalon, A., Matthews, A. & Frumkin, A. Pliocene–Pleistocene palaeoclimate reconstruction from Ashalim Cave speleothems, Negev Desert, Israel. *Geological Society, London, Special Publications* **466**, 201-216 (2018). <https://doi.org:doi:10.1144/SP466.10>
- 10 Herbert, T. D. *et al.* Late Miocene global cooling and the rise of modern ecosystems. *Nature Geoscience* **9**, 843-847 (2016). <https://doi.org:10.1038/ngeo2813>
- 11 Holbourn, A. E. Late Miocene climate cooling and intensification of southeast Asian winter monsoon. *Nature Communications* **9**, 1584 (2018).
- 12 Larsen, H. C. Seven Million Years of Glaciation in Greenland. *Science* **264**, 952-955 (1994). <https://doi.org:10.1126/science.264.5161.952>
- 13 Krijgsman, W., Hilgen, F. J., Raffi, I., Sierro, F. J. & Wilson, D. S. Chronology, causes and progression of the Messinian salinity crisis. *Nature* **400**, 652-655 (1999).
- 14 Hsü, K. J., Stoffers, P. & Ross, D. A. Messinian evaporites from the Mediterranean and Red Seas. *Marine Geology* **26**, 71-72 (1978). [https://doi.org:https://doi.org/10.1016/0025-3227\(78\)90046-4](https://doi.org:https://doi.org/10.1016/0025-3227(78)90046-4)
- 15 Gargani, J., Moretti, I. & Letouzey, J. Evaporite accumulation during the Messinian Salinity Crisis: The Suez Rift case. *Geophysical Research Letters* **35** (2008). <https://doi.org:10.1029/2007GL032494>

- 16 Zhang, Z. *et al.* Aridification of the Sahara desert caused by Tethys Sea shrinkage during the Late Miocene. *Nature* **513**, 401-404 (2014).
<https://doi.org/10.1038/nature13705>
- 17 Pickering, R. *et al.* U–Pb-dated flowstones restrict South African early hominin record to dry climate phases. *Nature* **565**, 226-229 (2019).
<https://doi.org/10.1038/s41586-018-0711-0>
- 18 Engel, J. *et al.* Using speleothems to constrain late Cenozoic uplift rates in karst terranes. *Geology* **48**, 755-760 (2020). <https://doi.org/10.1130/g47466.1>
- 19 Weij, R. *et al.* Cave opening and fossil accumulation in Naracoorte, Australia, through charcoal and pollen in dated speleothems. *Communications Earth & Environment* **3**, 210 (2022). <https://doi.org/10.1038/s43247-022-00538-y>
- 20 Vaks, A., Bar-Matthews, M., Matthews, A., Ayalon, A. & Frumkin, A. Middle-Late Quaternary paleoclimate of northern margins of the Saharan-Arabian Desert: reconstruction from speleothems of Negev Desert, Israel. *Quaternary Science Reviews* **29**, 2647-2662 (2010). <https://doi.org/10.1016/j.quascirev.2010.06.014>
- 21 Nicholson, S. L. *et al.* Pluvial periods in Southern Arabia over the last 1.1 million-years. *Quaternary Science Reviews* **229**, 106112 (2020).
<https://doi.org/https://doi.org/10.1016/j.quascirev.2019.106112>
- 22 Vaks, A. Pliocene–Pleistocene climate of the northern margin of Saharan–Arabian Desert recorded in speleothems from the Negev Desert, Israel. *Earth and Planetary Science Letters* **368**, 88-100 (2013). <https://doi.org/10.1016/j.epsl.2013.02.027>
- 23 Bajo, P. *et al.* Persistent influence of obliquity on ice age terminations since the Middle Pleistocene transition. *Science* **367**, 1235-1239 (2020).
<https://doi.org/doi:10.1126/science.aaw1114>
- 24 Gázquez, F., Calaforra, J. M., Evans, N. P. & Hodell, D. A. Using stable isotopes ($\delta^{17}\text{O}$, $\delta^{18}\text{O}$ and δD) of gypsum hydration water to ascertain the role of water condensation in the formation of subaerial gypsum speleothems. *Chemical Geology* **452**, 34-46 (2017). <https://doi.org/https://doi.org/10.1016/j.chemgeo.2017.01.021>
- 25 Crocker, A. J. *et al.* Astronomically controlled aridity in the Sahara since at least 11 million years ago. *Nature Geoscience* **15**, 671-676 (2022).
<https://doi.org/10.1038/s41561-022-00990-7>
- 26 deMenocal, P. B. Plio-Pleistocene African Climate. *Science* **270**, 53-59 (1995).
- 27 Muhs, D. R. The geologic records of dust in the Quaternary. *Aeolian Research* **9**, 3-48 (2013). <https://doi.org/https://doi.org/10.1016/j.aeolia.2012.08.001>
- 28 de la Vega, E., Chalk, T. B., Wilson, P. A., Bysani, R. P. & Foster, G. L. Atmospheric CO₂ during the Mid-Piacenzian Warm Period and the M2 glaciation. *Scientific Reports* **10**, 11002 (2020). <https://doi.org/10.1038/s41598-020-67154-8>
- 29 Wen, Y. *et al.* CO₂-forced Late Miocene cooling and ecosystem reorganizations in East Asia. *Proceedings of the National Academy of Sciences* **120**, e2214655120 (2023). <https://doi.org/doi:10.1073/pnas.2214655120>
- 30 Fleitmann, D., Burns, S. J., Matter, A., Cheng, H. & Affolter, S. Moisture and Seasonality Shifts Recorded in Holocene and Pleistocene Speleothems From Southeastern Arabia. *Geophysical Research Letters* **49**, 2021-097255
<https://doi.org/10.1029/2021GL097255>
- 31 Fohlmeister, J. *et al.* Main controls on the stable carbon isotope composition of speleothems. *Geochimica et Cosmochimica Acta* **279**, 67-87 (2020).
<https://doi.org/https://doi.org/10.1016/j.gca.2020.03.042>
- 32 Köhler, P. & van de Wal, R. S. W. Interglacials of the Quaternary defined by northern hemispheric land ice distribution outside of Greenland. *Nature Communications* **11**, 5124 (2020). <https://doi.org/10.1038/s41467-020-18897-5>

- 33 Kreft, H. & Jetz, W. Comment on “An Update of Wallace’s Zoogeographic Regions of the World”. *Science* **341**, 343-343 (2013).
<https://doi.org/doi:10.1126/science.1237471>
- 34 Holt, B. G. *et al.* An Update of Wallace’s Zoogeographic Regions of the World. *Science* **339**, 74-78 (2013). <https://doi.org/doi:10.1126/science.1228282>
- 35 Osborne, C. P. Atmosphere, ecology and evolution: what drove the Miocene expansion of C4 grasslands? *Journal of Ecology* **96**, 35-45 (2008).
<https://doi.org/https://doi.org/10.1111/j.1365-2745.2007.01323.x>
- 36 Cerling, T. E. *et al.* Global vegetation change through the Miocene/Pliocene boundary. *Nature* **389**, 153-158 (1997). <https://doi.org/10.1038/38229>
- 37 Westerhold, T. *et al.* An astronomically dated record of Earth’s climate and its predictability over the last 66 million years. *Science* **369**, 1383-1387 (2020).
<https://doi.org/doi:10.1126/science.aba6853>
- 38 Böhme, M. *et al.* Neogene hyperaridity in Arabia drove the directions of mammalian dispersal between Africa and Eurasia. *Communications Earth & Environment* **2**, 85 (2021). <https://doi.org/10.1038/s43247-021-00158-y>
- 39 Polissar, P. J., Rose, C., Uno, K. T., Phelps, S. R. & deMenocal, P. Synchronous rise of African C4 ecosystems 10 million years ago in the absence of aridification. *Nature Geoscience* **12**, 657-660 (2019). <https://doi.org/10.1038/s41561-019-0399-2>
- 40 Otero, O. in *Sands of Time: Ancient Life in the Late Miocene of Abu Dhabi, United Arab Emirates* (eds Faysal Bibi, Brian Kraatz, Mark J. Beech, & Andrew Hill) 79-109 (Springer International Publishing, 2022).
- 41 Haile-Selassie Y, Vrba ES & F, B. in *Ardipithecus kadabba: Late Miocene Evidence from the Middle Awash, Ethiopia* (ed WoldeGabriel G Haile-Selassie Y) pp. 277–330. (Berkeley: University of California Press, 2009).
- 42 Bibi, F. i. S. o. T. *Ancient Life in the Late Miocene of Abu Dhabi, United Arab Emirates*. (Springer International Publishing, 2022).
- 43 McDougall, I. *et al.* New single crystal ⁴⁰Ar/³⁹Ar ages improve time scale for deposition of the Omo Group, Omo–Turkana Basin, East Africa. *Journal of the Geological Society* **169**, 213-226 (2012). <https://doi.org/10.1144/0016-76492010-188>
- 44 Gentry, A. in *Les Faunes Plio-Pléistocène de la Basse Vallée de l’Omo (Éthiopie). Tom 1: Périssodactyles–Artiodactyles (Bovidae)* (eds Y. Coppen & F.C. Howell) pp 119–191 (CNRS, 1985).
- 45 Gilbert, C. C., Bibi, F., Hill, A. & Beech, M. J. Early guenon from the late Miocene Baynunah Formation, Abu Dhabi, with implications for cercopithecoid biogeography and evolution. *Proceedings of the National Academy of Sciences* **111**, 10119-10124 (2014). <https://doi.org/doi:10.1073/pnas.1323888111>
- 46 Vignaud, P. Geology and palaeontology of the Upper Miocene Toros-Menalla hominid locality, Chad. *Nature* **418**, 152-155 (2002).
- 47 Brunet, M. A new hominid from the Upper Miocene of Chad, Central Africa. *Nature* **418**, 145-151 (2002).
- 48 Chaldeckas, O., Vaks, A., Haviv, I., Gerdes, A. & Albert, R. U-Pb speleothem geochronology reveals a major 6 Ma uplift phase along the western margin of Dead Sea Transform. *GSA Bulletin* **134**, 1571-1584 (2021).
<https://doi.org/10.1130/b36051.1>
- 49 Burls, N. J. & Fedorov, A. V. 12888-12893.
- 50 Bosmans, J. H. C., Drijfhout, S. S., Tuenter, E., Hilgen, F. J. & Lourens, L. J. Response of the North African summer monsoon to precession and obliquity forcings in the EC-Earth GCM. *Climate Dynamics* **44**, 279-297 (2015).
<https://doi.org/10.1007/s00382-014-2260-z>

- 51 Clemens, S. C. & Tiedemann, R. Eccentricity forcing of Pliocene–Early Pleistocene climate revealed in a marine oxygen-isotope record. *Nature* **385**, 801-804 (1997). <https://doi.org:10.1038/385801a0>
- 52 Schulz, H., von Rad, U., Erlenkeuser, H. & von Rad, U. Correlation between Arabian Sea and Greenland climate oscillations of the past 110,000 years. *Nature* **393**, 54-57 (1998). <https://doi.org:10.1038/31750>
- 53 Clemens, S. C. & Prell, W. L. A 350,000 year summer-monsoon multi-proxy stack from the Owen Ridge, Northern Arabian Sea. *Marine Geology* **201**, 35-51 (2003). [https://doi.org:https://doi.org/10.1016/S0025-3227\(03\)00207-X](https://doi.org:https://doi.org/10.1016/S0025-3227(03)00207-X)
- 54 Jennings, R. P. *et al.* The greening of Arabia: Multiple opportunities for human occupation of the Arabian Peninsula during the Late Pleistocene inferred from an ensemble of climate model simulations. *Quaternary International* **382**, 181-199 (2015). <https://doi.org:https://doi.org/10.1016/j.quaint.2015.01.006>

Authors response to editorial and reviewer comments

Please note that all line reference numbers refer to the PDF version of the manuscript. Changes to text from the previous version of the manuscript are indicated in green.

Editorial comments:

- Provide stand-alone figures both for the main text and Extended Data (ED) items (details below). Main text figures may be AI, Vector EPS, layered PSD, postscript, PDF, PowerPoint, Word, or Excel files. ED files (figures and tables) may be JPG, TIF, or EPS. For multi-panel figures, submit a single file with the panels organized as you would like, rather than as separate files for each panel.

We have uploaded stand-alone figures for the main text figures as .PDF files and ED as .jpg files as requested.

- For the main text figures, please ensure that all text items are vectorized. Here is the specific guidance provided by my colleagues in our Art team: "... panels and figures ideally should be supplied as vector artwork – where lines, arrows, scale bars and text, etc. are all in an editable format and labels are on separate layers to images."

We have vectorised all the main text figures as requested.

- To the extent possible, please use declarative titles for the figure legends (i.e. state the figure's main point).

We have made the figure captions declarative titles as requested with the new titles listed below:

Line 292 Fig 1. **Hyper-arid Central Arabia is at the centre of two moisture bearing systems.**

Line 306-307: Fig. 2. **Central Arabian speleothems reveal the episodic occurrence of humid intervals over the last 8 Ma.**

Line 321-322: Fig. 3. **Recurrent central Arabian humid episodes over the Late-Miocene-to-Late-Pleistocene are associated with increasing regional aridity and higher NH meridional temperature gradients.**

Line 342-343: **Palaeo-isotopic composition of central Arabian speleothem fluid inclusion waters were associated with a monsoonal southerly source and exhibit lower values relative to present day rainfall.**

- Define all error bars within the figure legends. It looks like aspects of figures 2, 3, and 4 and SI figures 1-3 and 9-11 will require revision. Please check all.

Thank you for pointing this out. We have checked all the error bars for figures in the main text as well as the extended data.

In the main text:

Lines 314-315: "...and LA-MC-ICP-MS (red diamonds) with the 95% confidence interval uncertainty indicated by the horizontal error bars..."

Lines 333-334: "...and solution MC-ICP-MS (filled squares), with the 95% confidence interval uncertainty indicated by the horizontal error bars..."

Lines 350-51: "...The uncertainty error bars on the fluid inclusion values are 0.3 (‰ VSMOW) and 1.1(‰ VSMOW) for $\delta^2\text{H}$ and $\delta^{18}\text{O}$, respectively, with further information provided in the Methods section. .."

The figure captions in the Extended Data.

Lines 1355-57: "...Yemen and Oman¹¹ with the 95% confidence interval uncertainty indicated by the horizontal error bars. (i)..."

Lines 1377-78: "...The horizontal error bars of ages estimates refer to the 95% confidence interval uncertainty..."

Lines 1385-86: "...The horizontal error bars of ages estimates refer to the 95% confidence interval uncertainty..."

- I may have missed it, but it looks like panel f in figure 2 is not referenced in the legend. Please adjust as needed and check all figures.

Thank you for noticing this omission. We have changed this and included a reference for Panel f in the Fig. 2 caption as follows:

Lines 313-314: "(Panel e). **Panel f**; U-Pb ages (grey-filled circles), U-Th derived ages using solution MC-ICP-MS U-Th (yellow squares...)".

- It isn't clear to me why you are using a divergent color scale in panels b and c of figure 1 (at least in my experience divergent scales are usually reserved for anomaly color scales). Please check and revise as appropriate.

We have changed this to a sequential colour scale in the palette viridis as suggested.

[FIGURE REDACTED]

- Some of your figures, such as SI figure 9a-f and 10d superimpose colors on the line plots. Do these colors duplicate what is already shown on the y-axis? If so, please remove the colors. If the colors show additional information, you must provide a color bar and explain in the legend.

We have added the additional information on the two figures, both as a colour bar on the figure itself and as an explanation in the figure caption.

SFig 10. Pleistocene hydroclimate of the central Arabian desert and major climate variables. a) Benthic LR04 $\delta^{18}\text{O}$ stack²¹. The modern values are indicated by the dashed black line. b) Glacial index, where 1 is a full glacial and 0 is a full interglacial, for full details on calculation see Koeler and van de Wal²³. The blue dashed line is used as the cutoff (0.5) for significant interglacial phases as indicated by the red panels, where 1 indicates a full glacial endmember and 0 a full interglacial endmember. c) indicates the key for cool, lukewarm and warm interglacial phases. d) Meridional Temperature Gradients calculated from marine sedimentary cores 982 and 722¹⁶ where warm (MTG<11), lukewarm (11>MTG<12) and cool interglacials (MTG>12) are identified by red dashed, pink dashed and purple lines, respectively. e) Negev desert speleothem ages¹⁹. f) Central Arabian speleothem ages (this study) where grey circles indicate U-Pb ages, yellow diamonds U-Th ages by solution MC-ICPMS and red squares U-Th ages by LA-MC-ICPMS. g) Southern Arabia speleothem ages from a compilation of cave sites^{11,24-27}. The horizontal error bars of ages estimates refer to the 95% confidence interval uncertainty. The shaded background panels indicate periods where the glacial index is <0.5.

[FIGURE REDACTED]

SFig. 11 Arabian humid episodes and global climate cycles. Panel a shows the LR04 benthic stack²¹; Panel b is the June 21 insolation anomaly at 25° N²⁸; Panel c is the eccentricity anomaly (blue) and climate precession anomaly (black)²⁸; Panel d shows the central Arabian U-Th and U-Pb dates as well as two ages reported by Fleitmann et al.,²⁶; Panel e are the ages from a compilation of caves in southern Arabia^{11,24-27}. The horizontal error bars of ages estimates refer to the 95% confidence interval uncertainty. The green background shading indicates interglacial phases.

- The legend for 3 reads “Background shading relates to the histogram frequency in Fig. 2h” but there is no figure 2h and in any case all figures should be fully self-contained. Please add a color bar and explain fully within the legend.

We have added a colour bar to Fig. 3 and included this description in the caption.

Lines 339-340: “...Background shading shows the U-Pb histogram frequency in Fig. 2g, and is indicated by the colour bar.”

- Integrate SI figure 4 into figure 1 of the main text.

We have added an additional panel to Fig. 1 to incorporate what was previously SI Figure 4.

[FIGURE REDACTED]

- Ensure that precise latitudes and longitudes are provided for all caves (possible as another tab in the xlsx SI). If precise locations cannot be provided then the restrictions must be described within the Methods (I didn't exhaustively check for this information; if already provided please let me know where to look).

Unfortunately, we have been advised not to provide precise latitudes and longitudes for all the caves in our study from our colleagues at the Saudi Heritage Commission. This is due to significant and warranted concern for damage and access to caves which are proximal to a major road and are currently unmanned and ungated. In the past, damage to these fragile environments has occurred and further several of the caves are also risky to access without experienced cavers due to the vertical shaft entry passages. We provide a detailed false colour map in Fig 1d and the general coordinates of our study area in panels a,b,c. Further, we have specified both the conservation concern and access route in the Methods section (see below). Please let us know if we should discuss this further, or if general coordinates for the overall site area should be specified in the Methods section.

Lines 521-524: "The general cave locations have been provided (Fig. 1), further detail is not provided herein for conservation purposes. Precise locational information is available to professional organisations and scholars from the Saudi Heritage Commission and the Saudi Geological Survey."

- Ensure that all data sources and literature are appropriately cited in the figure legends, and precisely clarify which datasets arise from your own work and which are from other sources.

We have checked that we reference all datasets in the captions from data that did not arise from this publication and detail how we have calculated, for example, the meridional temperature gradients based on the previously published data.

- Our rights department is checking permissions with great zeal, and this can cause delays of many weeks. Therefore, in the Data Availability Statement, please note software and base maps used for any figure using a map. Also, for any figure showing data from a published source or website/archive, please include all necessary citations in the figure legend itself. If re-use permission is required please provide statements of permission.

We have double checked and included all the software and basemaps used for any figure containing a map in the Data Availability Statement to the best of our knowledge.

Lines 916-942: “Data availability statement

Data availability statement for the submission Nature 2023-05-07879 manuscript entitled “Recurrent humid phases in Arabia over the past 8 million years”.

Data publicly available in a repository:

Software used for basemaps includes ESRI Arc GIS Pro and the topographic hillshade underlay is the CGIAR Version 4 SRTM dataset (Fig. 1).

Jarvis, A., H.I. Reuter, A. Nelson, E. Guevara, 2008, Hole-filled SRTM for the globe Version 4, available from the CGIAR-CSI SRTM 90m Database (<http://srtm.csi.cgiar.org>).

The dataset on global land precipitation data from WorldClim is available at:

<https://www.worldclim.org/data/v1.4/worldclim14.html>

The dataset on the Sahara dust flux is used in Fig. 3d available at:

<https://doi.org/10.5281/zenodo.6594643>.

The dataset for the benthic d18O record is available at

<https://doi.org/10.5281/zenodo.6311999>.

The dataset for the East Africa Soil Carbonate Stable Isotope Data in Fig. 2d

<https://doi.org/10.1594/IEDA/100231>.

The dataset for the Glacial Index used in Extended Data File 8 is available in the Pangea repository at: <https://doi.org/10.1594/PANGAEA.914483>

The dataset used for the SSTs in the Meridional Temperature Gradient calculations (e.g. Fig. 3b) can be found at: <https://doi.org/10.1594/PANGAEA.885390>

Data available with the paper or in the supplementary information:

The authors declare that the data supporting the findings of this study are available within the paper and its supplementary information files.”

- Place all your data in a single multi-tabbed xlsx and submit as Supplementary Information. Then you must also place the data into a permanent, DOI-minting repository (details below).

We have combined all the supplementary data into a multi-tabbed excel file and submitted it as Supplementary Information as requested.

- We normally strive to avoid Supplementary Information. Here, it should be possible to merge your current SI into the Methods and ED. The text of the SI should be integrated into the Methods, where we can allow you up to 5000 words. Meeting the limit should be possible, with a mind towards concise wording. Then we can allow up to 11 Extended Data items, a limit that should be reachable, either by merging SI items or moving SI items to the main text (where we can allow up to six display items). The SI references should be moved to the Method References. Let's discuss if any of this seems problematic!

We have merged the SI text into the methods as suggested and kept below the suggested 5000 word count. We have moved the figures and tables from the SI to the ED and now have 11 extended data items associated with our manuscript. These have all been reformatted according to the *Nature* guidelines for extended data.

- As noted by referee 4 the numbering of the references requires adjustment. Also, we can allow up to 50 references in the main text, with the excess moved to the Methods References.

We have reduced the references in the main text to 50 as suggested, with the remaining references in a separate list after the Methods section. The references associated with the extended data have been provided in a separate list at the end of the manuscript. We have checked the numbering of all the references in the manuscript as suggested by reviewer 4 and believe the amended version is now correct.

- Remove "for the first time" from the summary paragraph. The central part of the paragraph should be reworded to something more straightforward, like "Here we show recurrent humid intervals in the central Arabian interior over the past 8 Ma. Our results are based on the longest Arabian paleoclimate record from desert speleothems."

We have changed this as suggested.

Lines 13-18: " While numerous humid phases occurred in southern Arabia during the last 1.1 Ma¹, little is known about Arabia's palaeoclimate before this time. **Our results draw upon a novel climatic record from desert speleothems, the longest paleoclimatic record currently available from Arabia. We show recurrent humid intervals in the central Arabian interior over the past 8 Ma.**"

- Are there specific permit numbers used for your collections? If so please include in the Acknowledgements.

We have no specific permit numbers related to the collections.

- I see that you acknowledge Saleh A. Alsoubhi, Mohammed Haptari, and Hisham Hashim. I'm wondering if these individuals might qualify for authorship? Note that our guide to authors states that authorship may be appropriate for individuals involved in "the acquisition, analysis, or interpretation of data" (<https://www.nature.com/nature-portfolio/editorial-policies/authorship#>).

We have checked the policy and have found that it would not be appropriate for the aforementioned individuals to be included on the authorship list as their contribution did not meet Nature's criteria for authorship.

- Please ensure that all isotopic standards are correctly defined (i.e. VSMOW or otherwise).

We have checked this throughout and included, ‰ VSMOW and , ‰ VPDB where appropriate as suggested.

- Organize your paper as: title page, text, references, figure legends, methods, data availability statement (moved from a stand-alone file to the main text), methods references, acknowledgements, author contributions, author information (containing interest declaration and corresponding author line), and extended data legends.

Reorganised as suggested.

Referee #1 (Remarks to the Author):

Thank you for your detailed responses to my queries - I believe you have fully addressed all of my concerns. Well done on an excellent piece of scientific research!

We thank Reviewer 1 for their recognition of our work and are pleased that all their concerns have been addressed.

Referee #2 (Remarks to the Author):

I thank the authors for addressing all my comments, I realise this was not a straightforward task and I appreciate their thorough responses. In my opinion, the authors have revised their manuscript satisfactorily:

We are pleased to hear that we have addressed all the comments of Reviewer 2 satisfactorily and that our thorough responses were appreciated.

- By reassessing their U-Pb data and by adding more [234U/238U] measurements, the authors have removed my concerns about the initial [234U/238U] disequilibrium correction and about a previously apparent trendline of the initial [234U/238U] with age. From the newly added [234U/238U] measurements it is now indeed evident that there is no significant trend in the initial [234U/238U] through time. I appreciate that they use speleothem-specific, back-calculated initial [234U/238U] values for (a now increased set of) samples younger than 1.5 Ma and now also support the authors' choice to use an average for samples beyond that. I realise this was a major undertaking, but I hope the authors understand that understanding variation in the initial [234U/238U] is paramount for the precision and accuracy of speleothem U-Pb data and, ultimately, its interpretation. I appreciate that they authors recalculated their ages using the spine fit type and agree that the ages are now more robust. Through this revision, I believe the data has high fidelity and now has my full support. I also appreciate that the authors have gone the extra mile by collating a database of the average [234U/238U] of speleothems and data of the activity ratios in modern aquifers in the greater Arabian Peninsula region – this gives further confidence (and also provides interesting insights into the region's initial [234U/238U] signal).

We agree with Reviewer 2 that our estimation of the initial $^{234}\text{U}/^{238}\text{U}$ is more robust in the revised version of this manuscript and that there is no significant trend through time. We also agree with Reviewer 2 that this data is now of high fidelity, particularly from the recalculation of all our ages using the Spine fit. We are further pleased that Reviewer 2 appreciated our efforts in collating a large database of all the regional $^{234}\text{U}/^{238}\text{U}$ to constrain the average values for the region.

- I appreciate that the authors have increased their U-Pb age data set. It is reassuring that the new ages fall within the previously identified humid phases. As the authors acknowledge, the sample size is still too small to identify discrete humid phases, and I appreciate it that the authors have removed references to the CAHP groupings. I believe the data are now more accurately presented and discussed.

We agree and think the data are more accurately presented in this way and appreciated the feedback from Reviewer 2 in the previous iteration.

- That brings me to my last point: by shifting the emphasise from humid phases being closely linked to faunal dispersals to the palaeoclimatic story, I believe this

manuscript now has a much stronger narrative, and, in my opinion, the revised version is now also more strongly supported by the underlying chronology.

Thank you for the commendation, we agree that the revised version is now more strongly supported by the underlying chronology, especially after removing some of the emphasis on specific dispersal events.

In summary, all points, and particularly my main concerns (the U-Pb age calculations, particularly the initial [234U/238U] disequilibrium correction and fit type for the U-Pb isochrons, sample size and preservation bias), have been satisfactorily addressed by the authors and are no longer concerns to me. I congratulate them on an impressive piece of work, and I now recommend this manuscript for publication.

We thank Reviewer 2 for recommending our manuscript for publication and appreciate all the feedback to get it to this new improved version.

Just a few minor comments/questions:

- The authors use an average initial 234U/238U activity ratio for samples >1.5 Ma. This is just something for the authors to consider whether an uncertainty-weighted average would be more appropriate.

We did not use the uncertainty-weighted average as it would potentially unfairly assign more weight to samples SA30 and SA31 which were young and consequently had low uncertainties and also slightly uncharacteristically low initial $^{234}\text{U}/^{238}\text{U}$ compared to the much larger range of older samples which were closer to 1. However, we calculated the uncertainty-weighted average in an earlier draft version of the paper and there was very little difference compared to the mean average, particularly considering the large uncertainty assigned to the initial $^{234}\text{U}/^{238}\text{U}$.

- How was the 2σ uncertainty (0.178) on the average initial $^{234}\text{U}/^{238}\text{U}$ activity ratio calculated?

This was calculated by using the standard deviation of all the initial $^{234}\text{U}/^{238}\text{U}$ data (see Supplementary Information) and multiplying the standard deviation by 2.

- Figure 3f: what does the colour of the filled data points correspond to?

The colour corresponds to the individual speleothems and matches the isotope box plots below. We have clarified this in the caption.

Lines 334-35: "Ages of gypsum 'crust' samples indicated by * and the colours of the filled symbols refer to the individual speleothem isotopic composition shown in the box plots in Panel g."

- Lines 276, 461 and 470 (in marked up version): probably the authors mean speleothem formation instead of speleogenesis (= cave formation, Fairchild & Baker 2012)?

We have changed all instances of "speleogenesis" to "speleothem formation" as suggested.

- Lines 66 (in marked up version): I'd suggest changing the wording here and use "speleothem ages" (or similar) instead of "dates"

We could not find a reference to "dates" in line 66, but we assume that Reviewer 2 meant line 661 of the marked-up version of the manuscript. We have changed all references to "speleothem dates" to "speleothem ages" as suggested throughout the manuscript.

- Lines 1113-1116: just to be specific, probably better to explicitly say that "234U/238U" is an activity ratio.

We have added the words "activity ratio" to explicitly state this point, as suggested.

Lines 603-605: "The $^{234}\text{U}/^{238}\text{U}_{\text{initial}}$ was estimated from the average measured $^{234}\text{U}/^{238}\text{U}$ activity ratios on younger (<1.5 Ma) U-Th and U-Pb dated samples to calculate the $^{234}\text{U}/^{238}\text{U}_{\text{initial}}$, based on the isotope decay constants yielding an average value of 1.013 ± 0.178 (2σ)."

Also, I suggest adding to this sentence that samples younger than 1.5 Ma were calculated with speleothem-specific initial [234U/238U] values obtained from the solution U-Th measurements, and that only the samples older than 1.5 Ma have been corrected for initial disequilibrium using an average initial [234U/238U]

Thanks for this comment. We have added the following sentence as suggested.

Lines 605-607: "The ages for samples <1.5 Ma were calculated using speleothem-specific initial $^{234}\text{U}/^{238}\text{U}$ values obtained from solution U-Th measurements and samples older than 1.5 Ma have been corrected using an average initial $^{234}\text{U}/^{238}\text{U}$."

Referee #3 (Remarks to the Author):

The manuscript is much improved, and I find the expanded presentation of the speleothem data even more compelling. The authors addressed my previous comments in full in their revision (down to presenting cave names in Arabic, which I appreciate). The faunal biogeographic discussion has been appropriately shortened. I still have some concerns about the presentation of the faunal context, but these are minor at this point. Comments follow:

We thank Reviewer 3 for their positive comments on the revised version of our manuscript. As the reviewer notes, we have shortened the biogeographic discussion. We have further addressed the additional minor concerns from Reviewer 3 about the faunal context in our responses below.

Line numbers refer to the tracked changes pdf.

169: the reader expects Fig 1 to show biogeography, but instead it shows rainfall

We thank Reviewer 3 for pointing this out, we left the reference to the previous figure here. We have deleted the reference to Fig. 1 as suggested.

170: are refs 3 and 14 really the right ones here? It seems you are discussing modern biogeography, and for that a reference like Holt et al 2013 might be the right one (or Kraft & Jetz's comment, even though I still think the original paper should be cited).

Thank you for this comment, we agree and have removed those references which are more related to the palaeobiogeography and have instead added the Holt et al 2013 reference as suggested.

FIG 1. B and C are hard to understand in terms of percentages. Would you consider converting these to mm/yr amounts?

Thank you for this comment. The reason we used % here is to show the dominant seasons of contributing to total precipitation over the Arabian Peninsula and that our study area sits in the middle of two moisture bearing sources. Overall, the precipitation map (see below) shows that there is very little precipitation across the peninsula in terms of total precipitation (mm) (which is shown in Fig. 1a). Consequently, it is much more difficult to see the seasonality of precipitation when it is plotted as totals (see below). For this reason, we wish to keep precipitation plotted as percentages as a key theme of our paper is demonstrating a shift in the source of moisture bearing systems to central Arabia over the last 8 Ma. Further, we wanted to incorporate the map below into the Extended Data section in response to Reviewer 3's comment, but unfortunately, we are at the limit of items that can be included in the Extended Data section according to *Nature's* guidelines.

[FIGURE REDACTED]

211: Peninsula with capital P?

We are not 100% sure what Reviewer 3 meant here, but we assume this comment is asking whether peninsula should have a capital p in this instance? As we are using it with respect to the “Arabian Peninsula” i.e. a proper noun, both Arabian and Peninsula should be capitalised.

243: ‘8Ma BP’ - BP is not necessary.

We have deleted “BP” as suggested.

244-245: ≥ 100 mm/yr puts it in arid climate, not hyperarid (< 100). Or borderline, but not “one the most hyperarid parts of modern central Arabia”. From Fig 1 it’s even a relatively humid part of the Peninsula considering most of appears to receive 0 mm.

Thank you for this comment, it’s always good to be diligent when defining areas and we have double checked how we classified aridity thanks to Reviewer 3’s comment. The classification of aridity widely depends on the definition used. The most commonly used aridity scale is the Aridity Index defined by UNEP in 1992. This is also the index that the IPCC use in their assessment reports. As such, we thought this the most appropriate scale to use. UNEP define hyperarid as an area with an aridity index of < 0.05 . Our study site has an estimated annual PET of ~ 2170 mm (Taher, 2004) and annual mean precipitation of 104 mm which yields an aridity index value (P/PET) of 0.048 and thus falls into the ‘hyper-arid’ classification. So, we have consequently kept the definition of ‘hyper-arid’, but we have changed the phrasing of the sentence to reflect that it is not the most hyperarid part of Arabia, in line with Reviewer 3’s comment.

Lines 82-83: “Our study area is in a hyperarid (~ 104 mm a^{-1} ; Fig. 1) part of modern central Arabia, positioned under the descending arm of the Hadley Cell..”

Fig 2. Could you place the panel titles a, b, c, etc on the left? It’s a bit confusing on the right.

We have changed this as suggested. See revised Fig. 2 below.

Also the apparent association of the data from d with the y-axis from c is confusing. I'd suggest more separation between panels vertically.

We have moved Panel c up vertically to separate it from Panel d as suggested (see revised Fig. 2 above).

Are the dashed lines needed in b? The grey bars should be sufficient.

The hominin timeline blocks refer to the first and last appearance dates and are further defined by the reference from Bobe et al., 2022 who provide a new and revised estimation of origination times of the hominins from the fossil record. The dashed lines were included to represent the fact that the earliest known fossil record almost always postdates the true origin period of a taxon, with the dashed lines representing the confidence intervals of the species first appearance. As this seems to be more distracting than informative, we have removed the dashed lines from the figure as per the suggestion of Reviewer 3 (see revised Fig. 2 above).

Also check some of your hominin age ranges: Sahelanthropus is proposed to be ~7.

Thank you for this comment. We initially used the reference of Vignaud et al (2002) which state a range of 6-7 Ma as the age of the fossil bearing sequence. However, we have now

updated this with a more recent reference (Lebatard et al., 2008) with a range of 6.8 to 7.2 Ma, which fits the ~7 Ma Reviewer 3 (we think) is referring to (see revised Fig. 2 above).

Ardipithecus is 4.4 to 5.6 Ma. Earliest Australopithecus is anamensis, around 4.1 Ma – why the dashed line going earlier?

We have removed the dashed lines in a response to a previous comment. We have checked the position of Ardipithecus, which is between 4.4-5.6 as Reviewer 3 suggested.

Also how is ‘Early Homo’ defined? If anything prior to Homo sapiens, then the bar should extend right to 300 ka or later.

The age range definition for early *Homo* in Fig. 2 includes *H. habilis* and *H. rudolfensis*. Here, we wanted to make the distinction that early Homo appears before *Homo erectus*, and then both existed at overlapping time intervals.

Is the hominin timeline even relevant for this article? This is described as ‘major junctions’ in the caption, but it’s basically a genus age range plot. Is there any correlation between the appearance/disappearance of hominin genera and the paleoclimate proxies shown? I think not.

We thank Reviewer 3 for bringing this up. While we have avoided making any correlations between our record and the hominin evolutionary timeline, we do believe it is important to provide the temporal context as the general audience is usually interested in the background within which our environmental record has evolved. This also ties into the longstanding potential relationship between environmental stressors and evolutionary adaptations. So, while we hesitate to draw any conclusions, we think it is useful to provide the overall context, further enhancing the significance of our findings. Further, we have changed the phrasing in the caption from “major junctions” to the following:

Line 308: “Temporal distribution of key events and hominin information based on appearance dates”

Why the ‘Old World Savanna’ bubble at 6-8 Ma with an arrow pointing earlier? Bear in mind the Old World Savanna Paleobiome is alive and well in places like sub-Saharan Africa and South Asia today, it was just more extensive in the past. I assume you are highlighting the peak of the OWSP (Late Miocene to Early/Middle Pliocene) but that is not evident. Not sure this is relevant to this figure either.

That is correct, we did mean the peak OWSP; however, we have removed this from the figure as suggested (see revised Fig. 2 above).

Panel f is missing a callout in the caption.

Thank you for spotting this omission. We have added “Panel f” to the caption in lines 1094-96.

Also, please state what the whiskers extending from the grey circles represent – the response to reviewers states these are 95% confidence intervals. That should be clearly stated in the caption.

Thanks for bringing that to our attention. We have added this to the caption as suggested.

Line 315: “with the 95% confidence interval uncertainty indicated by the horizontal error bars.”

“One age with an uncertainty greater than >1.5 Ma is not shown” - but one of your data points at ~4 Ma has an uncertainty > 2 Myr. Why is that one shown and the other left out? I would suggest not leaving any data out.

Thanks for looking at this in detail. We meant a $\pm >1.5$ Ma error, rather than a total error of 1.5 Ma. Reviewer 3 is correct that there is a speleothem age in our dataset at ~4 Ma with a total error > 2 Ma. We have clarified this in the caption. The one point we omitted was a replicate age already represented in the figure which had much lower uncertainty. We present the age in our results in the Supplementary Information for transparency, however, it is not very useful as the \pm error bars were ~ 3 Ma (almost covering the entirety of our record!). However, we also do note this in the caption for transparency again and we hope Reviewer 3 can understand the reasoning behind this.

Lines 315-316: “One age with an uncertainty greater than $>\pm 1.5$ Ma is not shown but can be found in the Supplementary Data.”

337-338: ~7.3 Ma until ~6.4 Ma... ~4.1 Ma until ~3.2 Ma – where are these numbers from? I suggest you do this quantitatively, by binning together all the ages including uncertainties of each humid intervals, and reporting the 95% age ranges.

We thank the reviewer for pointing out that this was unclear. We have clarified the text and provided the absolute (opposed to approximate) ages of the oldest and youngest sample in each grouping and reference these values to the ages on Fig 3 Panel f for clarity.

Lines 110-112: “...The oldest evidence for increased precipitation in our dataset occurred in the Late Miocene at 7.45 Ma (Fig. 2) until 6.25 Ma (Fig. 3f). A period of further humid episodes occurred from 4.10 Ma in the Early Pliocene until 3.17 Ma (Fig. 3f),..”

457: “centred around 2.1 and 1.0 Ma” - same here, provide a more quantitative value, like an average \pm uncertainty.

We have changed the text as requested and provided the range of quantitative equilibrium ages of the two main date clusters in the Early Pleistocene. Further, we have referenced Fig 3f with all the ages and respective uncertainties.

Line 113-114: “...identified in the Early Pleistocene, between 2.29-2.18 and 1.53-0.90 Ma (Fig. 3f)...”

636 “temperature gradients shifted to approximately modern-day values” - I understand from the reference cited that this was a temporary dip in temperature from 7-5.4 Ma. You might want to be clearer about this as a ‘temporary shift’ or ‘ephemeral dip’, rather than a ‘shift’.

We have changed this as suggested.

Line 229: “...temperature gradients temporarily shifted to..”

713: “Biogeographical significance” - I still find this section too speculative and not contributing much to the main findings of the article. Comments follow.

We thank reviewer 3 for their comments on this section. Accordingly, we have further shortened the biogeographical significance section (now only two paragraphs), to further deemphasise this as a main part of the paper. However, our multidisciplinary group of authors do believe it is warranted to provide some regional and global context for our environmental findings, such as noting the significance of the Baynunah fossil evidence. Further, we think it is important to note the previous palaeontological and environmental work in this area to highlight the significance of what our new findings suggest. For example, the past work describing the Pliocene as ‘a period of extreme hyper aridity and hence its consequential role as an extreme physical barrier for admixture and exchange’. In contrast, our data provides compelling in situ evidence that this was not always the case.

We have further addressed all Reviewer 3’s specific comments in our responses below and removed any potentially problematic text that has been highlighted.

729-735: the guenon, or primates in general, are negligible for habitat reconstruction in these assemblages. You already mentioned the hippos, giraffes, crocodiles, etc which indicate the presence of permanent water and grassland-woodland vegetation (ie ‘savannas’). There is nothing that ties the Baynunah specifically to the Chadian sites, except that they are of similar age and had similar large mammals. The same could be said of any number of Late Miocene sites from China to South Africa. Certainly the guenon is not a point of similarity.

We have removed the last three sentences in this paragraph which refer to the guenon and its similarities to *Sahelanthropus*. We agree with Reviewer 3 that the evidence of a savannah type environment in Arabia at this time is already established from the other taxa fossils discussed earlier in the paragraph.

734: “enough precipitation to support vegetation suitable for hosting primates” - here too the emphasis on the primates is misplaced. Hippos, crocodiles, river turtles, and fish need a lot more water, and are present in far higher abundances. If anything, monkeys are generally more indicative of trees than water. Maybe you mean “suitable for hosting a high diversity of large mammals” or similar.

We have removed this sentence in response to the previous comment, but incorporated the suggested text above into an earlier sentence as follows:

Lines 250-252: “...Formation (~7.7-7.0 Ma)², indicated that terrestrial Arabia was suitable for hosting a highly diverse array of large mammals...”

741-774: Quite right. Any argument for “a sustained arid barrier in northern Arabia (33° N) between 5.6 and 3.3 Ma” is using an absence of evidence as evidence of absence. There are simply no terrestrial deposits of that age in which to find fossils on the Peninsula (as far as we know) and that probably has more to do with regional tectonics (uplift / erosion / lack of deposition) than any kind of evidence for large mammals not being present. Your speleothem record is spectacular precisely for this reason.

We appreciate the comment and agree an absence of evidence is not evidence of absence and agree with Reviewer 3 that our speleothem record provides a unique record and robust information about past in situ humidity in this region, prior to which there was very little data.

780-781: reduction, rather than a breakdown. Also not just in Africa, but South Asia as well. And I think you mean genera Homo and Australopithecus, not ‘early hominins’ as

those were also around in the Late Miocene. Though here too I fail to see the significance of the hominins to the argument.

Thank you for the comment regarding clarification of the Savannah Biome. We have replaced the word 'breakdown' with 'reduction', as suggested. We have further clarified that we are referring to *Homo* and *Australopithecus* when we referred to 'early hominins' in the previous version, and have clarified that as suggested.

Lines 272-73: "led to the reduction of the Old World Savannah Palaeobiome and the development"

Also, in the Methods (former Supplement section)

Lines 881-882: "...during the Neogene and Quaternary and following the reduction of the 'Old World Savannah Palaeobiome'³¹"

782: "progressively more difficult to traverse and settle over time due to the gradual aridity trend" - a bit too speculative. More difficult for whom? Crocodiles maybe. But not oryx. And humans/hominins also had no problem occupying and presumably traversing the Peninsula at different times. Your data very nicely document a trend of increasing aridity, but what is the relationship between that trend and the ability of mammals to traverse the region? Aridity thresholds will be significantly different for different species, and can be fully modulated by seasonal waterways and lakes, especially on geological timescales. Bear in mind also that the Peninsula was probably always traversable along coastal / mountain routes, regardless of how dry the interior was.

We have removed this part of the sentence "progressively more difficult to traverse and settle over time due to the gradual aridity trend" and instead highlighted the progressive aridity through time, therefore removing the speculation that Reviewer 3 is concerned about.

Lines 279-281: "Our new data suggest that over this same period, Arabia became progressively more arid through time (Fig. 3g), triggered by a reduced contribution of monsoonal rainfall (Fig. 4)."

783-789: I find these two sentences contradictory. The first states rightly that mammals were present in the dry Mid-Late Pleistocene, and the second still tries to claim that the new climatic record identifies 'constraints' on range expansions. It's true the speleothem record is "providing insight into critical periods for which previously there was no data" but this is being conflated with data on fauna and biogeography, for which there is none. If anything, the sparse Arabian fossil record confirms that large mammals thrived there anytime we have the right deposits to find them in (Oligocene, Middle Miocene, Late Miocene, Mid-Late Pleistocene). As noted above, their absence is likely the result of geological preservation bias, not true absence.

We appreciate Reviewer 3's concerns here and have adjusted the phrasing in the paragraph as suggested. Specifically, we have removed the text: "difficult to traverse and settle over time due to the gradual..." and "that allowed dramatic biogeographic range expansions and local extirpations in the Arabian interior". Now the paragraph simply references existing literature, which states that there is increasing fragmentation of and reduced fauna exchange through time, and then we discuss the aridity trend that we subsequently observe in our record.

Lines 269-281: “Fossil evidence of interconnectivity of mammalian assemblages over the Late Neogene show that longitudinal dispersals across the ‘Old World Savannah Palaeobiome’ were favoured in the Late Miocene, but became increasingly latitudinally fragmented from the Pliocene onwards, with an overall trend towards reduced faunal exchange¹¹. This eventually led to the reduction of the Old World Savannah Palaeobiome and the development of Plio-Pleistocene African savannah fauna, including the presence of early *Homo* and *Australopithecus*¹¹. Our new data suggest that over this same period, Arabia became progressively more arid through time (Fig. 3g), triggered by a reduced contribution of monsoonal precipitation (Fig. 4). However, despite drier humid intervals in the Pleistocene relative to the Pliocene and Miocene, the presence in Arabia of both African and Asian mammals, particularly those with large water requirements, during humid periods³ in the Middle-to-Late Pleistocene indicates that dispersals were still possible. Our record constrains both the timing and frequency of such climatic amelioration phases, providing insight into critical periods for which previously there was no data.”

Referee #4 (Remarks to the Author):

After getting across the manuscript and the rebuttal letter, it seems to me that the authors managed to address most of the criticisms from the reviewers regarding the analysis and statistical significance of the dataset. They did so by either doing suggested analysis or by reformulating the manuscript. I would however let a reviewer whose interpreting U/Th and U/Pb data is the specialty to comment on this aspect of the paper. Authors also removed most of the paragraph on biogeography while keeping the synthesis they performed in the Supplementary Material and they instead increased the climate implication of their dataset. Biogeography is now more an opening for the paper which I think is good due to strong limitations of biogeography records from the region pointed out by one of the reviewers.

We thank Reviewer 4 for recognising that we have addressed most of the comments in the previous version of this manuscript.

One of the concerns raised was also that the ‘low’ number of samples would prevent identification of significantly humid periods. I do not think this should prevent the manuscript to be considered for publication as my recollection is that the size of the dataset is still impressive for this type of study in a place where barely any other records exist for the time period. Because of the latter, the results clearly have a strong importance for the understanding of climate evolution and climate dynamics of the entire northern Africa/Arabia region. In particular even with large uncertainty in the ages, the trends they identify in the $\delta^{18}O$ and δ^2H are by themselves interesting results, no matter the interpretation the authors have of them. Those could indeed motivate better integration of regional scale data by the paleo-climate community to gain insight into the regional Mio-Pliocene evolution of the ITCZ.

We agree with Reviewer 4's summary that the dataset is impressive, and we have doubled the number of ages from the previous version. Further, we agree that this data could motivate better integration of regional scale data by the palaeo-climate community to gain insight into the Mio-Pliocene evolution of the ITCZ.

The effort the authors made in this version of the manuscript not to oversell too much the data should be acknowledged. They for exemple now better underline there are still gaps in the understanding of the regional climate and limitations to the dataset but still managed to draw an interesting story out of it. In particular, the authors reconsider the way to refer to the humid intervals they described by replacing the formal naming they initially gave (CAHPs) for ‘humid periods’ which emphasizes to my opinion the uncertainty in the duration or frequency of such events.

We thank Reviewer 4 for acknowledging the considerable effort we have made in the revised version not to ‘oversell’ the humid episodes identified and we have focused more on a discussion of the pulsed nature of this episodes in our discussion in the revised manuscript.

I however think that now that the authors shifted the discussion more toward climate implications of their results they need to be more precise in the description of the climate system processes before the paper could be published (see my comments below).

We thank Reviewer 4 for their detailed look at our revised version and provide responses to all their specific concerns below:

Comments :

L. 54. In the abstract the authors state that ‘Recent research places the emergence of this barrier at 11 million years (Ma) ago’, referring to Crocker et al. (2022) paper. This statement is incorrect: Crocker et al. published a record that started at 11Ma as the title of the paper states it, therefore the record does not provide any indication on the regional environment prior to 11 Ma. A more exact formulation would be that ‘Recent research suggests this barrier has been in place since at least 11 millions years ago’

We have changed this as suggested in both instances where it is mentioned in the manuscript.

Lines 8-9: “Recent research suggests this barrier has been in place since at least 11 million years (Ma) ago¹.”

Lines 43-45 “Recent research suggests that drying across this desert barrier may have begun in the Late Miocene with fully arid conditions in the Sahara from at least 11 Ma¹ and hyper-aridity in the northern Arabian margins starting at 9 Ma¹³.”

L. 91. The authors now state that “Widespread aridification instigating C4 grass dominance in low and mid-latitude open environments has traditionally been linked to Quaternary cooling in the Northern Hemisphere (NH), triggered by the onset of glacial-interglacial cycles, which intensified at ~2.6 Ma.”. I might have misunderstood but I find this sentence confusing as the literature seems to place this transition way before the Quaternary, already in the late Miocene (see the Ceerling et al. (1997) and Herbert et al. (2016) reference authors cite in one of the previous sentence, though they have been new data published regarding this transition since then (see Tauxe and Feakins (2020) for review)). I therefore suggest the authors rephrase this part of the paragraph so it does not sound in opposition to the cited literature.

Many thanks for this comment, we appreciate this could be clearer. We were referring to two different late Cenozoic cooling phases, the long-term phase starting in the Late Miocene and the documented Plio-Pleistocene cooling/aridification phase thereafter. Some authors suggest this Neogene cooling and aridification trend is interrupted by Pliocene humidity (i.e. Sniderman et al., 2016). We have added an additional reference for the Plio-Pleistocene aridification shift due to changes in the Walker and Hadley Circulations: Etourneau et al (2020). Further we have clarified that we are talking about the late Cenozoic generally to include both the long-term drying shift that started in the late Miocene as well as a step change shift towards more aridification in the Plio-Pleistocene transition. The revised paragraph is as follows:

Lines 43-61: “Recent research suggests that drying across this desert barrier may have begun in the Late Miocene with fully arid conditions in the Sahara from at least 11 Ma¹ and hyper-aridity in the northern Arabian margins starting at 9 Ma¹³. This aridification appears in conjunction with an overall global expansion of tropical and sub-tropical deserts as a feature of long-term Late Cenozoic cooling. An acceleration in aridification is also evidenced at the Plio-Pleistocene transition and is associated with further cooling in the Northern Hemisphere (NH), triggered by the intensification of glacial-interglacial cycles at ~2.6 Ma together with atmospheric circulation changes¹⁵⁻¹⁷. Global cooling also marked the start of a pronounced global shift in terrestrial environments¹⁷, in which ecosystems transitioned from those dominated by plants following the C₃ photosynthetic pathway (e.g., trees, herbs, shade-loving grasses) to those dominated by C₄ plants (e.g. arid-adapted grasses)¹⁸ (Fig. 2). Although the increased distribution of arid-adapted vegetation and, consequently, C₄ mammalian grazers, has often been attributed to higher aridity, another plausible explanation is the ability of C₄ plants to outcompete C₃ in many parts of the world under decreasing atmospheric CO₂ levels where C₄ photosynthesis is physiologically advantageous¹⁹. As there are limited terrestrial records of water balance in continental desert interiors, it accordingly remains challenging to assert whether changing precipitation regimes are

responsible for the aforementioned shift in vegetation, and further, whether this aridification state was a permanent feature of the Late Cenozoic. This leaves the driving mechanisms behind what appears to be a key Late Cenozoic aridity trend unresolved.”

L. 94-95. ‘with fully arid conditions in the Sahara from 11 Ma’ : in line with the previous comment, I suggest the author to rephrase as ‘with fully arid conditions in the Sahara from at least 11 Ma’

Changed as suggested.

Line 43: “fully arid conditions in the Sahara from at least 11 Ma⁵ and..”

Figure 1 : Could the authors specify which months are included as ‘Autumn-Winter’ and ‘Spring-Summer’ respectively ? This is because most of the time the definitions of seasons are not following rules in paleo-climate papers at least.

We have added the months included in the ‘Autumn-Winter’ and ‘Spring-Summer’ definitions in the caption as follows:

Lines 295-296: “...Percentage contribution of autumn and winter precipitation (months September-February) and of spring and summer precipitation (months March - August) in Panel c...”

In addition, I strongly encourage the authors to add the main directions of the present day regional moisture transport they described lines 126-128 on Figure 1b and 1c. This is because this information is central for their discussion and having it display graphically would help the reader to understand.

We have included the predominant winds associated with the different seasons of rainfall in a revised Fig. 1b and c. as suggested (see below).

[FIGURE REDACTED]

L. 124-128. Because the authors decided to shift the paper discussion more toward climate significance of their findings, they need to be more precise in describing the moisture transport pathway in the region. For example, the text now reads “between westerlies to the north and east and the monsoon to the south and west receiving”; and while this is easy for the reader to understand where the winds comes from in the north and east part of the region, the reference to monsoon is a bit less straightforward and it would be more clear to provide additional information on the moisture provenance. I understand that there are character limitations, but there are some parts of the discussion that could be better organised to improve the fluidity of the manuscript and save space, as I mentioned in some of my comments below. I noticed that part of this necessary piece of information is in the Supplementary Material but my opinion is that it belongs to the main text as soon as an emphasis is put on the climate implication of the records.

We have substantially reorganised our discussion around the specific results and climatic implications that arise from our study. We now have sections entitled “Recurrent humid episodes” (where we incorporate orbital forcing controls on when humid periods are likely to occur and the alternating wet-dry and short-lived nature of these events), “Precipitation moisture sources”, and “Progressive aridity over the last 8 Ma”. We believe our discussion as a result is much more centred around the climate implications of the record. Further, our discussion in the Supplementary which contains more detailed information on the climate dynamics evoked in this paper has now been moved more centrally to the Methods section.

Regarding the specific comment on the sentence about moisture sources: We have clarified this to be more specific about the two origins of precipitation in winter vs summer, as well as adding the wind directions to Fig. 1 as requested above.

Lines 84-88: : “The region sits between two major moisture-bearing systems: mid-latitude westerlies, which originate in the eastern Mediterranean and traverse the Peninsula during boreal winter months (Fig. 1b) and monsoon derived precipitation from the southwest, which is limited to the southerly parts of the peninsula during boreal summer (Fig. 1c).”

One might consider it not highly relevant from present day climate perspective but because the region the authors study is located at the border of 2 precipitation systems (Figure 1b-c), one might also imagine that northward or southward shifts of the winds belts might make it more sensitive either to westerlies or to winds coming from the Indian Ocean, modifying regional moisture and thereby precipitation dynamics. This is what the authors tentatively describe in the discussion of the paper (lines 235-235) but fail to integrate in the large-scale atmosphere dynamics that is brought out in a later paragraph (from line 247).

We thank the reviewer for their comment regarding the integration of our findings directly with the large-scale atmospheric dynamics processes discussed in the text. We have now included a separate section on precipitation moisture sources where we have made the changes requested. Specifically, we have amended this paragraph to provide links to changes in the wind belts with our isotope data.

Precipitation moisture sources

Lines 142-168: “Over the last 8 Ma, there has been a shift in moisture-bearing systems delivering precipitation to central Arabia revealed by the oxygen ($\delta^{18}\text{O}_{\text{FI}}$, ‰ VSMOW) and hydrogen ($\delta^2\text{H}_{\text{FI}}$, ‰ VSMOW) isotope ratios of speleothem fluid inclusion waters (‘fossil dripwater’). This is important as the origin of moisture sources allows links to large-scale atmospheric circulation patterns, revealing the key mechanisms driving hydroclimate. The modern meteoric local waterline (Fig. 1a) reflects a mixture of precipitation derived from summer monsoon south-westerly and winter mid-latitude (Mediterranean) north-westerly sources (Fig. 1b,c). In contrast, the fossil $\delta^{18}\text{O}_{\text{FI}}$ and $\delta^2\text{H}_{\text{FI}}$ fluid inclusion waters from speleothems generally plot on the Global Meteoric Water Line, synonymous with a southerly moisture source (SMWL; Fig. 4b) and distinct from the modern local meteoric waterline (Fig. 4a). This strongly indicates a different precipitation pattern during these older humid episodes, relative to present (Fig. 4). These palaeo- $\delta^{18}\text{O}_{\text{FI}}$ and $\delta^2\text{H}_{\text{FI}}$ values are instead consistent with reported speleothem fossil water data from southern Arabia dominated by a tropical moisture source during recent insolation-driven peak interglacials^{9,24,29}(S-LMWL; Fig. 4b) and are similar to those commonly associated with low-intensity stratiform monsoonal precipitation³⁰. In Arabia, monsoonal winds bring greater moisture transport from the nearby Gulf of Aden and Arabia Sea source regions³¹, and are attributed to an increase in the zonal wind component over Africa and the Arabian Peninsula during wetter intervals. Further, the resulting precipitation³¹ has a lower isotopic composition that is in part due to an amount effect. Fluid inclusion isotope data indicate a general trend towards higher $\delta^{18}\text{O}_{\text{FI}}$ and $\delta^2\text{H}_{\text{FI}}$ values through time, consistent with a gradual move towards the present state (Fig. 4a), where annual precipitation is delivered by a combination of both winter and summer moisture sources and associated with very low overall amounts of precipitation due to its location at the boundary of the two atmospheric delivery systems (Fig. 1c, b). We ascribe this trend to the time-transgressive reduction in the contribution of southerly-sourced precipitation and influence of the monsoon to the Arabian interior over time during humid episodes.”

L. 154. paedogenic → pedogenic

Changed as suggested.

Figure 2e) Please indicate the meaning of error bars also in the figure caption.

This has already been addressed in a previous response.

Figure 3. Arabian Sea SST especially from the Miocene are derived from Uk'37, with temperatures for the late Miocene that are close to the saturation limit of the proxy. Therefore, the gradient that is derived is likely to be under-estimated for the pre-late Miocene cooling part of the record. I think it would be good to acknowledge it somewhere, maybe in the SI, although it won't change anything in the interpretations as the cooling toward present day and increasing gradient is not a matter of debate.

This is a good point raised by Reviewer 4. We have added an additional sentence to acknowledge this issue, but agree it does not change any of the interpretations in this paper.

Methods section Lines 748-750: **However, it should be acknowledged that there is attenuation of the Uk'37 response to temperature as the ratio approaches one, and the Uk'37 alkenone proxy becomes saturated when SSTs are above 28°C¹³ potentially leading to underestimates of the MTGs before the late Miocene cooling."**

L. 223. I suggest the authors find a different title to the paragraph so it relates better to the content. Indeed, they here mostly discuss sources of moisture rather than precipitation dynamics.

We have removed this heading as suggested. In our previous responses above we have restructured the discussion with new headings (see previous comment) and this section is now more appropriately entitled "Precipitation moisture sources" following the suggestions of Reviewer 4.

L. 232-233. The authors now state that 'The modern meteoritic local waterline is dominated by low precipitation amounts (Fig. 1a) from a mixture of summer and winter sources (Fig. 1b,c)'. The last part of the sentence is unclear. The authors should there specify what they mean by 'summer' and 'winter' sources respectively, especially because they suggest moisture source is changing over seasons. As mentioned in one of my previous comments, this would be easier to emphasize if the large-scale moisture pathway is identified on the figures 1b and c.

We have clarified the source regions as suggested and also added the wind patterns to Fig 1b and c.

Lines 147-152: **"The modern meteoric local waterline (Fig. 1a) reflects a mixture of precipitation derived from summer monsoon south-westerly and winter mid-latitude (Mediterranean) north-westerly sources (Fig. 1b,c). In contrast, the fossil $\delta^{18}\text{O}_{\text{FI}}$ and $\delta^2\text{H}_{\text{FI}}$ fluid inclusion waters from speleothems generally plot on the Global Meteoric Water Line, synonymous with a southerly moisture source (SMWL; Fig. 4b) and distinct from the modern local meteoric waterline (Fig. 4a)."**

L. 237. - Fig. 4 → Fig 4b, so the reader knows directly where to look.

Changed as suggested.

Line 151 "Fig. 4b"

L. 237-238. "These data plot on a modern meteoric waterline derived from a tropical moisture source (Indian Ocean) (Fig. 4b)". It seems to me that tropical moisture source

(Indian Ocean) is a reference to S-LMWL (Oman) in the figure. It would then be easier to read the figure and understand what the authors are explaining if they were also referring to that abbreviation directly in the text.

We have added the specific reference to the S-LMWL in Fig. 4b to the sentence. Further we have clarified the similarity between the GMWL and the SMWL.

Lines 149-152: “In contrast, the fossil $\delta^{18}\text{O}_{\text{FI}}$ and $\delta^2\text{H}_{\text{FI}}$ fluid inclusion waters from speleothems generally plot on the Global Meteoric Water Line, synonymous with a southerly moisture source (SMWL; Fig. 4b) and distinct from the modern local meteoric waterline (Fig. 4a).”

Lines 153-157: “These palaeo- $\delta^{18}\text{O}_{\text{FI}}$ and $\delta^2\text{H}_{\text{FI}}$ values are instead consistent with reported speleothem fossil water data from southern Arabia dominated by a tropical moisture source during recent insolation-driven peak interglacials^{9,24,29}(S-LMWL; Fig. 4b) and are similar to those commonly associated with low-intensity stratiform monsoonal precipitation³⁰.”

L. 238-239. Now the authors state “Our data reveal that the tropical rain-belt repeatedly reached our study area currently at 26° N, bringing precipitation of African/Indian origin”. Africa and Indian regions seem to be two opposite directions relative to the site location. It is therefore not super clear what the authors mean here.

In our restructuring of these paragraphs, we have removed this sentence.

L. 235-245. My understanding is that a lot of different processes are usually involved in explaining shifts in water isotopes including but not limited to change in precipitation regimes and moisture source. Change in temperature can also affect the isotopic signature especially when comparing warm periods like the late Miocene and the Pleistocene or change in the isotopic composition of the source can also play a role without requiring change in source by itself. I think the argument would be stronger if the authors first expand a bit on all plausible explanations for isotopes shift they see in the record and explain why they retain the moisture source shift. Those hypothesis would be in addition probably easy to test with already published water-isotope enable simulations (eg. Knapp et al. 2022, <https://zenodo.org/records/6953979> as an example for the Pliocene).

We appreciate Reviewer 4’s request for a more detailed description of the isotopic controls and changes in precipitation patterns. We have adjusted this section from the comments here and additional comments below and changed the text to centre around key aspects of this paper i.e. moisture sources and the aridity shift during humid episodes. We reference paleoclimate modelling studies which suggest lower isotopic composition of precipitation in the peninsula during wetter intervals which is in part due to an amount effect, and similar to the data in our study. Further, our data also correlate well to the fluid inclusion paleoclimate record in southern Arabia during pulsed humid phases in interglacial periods which suggests increased moisture from the south during the summer monsoon during humid phases. We now discuss the moisture sources earlier in the manuscript and then shift towards the aridity trend and the larger atmospheric circulation systems responsible. Please refer to the sections covered in Lines 129-242 where these changes have been made.

In addition, this paragraph is appearing very confusing when reading the ‘Meridional temperature gradient’ paragraph (l.247) just after. Here, different climate phenomena are mixed here without much explanations and lack at this point of the text integration into the large scale atmosphere dynamics so the reader understands clearly the reasoning. For example authors first aligns their data points with different water meteoric lines and late Pleistocene record from further south to discuss about moisture source but then mentioned shift in precipitation regime for the Pliocene and Miocene, which are two different processes but that might probably both be related to Hadley circulation and ITCZ. I think the authors have a plausible explanation that relates the isotopes signal, regional

atmosphere dynamics and global climate but they fail at properly accounting for it because the way the two paragraphs are structured.

We agree with Reviewer 4 and have already implemented this change in our response above (please refer to Lines 129-242). Specifically, we have reorganised the structure of our discussion of moisture sources and the large-scale atmospheric processes. We have included sentences to clarify that we interpret the fluid inclusion isotopes as a recorder of the original drip water composition and that subsequently, we can compare this to modern meteoric waterlines. This shows that most of our palaeo-water align with moisture from a southern source, but there appears to be a trend towards modern values from a mixed source through time. Importantly the modern precipitation comprises of very little precipitation from either of the aforementioned moisture sources (i.e. hyperarid). We ascribe this trend to the time-transgressive reduction in the contribution of southerly-sourced precipitation and influence of the monsoon to the Arabian interior over time during humid episodes. The gradual loss of moisture from the monsoon, is indicative of a gradual decrease in effective moisture availability during humid episodes.

The text would be improved by reformulating the paragraph on ‘Reduced tropical precipitation’ and better integrating it with the statements from the ‘Meridional temperature gradient’ part just below. This would enable to reduce text length and provide a more consistent picture integrating between large scale patterns and regional climate, which would help to better explain the isotopic data. For example mentioning the shift in ITCZ in Pliocene climate simulations (lines 253-255) straight after mentioning the very low $\delta^{18}\text{O}_{\text{FI}}$ and $\delta^2\text{H}_{\text{FI}}$ (lines 240-242) and Pleistocene contraction of Hadley cells following Pleistocene FI data (lines 243-245) would help.

We thank reviewer 4 for this pragmatic and helpful suggestion and have restructured the paragraphs as mentioned already above and believe this has improved the flow of the manuscript. For example, below we address the need to incorporate the discussion of Hadley Cells with respect to the results of our study directly in the text below:

Lines 205-214: “Higher meridional SST gradients are associated with the Pleistocene and Late Miocene global cooling intervals and are strongly coupled with NH polar ice extent (Fig. 3) and Hadley Circulation contraction, bringing largescale subsidence and prolonged dry conditions over Arabia. NH cooling is reflected in our record by the gradual shift to higher $\delta^{18}\text{O}$ values, both in speleothem calcite and fluid inclusion water, which provides evidence for a gradual shift towards precipitation originating from mid-latitude westerlies, and reduced effective precipitation during interglacials, particularly after the MPT, due to the reduced influence of the summer monsoon. This change in the position of the ITCZ and atmospheric circulation patterns in the Saharo-Arabian Desert are also evidenced by increased dust transport from the Sahara Desert from 2.3 Ma¹ (Fig. 3d) and greater terrigenous input into the Arabian Sea³³

L. 247. and following. I wonder why the authors do not make a better of other Arabian records (Böhme et al. (2021) or others) that they cite in the introduction of the paper and in the biogeography paragraph and that cover more or less the same period. Because those records are located in different types of regional climate zones at present day combining both could serve as an argument to infer change in the regional climate and link this to shift in large scale atmospheric patterns.

We refer to Nicholson et al 2020, Fleitmann et al 2022 and Fleitmann et al. 2003 in our comparison to in situ hydroclimate records for Arabia in our discussion on fluid inclusions and their interpretation. The Bohme record in this way does not really link to ours so well as it is much further north and may be dictated by different atmospheric processes, but more importantly, influenced by

precipitation from distal sources as far away as Turkey (as it is a riverine deposit). This ambiguity makes it more difficult to draw regional climate shifts in atmospheric circulation patterns. Consequently, we believe it better suited to be discussed in the Biogeographical significance section.

I note that the information provided in the paragraph that has been added I. 263-274 improved the manuscript.

We agree and thank reviewer 4 for this comment.

L. 286-291. The paper cited here to justify potential impact of Mediterranean Sea and red sea shrinkage during the Messinian crisis here, Zhang et al. (2014), does not test a scenario for desiccation of the Mediterranean sea. I would rather refer to Ivanovic et al. (2014) that actually tested the hypothesis of Mediterranean Sea shrinkage on the climate, though it seems that the mean annual precipitation change they simulated over Arabia was insignificant.

This is an excellent suggestion from Reviewer 4 and we have changed the Zhang et al (2014) reference and instead used the Ivanovic et al. (2014) reference and incorporated this in the discussion.

Lines 234-242 “The catastrophic shrinkage of the Mediterranean and Red seas³⁸ is likely to have contributed to regional aridity by reducing the amount of moisture available as a direct result of changes in regional land-sea distributions. Further, modelling of the Mediterranean-Atlantic exchange has revealed a significant influence on the North Atlantic, where the Messinian Salinity Crisis may have induced mid-high latitude cooling by a few degrees Celsius and a decline in the Atlantic Meridional Overturning Circulation³⁹. Higher induced meridional SST gradients (Fig. 3) provide a mechanism for a temporary equatorial excursion of the ITCZ away from the Arabian Peninsula during the Messinian Salinity Crisis in the terminal Miocene.

In addition, and regarding Zhang et al. (2014) paper result, I would consider another possible (or complementary) explanation for the supposed late Miocene aridity in the region. There has been large variations in water body in the Paratethys region (eastern Europe) during the interval (eg. Krijgsman et al. 2010) that could have altered the dynamics of ITCZ in the same manner Zhang et al. (2014) described for the long term transition from early Miocene to present-day. I have not reviewed the literature on the topic but timing of such events seems to fit and I therefore do encourage the authors to look for appropriate literature on this.

This is a good point by Reviewer 4, as already shown in our amended text in our response directly above, we have since reviewed further literature on the Messinian Salinity Crisis and in particular its relation to the Atlantic Meridional Overturning Circulation. Cooling of the North Atlantic has been shown to be linked to reduced saline outflow from the Mediterranean basin to the Atlantic during periods of Mediterranean Sea desiccation, potentially influences meridional temperature gradients and provides a further mechanism for a potential equatorial shift in the position of the ITCZ. Although the reference limit means we cannot expand further with more literature on this topic (i.e. Rogerson et al, 2012), Ivanovic et al. (2014) provides a very good summary. We have since modified this paragraph and included this as a potential explanation as follows:

Lines 234-242:

Lines: “The catastrophic shrinkage of the Mediterranean and Red seas³⁸ is likely to have contributed to regional aridity by reducing the amount of moisture available as a direct result of changes in regional land-sea distributions. Further, modelling of the Mediterranean-Atlantic exchange has revealed a significant influence on the North Atlantic, where the Messinian Salinity Crisis may have induced mid-high latitude cooling by a few degrees Celsius and a decline in the Atlantic Meridional Overturning Circulation³⁹. Higher induced meridional SST gradients (Fig. 3)

provide a mechanism for a **temporary** equatorial **excursion** of the ITCZ away from the Arabian Peninsula **during the Messinian Salinity Crisis in the** terminal Miocene.

L. 291. Mentioning ‘Tethys shrinkage’ is an incorrect (or incomplete) formulation even though Zhang et al. (2014) paper is making extensive reference to this. Zhang et al. (2014) actually accounted for two significant regional events in their geography 1) the closure of the Tethyan seaway between Mediterranean Sea and Indian Ocean - that it now mostly thought to have happen before and during the mid-Miocene though (see for example Bialik et al, 2019), and 2) the shrinkage of the Paratethys that is a more complicated event as there have been persistent lakes with episodes of large scale flooding event during the late Miocene/ early Pliocene (see for example Krijgsman et al. (2010) or Lazarev et al., (2021) among others). More exact formulation would therefore be ‘the Paratethys shrinkage’ or to a greater extent ‘the Tethys Sea shrinkage’.

We have removed all references to the Tethys specifically here in our responses to the previous comments from reviewer 4, so this is no longer relevant

L. 291-293. Because the authors states that the mid-Pliocene is likely to have been more humid and this sentence conclude a paragraph on the late Miocene cooling period I would rephrase slightly the last part of the sentence so it does not sound that this shift what permanent, which contradict the finding of the paper.

We have reformulated this sentence to specifically state that we are referring to a temporary excursion of the ITCZ away from the Arabian Peninsula in the Late Miocene.

Lines 239-242: “**Higher induced** meridional SST gradients (Fig. 3) provide a mechanism for a **temporary** equatorial **excursion** of the ITCZ away from the Arabian Peninsula **during the Messinian Salinity Crisis in the** terminal Miocene.”

L. 295. Humidty —> humidity

Changed as suggested.

L. 296 and following. While I think the addition of this section is valuable and relevant, I also think it could still be improved. In particular here the authors only quickly referred to effect of orbital changes on monsoon and totally neglect to refer to the link with possible moisture changes / detailed atmosphere dynamics that would be highly relevant for their study (eg. Lines 304-305 ‘...interglacial suggest a substantial increase in precipitation in the Arabian Peninsula in the order of 305 300 to 600 mm a-1, originating mainly from the North African monsoon’,)

We agree with Reviewer 4 that the addition of the orbital driven humidity section is valuable and relevant to this manuscript. We have, as requested, provided further detail to link additional moisture changes and atmospheric dynamical shifts to our results. As already mentioned, we have significantly restructured our discussion and removed the specific section heading about orbital controls and instead we have incorporated this discussion in our new revised implications-driven structure entitled “Recurrent humid episodes”, “Precipitation moisture sources” and “Progressive aridity over the last 8 Ma” sections”.

Some specifically examples include but are not limited to the following:

Lines 130-140: “**Humid episodes in subtropical drylands such as southern Arabia are typically pulsed and short-lived (several to tens of thousands of years) ^{9,24} and in phase with NH summer**

insolation maxima^{9,26}. The amount of incoming solar radiation is predominately influenced by 20 kyr orbital precession cyclicity, modulated by eccentricity²⁷, and to a lesser extent obliquity (40 kyr cycles) and is associated with meridional shifts in the African monsoon belt²⁶. Output from global climate models project that increased summer monsoonal precipitation over the Arabian Peninsula occurs at peak NH high latitude insolation (e.g. at 125 ka), as revealed by a series of time-slice experiments associated with different orbital configurations and global ice volumes²⁸. Consequently, low-latitude insolation is an important pacer of humid episodes in the Arabian Peninsula, the frequency of which is also governed by glacial boundary conditions and the degree of warming in the North Atlantic region.”

Lines 153-161 “...These palaeo- $\delta^{18}\text{O}_{\text{FI}}$ and $\delta^2\text{H}_{\text{FI}}$ values are instead consistent with reported speleothem fossil water data from southern Arabia dominated by a tropical moisture source during recent insolation-driven peak interglacials^{9,24,29}(S-LMWL; Fig. 4b) and are similar to those commonly associated with low-intensity stratiform monsoonal precipitation³⁰. Modelling studies have shown that in Arabia, monsoonal winds bring greater moisture transport from the nearby Gulf of Aden and Arabia Sea source regions³¹ and are attributed to an increase in the zonal wind component over Africa and the Arabian Peninsula during wetter intervals. Further, the resulting boreal summer precipitation³¹ has a lower isotopic composition compared to preindustrial values, that is in part due to an amount effect....”

I think being more precise and making use of the few papers that previously published modeling results centered on the region (even if it's only for the most recent interglacial period) would help to better integrate this part to the paper and probably make an easy link with the 'Reduced tropical precipitation' section. As an example I would refer to the Nicholson et al., (2020) paper on the last 1.1 Ma period in which they really well integrated the understanding of moisture transport and regional climate dynamics at orbital time scale to try explaining the records they have in the region.

We thank Reviewer 4 for this comment and have incorporated this into the revised manuscript. We refer to work from Jennings et al. (2015) who compare the results from several modelling outputs from a series of global climate model experiments set under different orbital configurations and northern hemisphere ice cover, specifically in relation to precipitation changes over Arabia. Further, we also refer to the Herold and Lohmann (2009) who investigated the last interglacial tropical and subtropical African monsoon transport in an isotope-enabled modelling study. Together these show that during the last interglacial the highest precipitation over the peninsula was associated with peak NH high latitude insolation and that the resulting summer monsoonal precipitation in Arabia was significantly isotopically lower compared to preindustrial values. Further, at the same time, there was little precipitation change during boreal winter over the Arabian Peninsula.

Lines 135-138: “Output from global climate models project that increased summer monsoonal precipitation over the Arabian Peninsula occurs at peak NH high latitude insolation (e.g. at 125 ka), as revealed by a series of time-slice experiments associated with different orbital configurations and global ice volumes...”

Lines 157-160: “Modelling studies have shown that in Arabia, monsoonal winds bring greater moisture transport from the nearby Gulf of Aden and Arabia Sea source regions³¹ and are attributed to an increase in the zonal wind component over Africa and the Arabian Peninsula during wetter intervals. Further, the resulting boreal summer precipitation³¹ has a lower isotopic composition compared to preindustrial values, that is in part due to an amount effect.”

L. 308. “Climate modelling of the Tethys suggests” —> “Climate simulation investigating the effect of paleogeography changes in the Paratethys region suggest” or “Climate simulation investigating the effect of paleogeography changes in the Tethys Sea region suggest” would be a more correct formulation.

We have removed all references to the Tethys specifically here in our responses to the previous comments from reviewer 4, so this is no longer relevant

L. 308-309. “a high sensitivity of the African monsoon to orbital forcing began after the Tethys closure by the Late Miocene”. The reference to Tethys closure is confusing as I mentioned above but directly link to what is written in Zhang et al. (2014). The main problem with Zhang et al. paper is that reader would understand from it that Paratethys disappeared during the late Miocene which is more or less correct as there was still megalakes in the region with highly variable size during the Mio-Pliocene (eg. Krijgsman et al. 2010) or that the Tethyan seaway did close during the late Miocene which is not true if one believe recent literature (eg. Bialik et al., 2019). I think this reference is useful when it comes to understanding the effect of paleogeography changes on climate because it uses paleogeography sensitivity tests but becomes less relevant when it comes to timing of such events when considering more recent research on this topic. More correct sentence would therefore only state that ‘those simulations show increased sensitivity of orbital forcing owing to changes in the land-sea mask in the region.’

We have removed this sentence in the previous reorganisation of this paragraph, in response to Reviewer 4’s comments.

L. 311-312. “may potentially be driven by orbital insolation cycles which are more result in more effective precipitation in a ‘warmer’ versus ‘ice-age’ world.”. This part of the sentence read weird and is therefore difficult to understand.

We have also removed this sentence in the previous reorganisation of this paragraph, in response to Reviewer 4’s comments.

Figure 4a. The red-filled circle described as central Arabia in the legend is confusing. I assumed after closer look to the figure it refers to the colored circle colors by age (from blue to red). I suggest the authors to rather use a blank-filled circle to avoid confusion and to move the label just above the age color scale so it’s easier to read. The size of the age color scale in the figure could be reduced to find space.

We have amended this figure as requested and moved the legend for fossil water above the colour scale and replaced the red filled circle with a black filled circle in both Panels a and b.

In addition, the symbols for the Northerly and Southerly source regions are very difficult to see on this figure.

We have amended this figure as requested and doubled the stroke thickness of the Northerly and Southerly precipitation symbols. Further, we used a dark yellow hue for the “AG” source so it was easier to see in the figure (see amended Fig. 4 above).

L 371-372. Because the authors decided to remove the biogeography focus of the paper I am not sure this conclusion holds anymore and I suggest changing it.

We have already changed this sentence in a previous response to reviewer comment above. The new sentence is as follows:

Lines 279-281: “Our record therefore constrains both the timing and frequency of such climatic amelioration phases, providing insight into critical periods for which previously there was no data.”

Ref 53 is incomplete and it seems there is an issue with reference numbering between Main text and Methods part.

We have checked and updated all the references within the manuscript.

S.Fig 9. I suggest the authors to at least add the coordinates in the caption for the oceanic sites that are used to derive gradients so one can directly understand what those gradients represent. Adding a small map with site location would be a better alternative though.

We have added the coordinates to the caption as suggested. We trialled adding an additional map to this figure but it unfortunately looked too crowded, so we preferred to just add the coordinates to the caption. Note that the Supplementary Figures have now been moved to Extended Data.

Lins 1347-1351: "Extended Data 6: North Hemisphere Sea Surface temperature (SSTs). The Arabian sea SSTs (site 722; 16°37.31'N, 59°47.76'E), North Atlantic SSTs (site 982; 57°30.05'N, 15°52.03'W), North Pacific (sites 887; 54°21.921'N, 148°26.765'W, and 883; 51°12' N, 167°46' E), and Norwegian Sea (site 907; 69°14.989'N, 12°41.894'E) from the compilation in Herbert et al. ¹ and references therein."

Lines 1353-1361: "Extended Data 7: Meridional Temperature Gradients (MTGs) in the northern (a-d) and southern hemispheres (e-f) ² over the last 8 Ma compared to speleothem growth periods in this study (h) as well as the northern Arabia in the Negev Desert ³ (g) and southern Arabia speleothems in Yemen and Oman ⁴ with the 95% confidence interval uncertainty indicated by the horizontal error bars. (i). Northern hemisphere MTGs for sites 982 (57°30.05'N, 15°52.03'W) (a), 907 (69°14.989'N, 12°41.894'E) (b), 883 (51°12' N, 167°46' E) (c), 887 (54°21.921'N, 148°26.765'W) (d), and southern hemisphere sites 594 (45° 31'24.6'S, 174° 56' 52.8'E) (e) and 1088 (41° 8' 9.96'S, 13° 33' 46.08'E) (f) from the data compiled in Herbert et al. ¹."

References:

Taher, S.A., 2004. Estimation of Potential Evaporation: Artificial Neural Networks Versus Conventional Methods, *Journal of King Saud University - Engineering Sciences*, 17;1.

Vignaud, P., Durringer, P., Mackaye, H. *et al.* Geology and palaeontology of the Upper Miocene Toros-Menalla hominid locality, Chad. *Nature* **418**, 152–155 (2002).
<https://doi.org/10.1038/nature00880>

Lebatard, A., Bourles, D.L. , Durringer, P., Jolivet, M. , Braucher, R., et al.. Cosmogenic nuclide dating of Sahelanthropus tchadensis and Australopithecus bahrelghazali: Mio-Pliocene hominids from Chad.. *Proceedings of the National Academy of Sciences of the United States of America*, 2008, 105 (9), pp.3226-3231.

Jennings, R. P. *et al.* The greening of Arabia: Multiple opportunities for human occupation of the Arabian Peninsula during the Late Pleistocene inferred from an ensemble of climate model simulations. *Quaternary International* **382**, 181-199 (2015). [https://doi.org:https://doi.org/10.1016/j.quaint.2015.01.006](https://doi.org/https://doi.org/10.1016/j.quaint.2015.01.006)

Herold, M. & Lohmann, G. Eemian tropical and subtropical African moisture transport: an isotope modelling study. *Climate Dynamics* **33**, 1075-1088 (2009).
<https://doi.org:10.1007/s00382-008-0515-2>

Review of Markowska et al. “Recurrent humid phases in Arabia over the past 8 million years” submitted to Nature 2023-05-07879

The manuscript by Markowska et al. presents an extensive data set from Saudi-Arabian speleothems including new U-Th and U-Pb chronology, stable isotope ($\delta^{18}\text{O}$) and fluid inclusion data together with a thorough literature study of first appearances dates of mammals in Eurasia and Africa. Based on the timing of the absence and presence of speleothem growth, the authors defined five humid periods in Central Arabia over the last 8 million years that seem to coincide with faunal dispersals between Africa and Eurasia.

The speleothem $\delta^{18}\text{O}$ and fluid inclusion data set are impressive. They found a clear increasing trend in their $\delta^{18}\text{O}$ data suggesting that this region became progressively drier towards the Pleistocene. $\delta^{18}\text{O}$ and $\delta^2\text{H}$ fluid inclusion analyses were then used to identify the atmospheric moisture source. This is very interesting palaeoclimatological data providing new insights into the climatic conditions during the Miocene to Pleistocene in Central Arabia – an area for which previously limited palaeoclimatological information from terrestrial records was available, particularly for the Miocene to Early Pleistocene.

The main implications of this manuscript, though, are linked the five identified humid periods based on the speleothem ages. The authors report an extensive collection of mammals first appearances dates which seem to coincide with the Central Arabian humid periods. They ultimately suggest that due to periodic climatic ameliorations, the Saharo-Arabian desert acted as an important corridor for faunal dispersals between Africa and Eurasia. While the implications of this narrative are important and of immediate interest to people from several disciplines, they stand or fall with the quality of the chronology and the statistical significance of the defined humid periods – and, unfortunately, there are issues with the U-Pb data set.

The two major issues are: 1) the sample size for ages beyond ~500 ka is only 22 and seems too small to identify five humid periods over such a long period of time (8 million years) with enough confidence, and 2) the disequilibrium correction using the initial $^{234}\text{U}/^{238}\text{U}$ activity ratios to calculate the final U-Pb ages was not done adequately, which could drastically change the timing of the identified humid periods. Both these issues undermine the main conclusions of this paper, i.e., that the humid phases coincided with major faunal dispersals between Africa and Eurasia.

I support the palaeoclimatological interpretations based on the identified trend in the $\delta^{18}\text{O}$ and regarding the moisture source based on fluid inclusion data. I am very positive about this data and believe they are of immediate interest to the palaeoclimate community and beyond. The chronological data set is impressive in the sense that, until this data set, only a handful of speleothem ages older than 500 ka existed from this region, and furthermore, this study is the first to report ages for this area beyond the Pleistocene.

However, due to the small sample size and issues with the disequilibrium corrections of the U-Pb ages, I cannot recommend this manuscript for publication in Nature in the current form. I will explain the abovementioned issues in more detail below and provide recommendations to the authors on how to address them and improve their manuscript.

Major issues:

Sample size

1. For the conclusions and implications of this paper, the absence of speleothems is just as important as their presence (because, simply put, present = wet, absent = dry). The U-Pb ages from this area are novel and very interesting, but from a statistical point of view, the sample

size is too small to identify statistically significant humid phases. 8 million years is a very long time – do you have any statistical means to evaluate whether it is true what you see, or what you don't see (=absence of speleothems)? How confident are the authors that the 22 U-Pb ages truly represent the speleothem growth phases in this area? A recent speleothem-based study (Weij et al. 2020) investigated this issue and from their simulations it is obvious that 22 ages are not enough to distinguish between randomness and the true signal. The authors have not discussed the effect of sample size on the statistical significance of the identified speleothem growth periods, but this should really be addressed.

2. Another issue is a potential preservation bias of speleothems, i.e., destruction of speleothems with time. Both Scroxton et al. (2016) and Weij et al. (2020) discuss an exponential decay or “half-life” of speleothems with age, meaning that the further back in time, the less likely a speleothem would have been preserved. Therefore, the absence of speleothems does not necessarily mean that there really weren't any speleothems growing at that time. What does that mean for the speleothem data set presented in this study?

Both these aspects should be addressed together with why the authors believe that the signal seen in the speleothems presented here is in fact the true absence of speleothems due to unfavourable climatic conditions.

U-Pb age calculation

1. Initial $^{234}\text{U}/^{238}\text{U}$ disequilibrium correction of the U-Pb ages: the authors have not directly measured the initial $^{234}\text{U}/^{238}\text{U}$ activity ratio in their U-Pb dated samples – why not? The final disequilibrium correction is crucial and can affect the final age quite significantly. For samples older than 3-4 Ma, there probably won't have any measurable $^{234}\text{U}/^{238}\text{U}$, but samples between 0.5 and 3 Ma probably do. In general, it is best practise to have corresponding residual $^{234}\text{U}/^{238}\text{U}$ measurements for U-Pb dated samples to back calculate the initial $^{234}\text{U}/^{238}\text{U}$ activity ratio. For any older samples without measurable $^{234}\text{U}/^{238}\text{U}$, one could then use a weighted average or trendline-based estimate from the younger U-Pb dated samples, depending on whether a trend with age is observed or not. Particularly for a paper like this with important implications that entirely depend on the chronology, I would expect that the U-Pb ages have a corresponding $^{234}\text{U}/^{238}\text{U}$ measurement. Instead, the authors have used a weighted average initial $^{234}\text{U}/^{238}\text{U}$ activity ratio obtained from the solution U-Th dated samples. The problem with this approach is that there is an obvious trend in initial $^{234}\text{U}/^{238}\text{U}$ with U-Th age, as seen in the plot below:

This trend suggests that the initial $^{234}\text{U}/^{238}\text{U}$ could have been much higher for older U-Pb ages than the weighted average currently used to perform the disequilibrium correction. It follows that the U-Pb ages then are actually younger than currently reported. This is very problematic. To demonstrate why, I have fitted a linear trendline through the data (excluding the one outlier) and recalculated the U-Pb disequilibrium ages using a trendline-based estimate initial $^{234}\text{U}/^{238}\text{U}$ activity ratio (in DQPB software, Pollard et al. 2023). I was able to reproduce the isochrons and most of the equilibrium ages with their errors and MSWD using a model 1 fit type (see table on the final page of this document). The authors should check

why some ages/age errors/MSWDs are different from the ones recalculated, perhaps some typos?

More importantly, the disequilibrium U-Pb ages I recalculated using a trendline-based value for initial $^{234}\text{U}/^{238}\text{U}$ are drastically lower than calculated by the authors (again see table one the final page of this document). This means that the humid periods will shift and do not coincide so well anymore with NH cooling periods nor with faunal dispersals, see figure below. If this is true, then the conclusions of this manuscript will have to be revisited and may not hold anymore.

[FIGURE REDACTED]

2. Fit-type U-Pb isochrons: first, the authors did not specify which fit-type model was used to calculate the U-Pb isochrons – this should be included for reproducibility of the ages. After recalculating U-Pb equilibrium ages, I was able to reproduce them using Model 1 fit type, so I assume it is this one the authors have used?

Model 1 result different age uncertainties than for example the spine fit type. The spine fit type is generally more robust than the classical model 1 fit type (see Powell et al. 2022 and Pollard et al. 2023). When using the model 1 or spine fit type, the ages themselves differ only within a few percent of each other, so that is fine. However, since the age uncertainties will be different for model-1 or spine ages, this will affect the timing of the humid periods if they are defined by their age uncertainties. The choice of fit type is ultimately up to the authors, but the issue above should be considered and addressed, and their choice of fit type should be specified and explained.

Humid phases and the terminal Miocene dry period:

1. I'm wondering how exactly the humid periods defined? In the text, they seem to be defined by their mean ages, but in the figures by their age uncertainties. The authors should be consistent and also explain their choice in the text (main or methods).
2. The small sample size and issues with the disequilibrium correction of the U-Pb ages have both raised the following question: how sure are the authors be that the speleothem growth phases we see are discrete events modulated wet climatic conditions? And that the absence of speleothems really means that there were no speleothems growing at that time? They could be absent in the dataset because they really weren't there, or simply missed because of a small sample size, or destroyed/buried due to a preservation bias.
3. This leads to the section addressing "*the Terminal Miocene dry period*": again, the underlying chronological data aren't very strong to support the conclusions drawn from it. The dataset includes 22 speleothem ages, of which 10 are older than 3 Ma. What makes the authors sure that there really weren't any speleothems growing during the terminal Miocene period? Also,

if the final disequilibrium-corrected ages will change, then this interpreted “dry period” will not coincide with the terminal Miocene anymore.

Recommendations

This manuscript has great potential, but with the current chronological data set, it does not have the statistical foundation to support the manuscript’s main claims. I suggest the following options:

- Disequilibrium correction of the U-Pb ages: the authors are strongly encouraged to measure the residual $^{234}\text{U}/^{238}\text{U}$ in their U-Pb dated samples using U-Th solution methods, at least for the samples between 0.5 and 3 Ma, and back calculate the initial $^{234}\text{U}/^{238}\text{U}$ for these samples. They should then correct the U-Pb samples with their corresponding initial $^{234}\text{U}/^{238}\text{U}$ value. The authors should then plot those values against age to look for any trends. If there is a trend, I suggest fitting a trendline through this data and, via extrapolation, using trendline-based initial $^{234}\text{U}/^{238}\text{U}$ estimates to correct the U-Pb samples for which no residual $^{234}\text{U}/^{238}\text{U}$ could be measured. If there is no trend, then an average weighted initial $^{234}\text{U}/^{238}\text{U}$ of the U-Pb samples, or possibly the entire data set, could be used.
- Sample size: unfortunately, there may be no quick solution for this issue if samples aren’t readily available. To identify humid periods over this time scale with confidence, I think the sample size should be much larger (triple, quadruple, more?). The current 22 U-Pb ages were measured in one day, so if there are more samples, then the nowadays’ rapid laser ablation dating techniques would allow to quadruple the data set over three nights. This, of course, also depends on the authors’ timeline and financial means.
- I think the current U-Pb data set is too small to interpret the speleothem ages as discrete humid periods with enough confidence and therefore cannot strongly be linked to faunal dispersals. However, the interpretations based on the oxygen isotopes and fluid inclusions are very interesting and could stand on their own. The authors could decide to leave out the part regarding humid periods linked to faunal dispersals, refocus the manuscript on the palaeoclimatological aspects and resubmit this data – whether to Nature or to a different journal is at the editor’s discretion.

General comments:

Line 87: “*dispersals out of Africa*”, preferable to use different wording, for example “dispersals from Africa to Eurasia”, or something similar.

Lines 127-129 “*desert speleothems*” and “(1) regional precipitation greater than ± 300 mm/a”: not sure if this hard number of 300mm/year is necessary and whether this generalization is true for all deserts across the globe. This number was based on a study in Israel, how sure are the authors that this number also applies to Saudi Arabia? And other deserts? It also depends on the seasonality of rain fall, for example, if it mostly rains in summer, then most of it will evaporate. I suggest reframing this statement and perhaps restricting this statement either to central Arabian deserts, or using a more generalized statement saying something along the lines that sufficient regional precipitation is required.

Lines 137-138: while it is true that U-Pb dating of speleothems is less frequently applied than the U-Th chronometer (arguably, partly due to a preservation bias of older speleothems in the geological record), there certainly have been other pioneering studies utilizing the speleothem U-Pb dating technique, published in high profile journals, e.g.: Polyak et al. 1998 (Science), 2008 (Science); Meyer et al. 2009 (QSR), 2011 (Geology); Bajo et al. 2012 (Quaternary Geochronology), 2020 (Science), Pickering et al. 2018 (Nature), Engel et al. 2020 (Geology), Weij et al. 2022 (Nature)

Communications Earth & Environments). The current number and choice of citations here doesn't seem to do justice to the years of work the abovementioned studies have put into U-Pb dating of speleothems. The manuscript's chronological data set is novel, but the dating technique is itself not; it is well-established and has been routinely applied.

Line 142: are these 76 ages from 76 different growth phases or do some growth phases within a speleothem have multiple ages?

Lines 146 Figure 2 panel b "*African palaeo-environments*": perhaps a better label would be "hominin evolution" or something similar.

Line 154: First Appearance Dates in figure 2, does each bar represent an age range of FADs reported in different studies from different sites?

Line 158: please define red diamonds and yellow squares as in figure 3.

Lines 159-160: I appreciate the attempt to account for varying age uncertainties, however, since the resolution of the identified humid periods are similar within the U-Th and U-Pb domains, it might be better to keep a similar bin width for the U-Th ages. That way it's more obvious for the reader that the data density within the U-Th domain is much higher and it won't really affect the conclusions drawn from this figure.

Lines 160-161: I suggest removing the 2-point binomial smoothed average line for the U-Pb ages unless the U-Pb age errors is considered. However, I suspect that when the errors are taken into account, it will result into quite large uncertainties in the timing of the peaks.

Line 171: if this period is resolved at glacial-interglacial scale, what implications does that have for the previous humid periods, for which less data are available? Possibly, there were many more speleothems growing then, especially with a progressive aridity trend over time, but due to a preservation bias, we don't find these older speleothems as readily available as younger speleothems.

Lines 177-178 "*despite increased recharge to groundwater reservoirs, only a slight increase in local precipitation occurs*": this statement seems a bit premature here or needs more explanation. Based on what information could the reader know that recharge to groundwater reservoirs was increased, and that local precipitation only increased slightly?

Line 181: please define *aridity*: are we looking at precipitation, soil moisture, humidity, run-off, precipitation-evaporation? There are many definitions, and they all mean something slightly different in terms of wetness vs. dryness.

Line 182: I suggest referring to figure 3 here already so that one can immediately look at the values when reading a statement like this.

Lines 190-198: I appreciate this link to speleothem fabric – often overlooked even though it can convey important information about drip rate and water supply, which are ultimately linked to the hydroclimate above the cave.

Line 198 "*caused by a decrease in precipitation*": and what about the effect of evaporation?

Line 201 Figure 3 panel d "*increasing aridity*": what is this blocky line? It now looks like the thickness of the little blocks are related to the peaks in dust, but it's probably just the line pattern? I suggest using a thinner line with a clearer arrow.

Line 213: please add a similar description of the U-Th ages to figure 2.

Line 234: “Fig 4”: Are you referring to panel b here? Would be useful to add that to the figure reference. “modern meteoric waterline derived from a tropical moisture source (Indian Ocean)”: which one in your figure 4?

Line 235 “26 degrees N”: is this where the study site is? It looks like it in your figure 1. If so, I suggest adding a few words to this sentence to clarify that.

Lines 251-252: I think it reads better if you refer to figure 3 after “during periods of lower meridional SST gradients”, otherwise it reads like there should be a plot of the global climate model simulations.

Line 254-255: we are now in the “glacial-interglacial world” when dust transport was mainly increased during glacial periods. For long, the palaeoclimate community believed that an increase in dust meant more arid conditions. However, Muhs (2013), for example, also showed that other factors play a role: increased source areas due to more continental exposure and more shrub-like vegetation, stronger winds, greater production of dust-sized particles. Therefore, I’m not entirely convinced that increased dust transport from the Sahara Desert is strong evidence the Saharo-Arabian aridification.

Line 274: Figure 4 panel a and c “Fossil Water Central Arabia”: where is this plotted in panel a and c?

Line 325 “the aforementioned absence of speleothems”: again, how sure are you that this absence is statistically true?

Line 551: are those 28 sub-samples for U-Th solution work? From how many speleothem samples? And are they from distinct growth phases separated by hiatuses or are some growth phases dated multiple times?

Line 559: what is the error on the $^{232}\text{Th}/^{238}\text{U}$ weight ratio of 3.8 and was this error propagated to the final age uncertainty?

Line 562: again, from how many speleothem samples? And are they from distinct growth phases separated by hiatuses or are some growth phases dated multiple times?

Lines 575-576: see comment about line 559.

Lines 582-583: from the supplementary file I gather that this is done at Frankfurt University, but please specify where the analyses were done.

Line 596: I assume the ages were calculated in Isoplot? Which fit type was used and could you briefly elaborate on the choice of fit type? This information should be added to the Methods.

Lines 599-600: please provide a plot of the initial $^{234}\text{U}/^{238}\text{U}$ ratios against age like I have done in the main comments.

Line 605: I assume this is done at the MPIC but please specify where the analyses were done.

Lines 610 and 614: I see the $\delta^{13}\text{C}$ data in the supplementary file, but it is not discussed in the manuscript. Could you briefly explain why not?

Line 625: again, I assume this was done at the MPIC but please specify in the Methods.

Supplementary Methods

Lines 173-174: I suggest referring to SFig 10 here.

Lines 206-208: this could be true, but again I'm wondering about the effect of your sample size and speleothem preservation?

Line 214 "*the blue dashed line ... used as the cutoff for **significant** interglacial phases*": please specify what significant means here. Is it statistically significant? Or is the position of this line the author's subjective choice? Please briefly explain.

Lines 334-335: this is good to point out: the importance of understanding the absence of something. Here it is related to the hiatuses in animal movement, but more important for this paper is to first understand the meaning of the absence of speleothems. So again, please consider and discuss speleothem preservation and the sample size. Why are the authors sure that the speleothem growth phases are true?

Supplementary U-Pb data

U-Pb calculated ages sheet

Footnote #1: please specify which software was used to calculate the ages and what fit type was used.

Footnote #6: please also add the uncertainty of the initial $^{234}\text{U}/^{238}\text{U}$ activity ratio here.

Footnotes #7 and #8: I'm not following here. Where does this activity ratio of 0.885 come from and what is it used for?

Sample raw data sheet

I see several measurements were excluded. After replotting the data and examining the isochrons, I see that most of these excluded measurements are obvious outliers, however, some are not. Could you please add a sentence explaining the reader why some measurements were excluded?

Other comments:

The figures in the manuscript are well done, but could you please check the colour schemes for suitability for colour blind people?

The supplementary files contain "raw data", however, with "raw data" I'm thinking of the unprocessed numbers that come off the mass spectrometer. The data presented in the supplementary files are corrected for systematic and/or statistical uncertainties, so are technically not "raw". I suggest using different wording here.

[TABLE REDACTED]